# TIME AFTER TIME: DEEP-Q EFFECT ESTIMATION FOR INTERVENTIONS ON WHEN AND WHAT TO DO

**Yoav Wald**[1]      **Mark Goldstein**[1]   **Yonathan Efroni**[2]   **Wouter A.C. van Amsterdam**[3]
`yoav.wald@nyu.edu`

**Rajesh Ranganath**[1]

## ABSTRACT

Problems in fields such as healthcare, robotics, and finance requires reasoning about the value both of what decision or action to take and when to take it. The prevailing hope is that artificial intelligence will support such decisions by estimating the causal effect of policies such as how to treat patients or how to allocate resources over time. However, existing methods for estimating the effect of a policy struggle with *irregular time*. They either discretize time, or disregard the effect of timing policies. We present a new deep-Q algorithm that estimates the effect of both when and what to do called Earliest Disagreement Q-Evaluation (EDQ). EDQ makes use of recursion for the Q-function that is compatible with flexible sequence models, such as transformers. EDQ provides accurate estimates under standard assumptions. We validate the approach through experiments on survival time and tumor growth tasks.

## 1 INTRODUCTION

Sequential decision-making is common in healthcare, finance, and beyond (Chen et al., 2021a; Upadhyay et al., 2018). In hospitals, medical professionals administer treatments at different times based on the evolving observations of a patient's condition; in financial markets, traders execute orders based on sequential information flows. Algorithmic decision support systems can optimize these processes by evaluating different *policies* with respect to their expected outcomes. Estimating the difference in expected outcomes between various policies is a causal effect estimation question (Chen et al., 2021a; Joshi et al., 2025; van Amsterdam et al., 2024). This question involves several future treatment decisions taken at varying time points, hence it is a sequential decision-making problem.

Formally, this problem falls within the framework of off-policy evaluation (Fu et al., 2021; Uehara et al., 2022). A defining feature is that timings of observations and treatments are irregularly spaced, represented by a stochastic point process with intensity $\lambda$, whereas the *type* of the treatments at those times are specified as the marks of this process, governed by a distribution $\pi$. Unlike traditional formulations of off-policy evaluation that focus on action types, here the times must be accounted for, as they have a large effect on the outcome. This formulation is relevant to many decision-making scenarios, for example, in transplants, where it is often desirable to delay treatment as much as possible to lower the risk of complications. However, delaying too much can result in a deterioration in the patient's condition. Here, the type of treatment is fixed and what matters is the timing.

*Estimating the effect of intervention on treatment timing* is a crucial part of evaluating sequential policies. With irregular times, frameworks for sequential decision-making that discretize time can be problematic, as discretization can be inaccurate or inefficient and requires choosing appropriate time scales. Further, length scales for decision-making within a single trajectory can vary dramatically. Consider for example a patient with heart failure; such a patient may be stable for months or years with occasional treatment adjustments made during visits to the cardiologist. However, at some point, they may experience acute decompensation (Felker et al., 2011), which requires rapid treatment and

---

[1]New York University
[2]Meta
[3]University Medical Center Utrecht

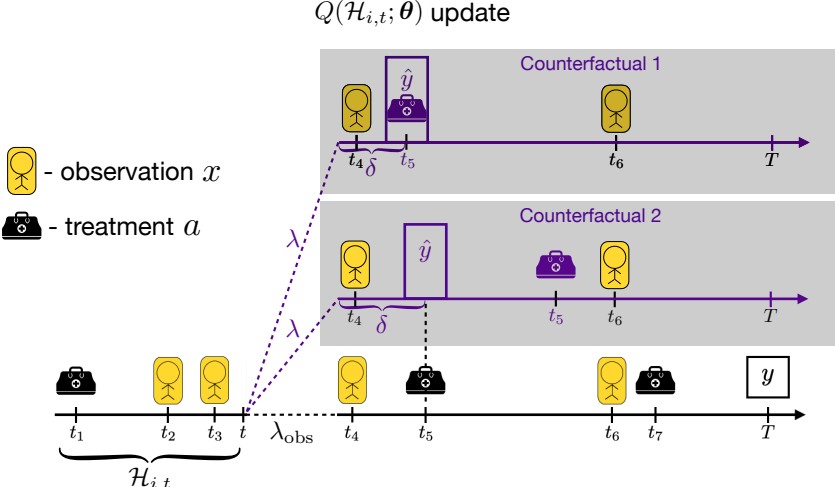

Figure 1: A summary of EDQ. Conditional expectations of the outcome, $\mathbb{E}_P[Y|\mathcal{H}_t]$, are estimated by $Q_t(\mathcal{H}_t; \boldsymbol{\theta})$. Given an observed trajectory $\mathcal{H}_i$ sampled under the training policy $\lambda_{\text{obs}}$ (bottom trajectory), we fit values $Q_t(\mathcal{H}_{i,t}; \boldsymbol{\theta})$ by regressing them on a label $\hat{y} := Q_{t+\delta}(\widetilde{\mathcal{H}}_{i,t+\delta})$ determined by a "counterfactual" trajectory $\widetilde{\mathcal{H}}$ sampled from the target policy $\lambda$. $\delta$ is the earliest disagreement time between the observed and counterfactual trajectories. It is either the time of the next observed treatment (counterfactual 1) or that of the treatment under the target policy (counterfactual 2).

intensive monitoring during hospitalization. Existing methods for continuous-time causal inference do not scale gracefully, since they solve complex estimation problems, such as integrating importance weights across time (Røysland, 2011). Those methods that scale to high-capacity models and large datasets do not handle dynamic policies (i.e., policies that take past states into account) and are implemented with differential equation solvers (Seedat et al., 2022), restricting architectural choices.

In this work, we give two methods for off-policy evaluation with irregularly sampled data. Our contributions are as follows:

- We define off-policy evaluation with *decision point processes* and develop Earliest Disagreement Q-Evaluation (EDQ), a model-free solution to the problem. While other methods are intractable in high dimensions or are limited to static treatments, EDQ eliminates these restrictions. EDQ is based on direct regressions and dynamic programming, which makes it easily applicable to flexible architectures including sequence models such as transformers.

- In Theorem 1, we prove that EDQ is an empirical estimator of the correct policy value. The estimator produces accurate causal effects under assumptions on causal validity based on Røysland (2011); Røysland et al. (2022).

- We validate the efficacy of EDQ through an experimental demonstration on time-to-failure prediction and tumor growth simulation tasks. For these tasks, we implement a transformer-based solution. The results show EDQ's advantage relative to baselines that rely on discretization.

We define the estimation problem, develop a solution, discuss related work and validate empirically.

## 2 OFF-POLICY EVALUATION WITH DECISION POINT PROCESSES

Consider a decision process defined by a marked point process $P$ (Andersen et al., 2012; Snyder and Miller, 2012) over observations (which take values in $\mathcal{X}$), treatments (in $\mathcal{A}$ respectively) and real outcomes. We are interested in estimating an overall quantity $Y \in \mathbb{R}$ that is a function of a sequence $Y_1, Y_2 \dots$ of observed rewards. For convenience, we let $Y = \sum_{k=1}^{\infty} Y_k$, where $k$ is an index for observed outcomes along the trajectory.[1] Though, the methods extend to other outcome functions like discounted future outcomes. We assume the number of rewards in the segment $[0, T]$ is finite.

---

[1] We define $Y_k(\omega) = 0$ for an event $\omega$ where $k$ is larger than the number of outcomes in the trajectory.

**Marked point processes.** A marked point process is a distribution over event times, along with distributions over *marks*, or details of the events at each time (i.e. treatment times and which treatment was given). We consider multivariate counting processes $N(t) = (N^a(t), N^b(t), \dots)$ on the time interval $[0, T]$. For a univariate process, e.g. $N^b$, $N^b(t)$ is the number of events of type $b$ until time $t$. A trajectory of event times and their marks is a set $\mathcal{H} = \{(t_0, \mathbf{e}_0), (t_1, \mathbf{e}_1), \dots, (t_n, \mathbf{e}_n)\}$. We denote events up to time $t$ by $\mathcal{H}_t = \{(t_k, \mathbf{e}_k) \in \mathcal{H} : t_k \leq t\}$ and $\mathcal{H}_{(t, t+\delta]}$ for events in the interval $(t, t + \delta]$. We use $\mathcal{H}^b$ to refer to events of type $b$ on the trajectory and $\mathcal{H}^{\backslash b}$ for events of all types other than $b$.

Finally, we assume intensity functions $\lambda(t|\mathcal{H}_t) = \mathbb{E}[dN(t)|\mathcal{H}_t]$ for processes exist, $N(t)$ is almost surely finite for any $t \in [0, T]$, and that the process can depend on its own history. That is, the filtration is the $\sigma$-algebra generated by random variables $N(t)$ and their marks (Aalen et al., 2008).

## 2.1 PROBLEM DEFINITION

We follow notation from the RL literature (Upadhyay et al., 2018). We begin with the data generating process and then summarize our goal of inferring causal effects. This involves off-policy evaluation under a distribution $P$, while observing samples from $P_{\text{obs}}$.

**Definition 1.** *A marked decision point process $P$ is a marked point process with observed components $N^e$ for $e \in \{x, y, a\}$ that have corresponding intensity functions $\lambda^e$, and mark spaces $\mathcal{X}, \mathbb{R}, \mathcal{A}$, and a multivariate unobserved process with intensity $\lambda^u$. By default, we omit unobserved events from the trajectories $\mathcal{H}$, hence $\mathcal{H} = \{(t_0, \mathbf{e}_0), (t_1, \mathbf{e}_1), \dots, (t_n, \mathbf{e}_n)\}$ where $\mathbf{e}_k \in \{\mathcal{X} \cup \mathcal{A} \cup \mathbb{R}\}$. The intensity function and mark distribution $\lambda^a(t|\mathcal{H}_t), \pi(A_t|\mathcal{H}_t)$ are called the policy. The mark distributions for $X, Y$ are denoted by $P_X(\mathbf{x}_t|\mathcal{H}_t), P_Y(y_t|\mathcal{H}_t)$.*

**Off-policy evaluation.** We are given a dataset of $m$ trajectories, where trajectory $\mathcal{H}_i$ has $n_i$ observations: $\{(t_{i,k}, \mathbf{e}_{i,k})\}_{i \in [m], k \in [n_i]}$. These are sampled from an observed decision process $P_{\text{obs}}$ with policy $(\lambda^a_{\text{obs}}, \pi_{\text{obs}})$. Treatment times are samples from a counting process with intensity $\lambda^a_{\text{obs}}$ and treatments at those times are sampled from $\pi_{\text{obs}}$. We reason about outcomes when $(\lambda^a_{\text{obs}}, \pi_{\text{obs}})$ is replaced with a target policy $(\lambda^a, \pi)$, with other processes in $P_{\text{obs}}$ fixed. The resulting decision process is denoted $P$, and our goal is to estimate the expected future outcome $\mathbb{E}_P[Y|\mathcal{H}_t]$ for all $t \in [0, T)$ and $\mathcal{H}_t$ in the natural filtration associated with the point process.

**When to treat.** To simplify notation, we omit the marks $\pi(A_t|\mathcal{H}_t)$ and focus on intensities $\lambda^a(t|\mathcal{H}_t)$. That is, we explore interventions on when to treat (*medicine weekly or monthly?*) instead of how to treat (*which medication?*). Technically, "when" is the more challenging and underexplored part of the problem, and solutions can be easily extended to incorporate interventions on $\pi$ using existing methods (Chakraborty and Murphy, 2014; Li et al., 2021). In the ASCVD example, this corresponds to reasoning about questions like: "Consider prescribing statins to patients with characteristics $\mathcal{H}_t$, whose LDL cholesterol is below 180 mg/dl up to time $t$. What would be the expected change in 10-year ASCVD risk if going forward, we prescribe a daily dose of statins for patients whose LDL cholesterol goes above 180 mg/dL, instead of the existing policy followed in the population?".

## 2.2 ROADMAP TO IDENTIFIABILITY VIA LOCAL INDEPENDENCES

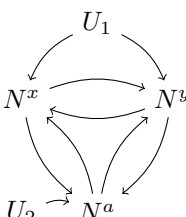

Figure 2: The assumed local independence graph for a decision point processes, where our estimand is identifiable from observed data $(N^x, N^a, N^y)$.

The goal of this section is to elucidate the conditions under which the algorithm we present in section 3 estimates valid causal effects. We briefly summarize the essential conditions and supplement this summary in appendix D. Our assumptions to ensure identifiability of causal estimands follow Didelez (2008); Røysland (2011); Røysland et al. (2022), who study graphical models for point processes. In this setting, where the goal is to intervene on $N^a$ and estimate $E_P[Y|\mathcal{H}_t]$ under $P$ rather than $P_{\text{obs}}$, in the presence of unobserved processes $U$, Røysland (2012) define and analyze the following notions:

- A graphical condition called *causal validity* ensures that changing the treatment intensity from $\lambda^a_{\text{obs}}$ to the interventional $\lambda^a$, while changing no other intensities, changes the joint distribution from $P_{\text{obs}}$ to $P$. A graph may not be causally valid when it contains unobserved variables $U$.

- *Local independence* (Aalen, 1987; Schweder, 1970) adapts sequential exchangeability (Hernan and Robins, 2023; Robins, 1986) to continuous time. It is an asymmetric form of independence where one process' intensity may depend on the history of another, but not vice versa.

- A certain set of local independences together are referred to as *eliminability* (a generalization of the backdoor criterion), which implies casual validity, even with unobserved variables $U$.

Consider the graph in Figure 2, where an edge means that the history of the source node affects the future of the target node (Didelez, 2008). It is possible to show that it satisfies eliminability (see appendix D). To understand this condition, we start with the basic local independence requirements.

**Definition 2.** *For a multivariate process $N(t) = (N^a(t), N^b(t), \ldots)$ on variables $V$ we say that $N^a$ is locally independent of $N^b$ given $N^{\backslash b}$, or $N^b \not\to N^a | N^{\backslash b}$, if $\lambda^a(t|\mathcal{H}_t) = \lambda^a(t|\mathcal{H}_t^{\backslash b})$. A graphical local independence model $(\mathcal{P}, G)$ is a class of processes $\mathcal{P}$ on $V$ and directed graph $G = (V, E)$, such that $(b \to a) \notin E \Rightarrow N^b \not\to N^a | N^{\backslash b}$ holds for all $P \in \mathcal{P}$.*

This condition means that the intensity of a process only makes use of certain information from other processes. Røysland et al. (2022) package together the set of local independences that imply causal validity under the name *eliminability*, defined below. They use a graphical criterion that is akin to using d-separation in graphical models, while we state the conditions in terms of the implied functional independencies of intensities. We expand on this in appendix D.

**Definition 3.** *Let $U$ be the set of unobserved variables. Suppose $U$ can be written as a sequence $(U_1, \ldots, U_K)$ such that for each $k$, either*

- *$(N^y, N^x, U_{>k})$ is locally independent of $U_k$ given $(N^x, N^y, N^a, U_{>k})$, or*

- *$N^a$ is locally independent of $U_k$ given $(N^x, N^y, N^a, U_{>k})$.*

*Then the graph is said to satisfy Eliminability. Here, for sets $A, C$ and variable $b$ we say $A$ is locally independent of $b$ given $A \cup C$ if each $a \in A$ is locally independent of $b$ given $A \cup C \setminus \{a\}$.*

We summarize that *causal validity* holds under the *local independences* that satisfy *eliminability*. We assume the graph in Figure 2, which satisfies these assumptions. We also assume mutual independence of increments of all processes at a one time to rule out instantaneous effects. We refer to validity and this independence together as **ignorability**, in accordance with existing terminology on confounding.

**Assumption 1.** *Ignorability (in continuous time) is satisfied when:*

1. *the graph satisfies causal validity,*

2. *the increments of features, treatments, and outcome are mutually independent given the history, i.e., $((dN^x(t), X_t) \perp\!\!\!\perp (dN^a(t), A_t) \perp\!\!\!\perp (dN^y(t), Y_t))|\mathcal{H}_t$.*

In addition to ignorability, we require a second, standard assumption, **overlap**, for the conditional expectations we estimate to be well-defined. Recall that the interventional distribution $P$ is defined by replacing the treatment distribution in $P_{\text{obs}}$, i.e., replacing $\lambda^a_{\text{obs}}$ with $\lambda^a$.

**Assumption 2.** *Overlap is said to hold between the observational and interventional distributions, $P_{\text{obs}}$ and $P$, if $P$ is absolutely continuous with respect to $P_{\text{obs}}$, denoted by $P \ll P_{\text{obs}}$.*

Ignorability and overlap are the core assumptions that allow identification in our setting. Under these assumptions, we can now present algorithms for estimating causal effects in continuous time.

## 3 MODEL FREE OFF-POLICY EVALUATION FOR DECISION POINT PROCESSES

To estimate $\mathbb{E}_P[Y|\mathcal{H}_t]$ for times $t$ and $\mathcal{H}_t$ that overlap with $P_{\text{obs}}$, we express the expectation recursively as a function of expectations $\mathbb{E}_P[Y|\widetilde{\mathcal{H}}_{t+\delta}]$ for some $\delta > 0$ and trajectory $\widetilde{\mathcal{H}}$. Then, assuming that expectations at times larger than $t$ have been learned correctly, this recursive expression allows us to propagate information for conditioning on earlier histories.

Let us describe this solution in more detail, as applied in the context of $Q$-evaluation.

**Fitted Q evaluation (FQE) in discrete time.** Q-evaluation relies on the tower property of conditional expectations, given below in eq. (1). In discrete time decision processes, where we consider $\delta = 1$, the property suggests a dynamic programming solution that we lay out in algorithm 1 (Le et al., 2019; Watkins and Dayan, 1992). Here, $\mathcal{H}_t$ includes all treatments and observations up to and including time $t$, and since they occur simultaneously, there are exactly $t$ of each. $\widetilde{\mathcal{H}}_{t+1}$ is defined in the same manner, except that it includes $\tilde{a}_{t+1}$ sampled from the target policy $\pi$.

---

**Algorithm 1** Fitted Q-Evaluation (discrete time)

1: **Input**: Trajectories $\{\mathcal{H}_i\}_{i=1}^m$,
2:      Policy $\pi : \cup_{t=1}^T \mathcal{X}^t \times \mathcal{A}^{t-1} \to \Delta^{|\mathcal{A}|}$
3:      Model class $\mathcal{F}$ where $f : \cup_{t=1}^T \mathcal{X}^t \times \mathcal{A}^t \to \mathbb{R}$
4: Initialize $Q$ randomly
5: Set $T_i \leftarrow \max\{t : (t, \mathbf{z}) \in \mathcal{H}_i\} \quad \forall i \in [m]$
6: **for** $N$ rounds **do**
7:      Draw $\tilde{a}_{t+1} \sim \pi(\cdot|\mathcal{H}_{i,t}, \mathbf{x}_{i,t+1}) \quad \forall i, t \in [T_i]$
8:      $\widetilde{\mathcal{H}}_{i,t+1} \leftarrow \mathcal{H}_{i,t} \cup \{(t+1, \mathbf{x}_{t+1}), (t+1, \tilde{a}_{t+1})\}$
9:      Set $\hat{y}_{i,t} = y_{i,t} + Q(\widetilde{\mathcal{H}}_{i,t+1}) \quad \forall i, t \in [T_i]$
10:      $Q \leftarrow \arg\min_{f \in \mathcal{F}} \sum_{i \in [m], t \in [T_i]} (f(\mathcal{H}_{i,t}) - \hat{y}_{i,t})^2$
11: **end for**
12: **Return** $Q$

---

$$\mathbb{E}_P\left[Y|\mathcal{H}_t\right] = \mathbb{E}_{\widetilde{\mathcal{H}}_{t+\delta} \sim P(\cdot|\mathcal{H}_t)}\left[\mathbb{E}_P\left[Y|\widetilde{\mathcal{H}}_{t+\delta}\right]\right], \tag{1}$$

An attractive property of this algorithm is that it is *model-free*. That is, to form the label $\hat{y}_i$ we only need to sample $\tilde{a}_{t+1}$ from our target policy $\pi$, while a model of $P(\mathbf{x}_{t+1}|\mathcal{H}_t)$ is not necessary.

With accurate optimization over a sufficiently expressive hypothesis class and arbitrarily large datasets, algorithm 1 returns correct estimates. This is because if we fix $Q_{t+1}(\mathcal{H}_{t+1})$ and assume that it accurately estimates $\mathbb{E}_P[\sum_{s \geq t+1} Y_s|\mathcal{H}_{t+1}]$, then the minimizer of the regression is the conditional expectation, equal to $\mathbb{E}_P[\sum_{s \geq t} Y_s|\mathcal{H}_t]$ according to eq. (1). The model-free solution is enabled by the equality $P(\mathbf{x}_{t+1}|\mathcal{H}_t) = P_{\text{obs}}(\mathbf{x}_{t+1}|\mathcal{H}_t)$, which validates the use of $\mathbf{x}_{i,t+1}$ to form $\hat{y}_i$. In practice, we take gradient steps on randomly drawn times and training samples instead of walking backward from $T$ to 1. Crucially, for $\delta > 1$, e.g. $\delta = 2$, we have $P(\mathbf{x}_{t+2}|\mathcal{H}_t) \neq P_{\text{obs}}(\mathbf{x}_{t+2}|\mathcal{H}_t)$. Hence, an algorithm using eq. (1) must either be model-based, or resort to solutions such as importance weights that suffer high variance (Hallak et al., 2016; Precup et al., 2000), or restrict the problem, e.g., by discounting rewards (Harutyunyan et al., 2016; Munos et al., 2016; Precup et al., 2000).

**Challenges in application to continuous time.** Moving to continuous time, the tower property turns into a differential equation, and solving it requires tools that go beyond common FQE (e.g., Jia and Zhou (2023)). Regressing to an outcome that is arbitrarily close to the observation at time $t$ is ill-defined. While we may work under a fine discretization of time, this approach is wasteful, as a single update in the minimization for estimating $Q_t$ takes into account the development of the process in the interval $[t, t+\delta]$, and for small values of $\delta$ this will usually yield a very small change to the estimate. Hence, intuitively, when updating $Q_t$, we would like to use estimates of $Q_{t+\delta}$ for a large $\delta$. As explained above, this is seemingly difficult to achieve in a model-free fashion. However, for point processes, since the number of decisions over $[0, T]$ is countable, it seems plausible that a simple and efficient dynamic programming solution can be devised. In what follows, this is what we present.

## 3.1 EDQ: FITTED Q-EVALUATION FOR DECISION POINT PROCESSES VIA EARLIEST DISAGREEMENT TIMES

We wish to reason about what the outcome would have been for an observed trajectory sampled under policy $\lambda_{\text{obs}}^a$, had we treated it with policy $\lambda^a$ from time $t$ onward. Intuitively, it seems plausible that we can use a similar approach to FQE, but instead of going one time unit forward, we can move to the first time $t + \delta$ where the two policies differ in their treatments. That is, we will sample alternative treatment trajectories, asking what the target policy $\lambda^a$ would have done at each time, given the observed history up until that time and find the earliest disagreement. The resulting algorithm is rather simple, and we summarize it graphically in fig. 1. The attractive property of this approach is that the "lookahead" time $\delta$ is adaptive. It will likely be short when applied in parts of trajectories where many treatments are applied, and longer when treatments are scarce.

To formalize the method, we present some additional notation, prove an appropriate variation of the tower property, and explain how it is operationalized by the implementation in algorithm 2.

**Definition 4.** *For process $P_{\text{obs}}$ and policy $(\lambda^a, \pi)$, define an augmented process $\tilde{N} = (N^{a_{\text{obs}}}, N^y, N^x, N^a)$ where intensities are independent of the history of $N^{a_{\text{obs}}}$: (1) the intensity of $N^a$ is $\lambda^a(t|\mathcal{H}_t^{\backslash a}) := \lambda^a(t|\mathcal{H}_t^{x,y}, \mathcal{H}_t^a = \mathcal{H}_t^{a_{\text{obs}}})$, i.e. $\lambda^a$ where history of treatments is given by $\mathcal{H}_t^{a_{\text{obs}}}$, [2] and (2) $\lambda^e(t|\mathcal{H}_t) = \lambda_{\text{obs}}^e(t|\mathcal{H}_t^{\backslash a})$ for $e \in \{a_{\text{obs}}, x, y\}$. For a trajectory $\widetilde{\mathcal{H}} \sim \tilde{P}$ and time $t \in [0, T)$, define $\delta_{\widetilde{\mathcal{H}}}(t) = \min\{u - t : u > t, (u, \cdot) \in \widetilde{\mathcal{H}}^{a,a_{\text{obs}}}\}$, where $\delta_{\widetilde{\mathcal{H}}}(t) = T - t$ when the set is empty.*

The augmented process maintains an additional treatment trajectory, $\widetilde{\mathcal{H}}^a$, as an alternative to $\widetilde{\mathcal{H}}^{a_{\text{obs}}}$, the one observed under $\lambda_{\text{obs}}$. From definition 4, we observe that the intensities of the augmented process $\tilde{N}$ do not depend on $N^a$'s history. It follows that the marginal over $\{x, a, y\}$ is $P_{\text{obs}}$, while $\widetilde{\mathcal{H}}^a$ has a similar role to the alternative treatment $\tilde{a}_{t+1}$ in algorithm 1. This notation is helpful for denoting sampled alternative trajectories over time intervals. Finally, $\delta_{\widetilde{\mathcal{H}}}(t)$ denotes the earliest disagreement between observed and target treatments after time $t$.[3] Our model-free evaluation method is based on the following result, which expresses our estimand as an expectation over trajectories $\widetilde{\mathcal{H}}$.

**Theorem 1.** *Let $P_{\text{obs}}$ be a marked decision point processes, $P$ the process obtained by replacing the policy with $(\lambda^a, \pi)$, and $\tilde{P}$ the augmented process obtained from $P, P_{\text{obs}}$ in definition 4. Further, let $t \in [0, T)$, and $\mathcal{H}_t$ measurable w.r.t $P$. Under Assumption 2, we have that*

$$\mathbb{E}_P[Y|\mathcal{H}_t] = \mathbb{E}_{\widetilde{\mathcal{H}} \sim \widetilde{P}(\cdot|\mathcal{H}_t)}\left[\mathbb{E}_P\left[Y \mid \mathcal{H}_{t+\delta_{\widetilde{\mathcal{H}}}(t)} = \mathcal{H}_t \cup \widetilde{\mathcal{H}}_{(t,t+\delta_{\widetilde{\mathcal{H}}}(t)]}^{\backslash a_{\text{obs}}}\right]\right]. \tag{2}$$

**Takeaways from Theorem 1 and derivation of EDQ.** Equation (2) suggests a method to calculate expectations $\mathbb{E}_P[Y|\mathcal{H}_t]$, similar to how FQE follows from eq. (1). The practical version of the resulting method, EDQ, is given in algorithm 2. To arrive at the method from eq. (2), let us examine the regressions solved in algorithm 2. Taking a random variable $Y_{>s}$ which is the sum of all outcomes after some time $s$, eq. (2) can be rewritten as follows (see Appendix A for a detailed derivation).

$$\underbrace{\mathbb{E}_P[Y|\mathcal{H}_t] - \sum_{(t_k, y_k) \in \mathcal{H}_t^y} y_k}_{Q(\mathcal{H}_t)} = \mathbb{E}_{\widetilde{\mathcal{H}}|\mathcal{H}_t}\left[\sum_{\substack{(t_k, y_k) \in \widetilde{\mathcal{H}}^y: \\ t_k \in (t, t+\delta_{\widetilde{\mathcal{H}}}(t)]}} y_k + \underbrace{\mathbb{E}_P\left[Y_{>t+\delta_{\widetilde{\mathcal{H}}}(t)} \mid \mathcal{H}_{t+\delta_{\widetilde{\mathcal{H}}}(t)} = \mathcal{H}_t \cup \widetilde{\mathcal{H}}_{(t,t+\delta_{\widetilde{\mathcal{H}}}(t)]}^{\backslash a_{\text{obs}}}\right]}_{Q(\mathcal{H}_{t+\delta_{\widetilde{\mathcal{H}}}(t)})}\right] \tag{3}$$

Algorithm 2 uses gradient descent to fit $Q$ functions that, at optimality, satisfy a self-consistency condition that appears in the equation above, $Q(\mathcal{H}_t) = \mathbb{E}_{\widetilde{\mathcal{H}}}[\sum y_k + Q(\mathcal{H}_{t+\delta_{\widetilde{\mathcal{H}}}(t)})]$. To this end, it solves regression problems where $Q(\mathcal{H}_{i,t})$ is fitted to a label $\hat{y}_i$. This label is defined in line 7 and coincides with the term in the expectation $\mathbb{E}_{\widetilde{\mathcal{H}}|\mathcal{H}_t}[\cdot]$ above. To conclude that the algorithm estimates the effect of interest, two simple arguments suffice: (1) a uniqueness argument showing that the self-consistency equation holds only when $Q(\mathcal{H}_t)$ matches $\mathbb{E}_P[Y_{>t}|\mathcal{H}_t]$, and (2) that despite ignoring the unobserved process $\lambda^u$, the estimation yields the desired causal effect due to the ignorability assumption. We summarize the conclusion below and provide detailed steps in appendix D.

**Corollary 1.** *Under assumption 1, a Q-function satisfying eq. (3) yields the causal effect of the intervention that replaces $(\lambda_{\text{obs}}^a, \pi_{\text{obs}})$ with $(\lambda^a, \pi)$.*

An analogue of eq. (2) holds for discrete-time decision processes, where EDQ bears some resemblance to the eligibility traces approach of Precup et al. (2000). We provide this result in appendix B.3 for completeness but focus here on the point process case, as this is our main motivation and where the earliest disagreement approach is most fruitful.

## 4 RELATED WORK

Our coverage of related work is divided into an overview of works that solve adjacent tasks to ours, before transitioning into a detailed discussion in section 4.1 about techniques more closely aligned with our goal of large scale causal inference in sequential decision making.

**Causal inference with sequential decisions.** Causal effect estimation for sequential treatments is usually studied in discrete time under the sequential exchangeability assumption (Hernan and

---

[2]Note that this is a slight abuse of notation, since $\mathcal{H}_t^a$ is not a random variable.

[3]The minimum treatment time is also the earliest disagreement, since under some regularity conditions, the processes $N^a$ and $N^{a_{\text{obs}}}$ have probability 0 of jumping simultaneously.

---

**Algorithm 2** Earliest Disagreement Fitted Q-Evaluation

---

1: **Input**: Trajectories $\{\mathcal{H}_i\}_{i=1}^m$,
2:        Policy $\lambda_a(\cdot|\mathcal{H}_t), \pi(\cdot|\mathcal{H}_t)$
3: Initialize $\boldsymbol{\theta}$ randomly
4: **for** $N$ rounds **do**
5:    Draw $t \sim \text{Unif}([0, T])$ and $i \sim \text{Unif}([m])$
6:    Draw $\widetilde{\mathcal{H}} \sim \widetilde{P}(\cdot|\mathcal{H}_{i,t})$ and set $\mathcal{H}'_{i,t+\delta_{\widetilde{\mathcal{H}}}(t)} = \mathcal{H}_t \cup \widetilde{\mathcal{H}}^{\backslash a_{\text{obs}}}_{(t,t+\delta_{\widetilde{\mathcal{H}}}(t)]}$
7:    $\hat{y}_i = \sum_{\substack{(t_k,y_k) \in \widetilde{\mathcal{H}}^y: \\ t_k \in (t, t+\delta_{\widetilde{\mathcal{H}}}(t)]}} y_k + Q(\mathcal{H}'_{i,t+\delta_{\widetilde{\mathcal{H}}}(t)}; \boldsymbol{\theta})$
8:    $\boldsymbol{\theta} \leftarrow \boldsymbol{\theta} - \eta \nabla_{\boldsymbol{\theta}} \left( Q(\mathcal{H}_{i,t}; \boldsymbol{\theta}) - \hat{y}_i \right)^2$
9: **end for**
10: **Return** $Q(\cdot; \boldsymbol{\theta})$

---

Robins, 2023; Robins, 1986). Addressing unobserved confounders is of interest (Namkoong et al., 2020; Tennenholtz et al., 2020), but this is beyond our scope here. As described in section 2.2, the framework of Røysland et al. (2022) draws a parallel to sequential exchangeability for continuous time, which we adopt here. For estimation in these continuous-time problems, several methods have been explored (Lin et al., 2004; Lok, 2008; Røysland, 2011; Rytgaard et al., 2022; Zhang et al., 2011). Most do not scale to large and high-dimensional datasets. For instance, Røysland (2011) requires estimating an integral of propensity weights over time; Rytgaard et al. (2022) propose a targeted estimator that fits each process in $P_{\text{obs}}$, but it is limited to interventions at fixed times (i.e. no interventions on treatment schedules) and it is unclear how to implement it with expressive models. Nie et al. (2021) study the effect of timing $t$ when applying a fixed policy from time $t$ onward. However, they do not discuss interventions on all treatment timings, as we do here.

**Reinforcement learning (RL) techniques.** As discussed in section 3, EDQ is related to FQE and Q-learning (Le et al., 2019; Murphy, 2005; Watkins and Dayan, 1992), and to $n$-step methods from RL (De Asis et al., 2018; Munos et al., 2016; Precup et al., 2000). In the causality literature, Q-learning often appears in the context of dynamic treatment regimes (Chakraborty and Moodie, 2013; Chakraborty and Murphy, 2014), where discrete-time policies are learned off-policy. All these methods are not applicable to irregularly sampled times. Works in RL that consider irregularly sampled times, either do not intervene on treatment times, or operate in the on-policy setting (Qu et al., 2023; Upadhyay et al., 2018), or incorporate continuous-time positional embeddings into decision transformers (Chen et al., 2021b). The latter facilitates the recommendation of sets of actions to arrive at a desired outcome (Zhang et al., 2023) rather than evaluating a policy of interest. Recent work suggests that goal-conditioned imitation learning methods such as decision transformers may fail to estimate the causal effect of actions (Malenica and Murphy, 2023) in some scenarios where there are no unobserved confounders, whereas $Q$-learning methods produce correct estimates.

We next discuss *scalable* causal estimation methods for sequential treatments. We delineate assumptions, implementation choices, and subsequent properties of solutions from recent work.

## 4.1 LARGE SCALE ESTIMATION APPROACHES FOR SEQUENTIAL TREATMENTS

Notable early work on machine learning for estimating counterfactual quantities related to treatment timelines, Schulam and Saria (2017), used Gaussian Processes for estimation. Limitations such as scalability and incorporating various features prompted the development of deep learning approaches.

**Large scale models.** One family of solutions (Bica et al., 2020; Lim, 2018; Melnychuk et al., 2022) took an important step forward by using RNNs and transformers for estimation. All of these methods build on the idea of learning balancing representations (Johansson et al., 2016). Roughly, these are representations under which the treatment is randomly assigned. This facilitates the training of high-capacity effect estimators on large datasets, but these works are restricted to discrete times.

**Dynamic policies.** The above methods can only estimate effects of *static* treatments, meaning the treatment plan cannot dynamically depend on future observations. For instance, consider a policy that prescribes a daily dose of statins, and if the patient develops side effects in the future, switches

---

[4]We mark TE-CDE with as non-scalable, since the algorithm relies on differential equation solvers which limits the scalability of methods one can use.

Table 1: Qualitative comparison of estimation methods under sequential treatments, compatible with large scale ML. Our method EDQ is marked in bold, while others can be found in (Bica et al., 2020; Chakraborty and Moodie, 2013; Le et al., 2019; Li et al., 2021; Lim, 2018; Melnychuk et al., 2022; Schulam and Saria, 2017; Seedat et al., 2022). The acronym DP stands for Dynamic Programming.

| | Properties | | | Estimation Method | | | |
| | Irregular Times | Large Scale | Dynamic Policy | Prop. Weights | Model Based | Balancing Rep. | DP |
|---|---|---|---|---|---|---|---|
| CGP | ✗ | ✗ | ✓ | | ● | | |
| CRN , CT | ✗ | ✓ | ✗ | | | ● | |
| R-MSN | ✗ | ✓ | ✗ | ● | | | |
| TE-CDE | ✓ | ✗[4] | ✗ | | ● | ● | |
| G-Net | ✗ | ✓ | ✓ | | ● | | |
| FQE | ✗ | ✓ | ✓ | | | | ● |
| **EDQ** | ✓ | ✓ | ✓ | | | | ● |

to another medication. This policy depends on possible future states, which the above methods do not accommodate. G-Net (Li et al., 2021; Xiong et al., 2024) takes a model-based approach, where models are fit for both $\pi_{\text{obs}}(A(t)|\mathcal{H}_{t-1}, X(t))$, and $P_{\text{obs}}(X(t), Y(t)|\mathcal{H}_{t-1})$. Then at inference time, a dynamic policy is estimated when $\pi_{\text{obs}}$ is replaced with the desired policy $\pi$ and conditional expectations of $Y$ are estimated with Monte-Carlo simulations.

**Irregular times.** None of the above solutions address irregular observation and action times, a key focus of this paper. Some steps in this direction have been taken by Seedat et al. (2022); Vanderschueren et al. (2023). Past work on irregular times includes Seedat et al. (2022), who combine balanced representations with a neural controlled differential equation architecture suited for irregular sampling, while Vanderschueren et al. (2023) use reweighting to adjust for sampling times that are informative of the outcome. However, these methods mitigate sampling-induced bias rather than estimate outcomes under interventions on treatment times. We further discuss them in appendix C.

Table 1 summarizes the above techniques, along with FQE and EDQ. Notably, EDQ possesses the desirable qualities mentioned here and handles interventions on $\lambda^a$, which other solutions do not.

## 5 IMPLEMENTATION AND EXPERIMENTS

To implement EDQ for experimentation we use the GPT-2 architecture. Each token is a concatenation of embeddings of time $t_i$, value $\mathbf{z}_i$ and event type $e_i \in \{A, X, Y, \Delta T\}$. The event types $A, X, Y$ correspond to actions, features, and outcomes, while $\Delta T$ is introduced for convenience to represent cases where time passes but no events occur. Both absolute times $t_i$ and time gaps $\Delta T$ use continuous time positional embeddings: the $k$-th dimension is $\sin\left(tC^{-k/d_{\text{time}}}\right)$ for even $k$, and $\cos\left(tC^{-(k-1)/d_{\text{time}}}\right)$ for odd $k$. Here $C = 10^5$ and $d_{\text{time}}$ is the embedding dimension. We also keep a target network and update it with soft-Q updates, as is common in Deep Q-Networks, e.g., Van Hasselt et al. (2016).

### 5.1 BASELINES

We are unaware of baselines for effect estimation on treatment timing with high-dimensional or long-sequence data. Thus, we implement two baselines that let us glean important aspects of EDQ.

**ERM / MC** is an Empirical Risk Minimizer (ERM) trained to predict observed outcomes, which, also known as Monte-Carlo (MC) prediction in RL and policy evaluation. We use the same GPT-2 architecture and data representation as EDQ, but instead of running algorithm 2, we train $f_{\boldsymbol{\theta}}(\mathcal{H}_t)$ by minimizing prediction loss on observed data. Given each training trajectory $\mathcal{H}_i$ with an outcome label $y_i$, we solve $\min_{\boldsymbol{\theta}} \sum_{i,(t_i, \mathbf{z}_i) \in \mathcal{H}_i} (\ell(f_{\boldsymbol{\theta}}(\mathcal{H}_{i,t}), y_i)$ , where $\ell(\cdot, \cdot)$ is the squared loss. This method estimates outcomes under $\lambda_{\text{obs}}$, therefore we expect it to perform as well as or better than off-policy evaluation methods, including EDQ, when $\lambda = \lambda_{\text{obs}}$, and to suffer a drop otherwise.

**FQE** is implemented following section 3, but with discretized time and Q-updates using one timestep forward. At each iteration, we draw training example $i \in [m]$ and time $t \in [0, T]$, define $\hat{y}_i = y_{t+1} + Q_{t+1}(\widetilde{\mathcal{H}}_{t+1}; \boldsymbol{\theta}))$, and take a gradient step on loss $\ell(Q_t(\mathcal{H}_{i,t}), \hat{y}_i)$. Similarly, we define discrete-time approximations of our policies of interest, described later. The positional embeddings

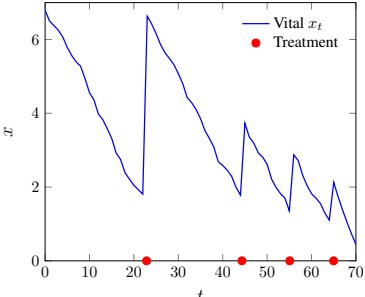

| | ERM / MC | FQE | EDQ |
|---|---|---|---|
| | | $\lambda_{\text{int}} = 0.1$ | |
| $\lambda_{\text{obs}} = 0.1$ | $0.20 \pm 0.007$ | $0.21 \pm 0.02$ | $0.20 \pm 0.005$ |
| $\lambda_{\text{obs}} = 0.5$ | $0.38 \pm 0.011$ | $0.23 \pm 0.04$ | $\mathbf{0.20 \pm 0.006}$ |
| | | $\lambda_{\text{int}} = 0.5$ | |
| $\lambda_{\text{obs}} = 0.5$ | $0.11 \pm 0.004$ | $0.197 \pm 0.013$ | $0.10 \pm 0.003$ |
| $\lambda_{\text{obs}} = 0.1$ | $0.28 \pm 0.004$ | $0.31 \pm 0.01$ | $\mathbf{0.11 \pm 0.006}$ |

Figure 3: **Left.** An example of a trajectory from our simulation. The blue curve denotes the value of the vital $x_t$ and red dots mark treatment times. **Right.** Normalized RMSE under the different simulation settings. The mean is taken over all points in the history of patients in the test data. Rows colored blue have $\lambda_{\text{obs}} = \lambda_{\text{int}}$, and we expect all methods to perform well since train and test distributions match. Red rows are those where the effect of an intervention needs to be estimated.

correspond to discrete times, and the representations of actions, features and outcomes at each time are concatenated. Other than this, we use the same architecture and hyperparameters of EDQ. This baseline examines the effects of time discretization on estimation and optimization.

*Computational complexity:* The per-iteration runtime of EDQ is similar to FQE, which is a common tool for large-scale offline RL problems, e.g. Paine et al. (2020); Voloshin et al. (2021). EDQ and FQE differ in computation times due to sampling methods from the target policy used to draw the treatments used in the $Q$-update. We discuss this in appendix A.

## 5.2 SIMULATIONS ON TIME TO FAILURE AND CANCER TUMOR GROWTH PREDICTION

To validate our method, we construct two settings. The first is to predict the effect of treatment timing policies on patients' time-to-event. The second uses a cancer tumor growth simulator from Geng et al. (2017) to form a policy evaluation problem on applications of chemotherapy and radiotherapy.

**Simulators.** We use two simulators. **(i) Time-to-failure:** In this setting, each data point simulates the vital of a patient $x_t \in \mathbb{R}_+$ measured regularly at a frequency of one time unit, and treatments $a_t \in \mathbb{R}_+$ are assigned irregularly in time according to an observational policy. Without treatment, the vital drops linearly $dx_t/dt = -(\alpha + \xi_t)$ where $\alpha > 0$ and $\xi_t \sim \mathcal{N}(0, \sigma)$ is a noise term drawn at each time unit. Upon receiving treatment, the vital rises by an amount inversely proportional to the number of treatments, $1 \leq k \leq m$, applied up until that time, where $m$ is the maximal number of treatments that a patient can receive. That is, the efficacy of treatment reduces with repeated applications. We also inject small noise terms into the treatment dosage that a patient receives, which further affect the vital and add randomness to the problem. Section 5.2 shows an example simulated patient trajectory. **(ii) Tumor growth:** We use the experimental setting from Bica et al. (2020), which other works use to study irregular sampling (Seedat et al., 2022; Vanderschueren et al., 2023). As this is a commonly used simulator, we defer its details to appendix A and focus on the type of irregular sampling and policies we use. The simulator works in discrete time $t \in [T]$, and irregular sampling is induced by the features being unobserved at certain times. Namely, the covariate $x_t \in \mathbb{R}_+$ which represents tumor volume is observed with probability $\sigma((\bar{x}_{t-d:t-1}/d_{\max}) - 1.5)$, where $\bar{x}_{t-d:t-1}$ is the average tumor volume over the last $d$ timesteps, and $d_{\max}$ is the maximum considered volume.

**Outcomes and policies.** For **(i) time-to-failure**, our outcome of interest is failure time $y \in \mathbb{R}_+$, where a patient dies if the vital drops to a value of 0. [5] We focus on effect estimation for interventions on a rate parameter $\lambda^a$ that controls the timing of treatment. At each time $t$ where the observed vital crosses a threshold, i.e. $x_t < r$ for some predetermined $r \in \mathbb{R}_+$, a random time is drawn from an exponential distribution $\delta \sim \exp(\lambda^a)$ and treatment is applied at $t + \delta$. The threshold $r$ and the dosage of treatment given are also part of the policy $\pi$, yet to focus on the effects of timing we do not intervene on them in this experiment. At each experiment we observe $m$ patients treated under a policy with $\lambda^a = \lambda_{\text{obs}}$ and aim to reason about the expected failure times under the interventional $\lambda_{\text{int}}$. In **(ii) tumor-growth**, the goal is to predict $x_T$, where $T = 20$, under a policy that at time $t$ assigns treatment $a_t$ from four possible options: no-treatment, radiotherapy, chemotherapy, or combined

---

[5]Note that the vital changes outside measurement times, hence death time does not generally coincide with vital measurement times.

| | ERM / MC | FQE | EDQ |
|---|---|---|---|
| | $(\gamma,\beta)_{\text{int}} = (6, 0.75)$ | | |
| $(\gamma,\beta)_{\text{obs}} = (6, 0.75)$ | **0.034 ± 0.001** | 0.048 ± 0.001 | 0.037 ± 0.001 |
| $(\gamma,\beta)_{\text{obs}} = (10, 0.5)$ | 0.07 ± 0.004 | 0.080 ± 0.013 | **0.052 ± 0.006** |

| | ERM / MC | FQE | EDQ |
|---|---|---|---|
| | | $\lambda_{\text{int}} = 0.2$ | |
| $\lambda_{\text{obs}} = 0.2$ | **0.17 ± 0.01** | 0.18 ± 0.004 | 0.178 ± 0.01 |
| $\lambda_{\text{obs}} = 2$ | 0.28 ± 0.01 | 0.20 ± 0.03 | **0.178 ± 0.01** |
| | | $\lambda_{\text{int}} = 2$ | |
| $\lambda_{\text{obs}} = 2$ | 0.22 ± 0.01 | **0.197 ± 0.013** | 0.22 ± 0.004 |
| $\lambda_{\text{obs}} = 0.2$ | 0.32 ± 0.02 | 0.31 ± 0.01 | **0.22 ± 0.007** |

Figure 4: **Left.** Normalized RMSE on the **tumor-growth** simulation. All methods are affected by distribution shift. EDQ is the most robust out of the methods considered. **Right.** Normalized RMSE for **time-to-failure** simulation on short trajectories.

therapy. Therefore, the estimation here is both on "when" and "what" to do. Policies are determined by two parameters $(\gamma, \beta)$ and assign each type of treatment with probability $\sigma(\gamma(x_{\text{last}} - \beta) + t - t_{\text{last}})$. Here, $x_{\text{last}}$ is the last observed volume and $\beta$ is an intercept controlling how often treatments are applied, while $\gamma$ controls the dependence of treatment assignment on tumor volume. Finally $t_{\text{last}}$ is the time of the last treatment, and the term $t - t_{\text{last}}$ induces a lag between consecutive treatments.

**Experiments.** We perform two sets of experiments for **time-to-failure**. In the first set, trajectory lengths range in $10 - 100$, and there are 5 treaments. In the second set (results in Figure 4, right), we change the parameters of the problem by taking a high slope $\alpha$ and capping at one treatment. This creates short trajectories of length between 3 and 10. To evaluate the performance of the estimator, we sample trajectories $(\mathcal{H}_i, y_i) \sim P_{\lambda_{\text{int}}}$ under the target policy and treat every $(\mathcal{H}_{i,t_i}, y_i)$ as a labeled data point. We then evaluate normalized RMSE between $f_{\boldsymbol{\theta}}(\mathcal{H}_{i,t_i})$ and the true labels $y_i$. For **tumor growth**, we evaluate a policy that increases the likelihood of treatments (i.e increases $\beta$) and reduces $\gamma$, the correlation to the observed volume. Error is also calculated with normalized RMSE.

**Results.** The tables in fig. 3 and fig. 4 present results. They show that for **time-to-failure**, EDQ solves the estimation problem both when $\lambda_{\text{obs}} = \lambda_{\text{int}}$ (blue rows, no intervention performed), and when $\lambda_{\text{obs}} \neq \lambda_{\text{int}}$ (red rows). This is evident by comparing to ERM under the setting where $\lambda_{\text{obs}} = \lambda_{\text{int}}$, where ERM should be nearly optimal. [6] ERM takes a significant performance drop when $\lambda_{\text{obs}} \neq \lambda_{\text{int}}$, as expected. For FQE, while in the first set of experiments (fig. 3), discretization should not result in significant information loss, it does create a more difficult optimization problem. This is because the updates to $Q(\mathcal{H}_t)$ need to propagate backwards and most updates get noisy gradient signals by fitting to the $Q$ value of a trajectory sampled one time step ahead, $Q(\widetilde{\mathcal{H}}_{t+1})$. This challenge for FQE is most evident in Figure 3, where $\lambda_{\text{int}} = 0.5$ and FQE incurs a significant loss both when $\lambda_{\text{obs}} = \lambda_{\text{int}}$ and when $\lambda_{\text{obs}} \neq \lambda_{\text{int}}$. The results in Figure 4 (right) demonstrate the potential effects of information loss due to discretization. Here, since trajectories are short, the optimization problem of losses propagating is likely less pronounced. However, we see that there is still a significant drop when $\lambda_{\text{int}} = 2$ and $\lambda_{\text{obs}} = 0.2$ (approximating a high rate of treatment when data was sampled under low rates). Taken together, these two experiments demonstrate two possible drawbacks of discretization. For **tumor-growth**, EDQ still outperforms the alternatives but suffers a decrease in performance due to distribution shift between observational and interventional distributions.

# 6    LIMITATIONS AND FUTURE WORK

We have developed a method for off-policy evaluation with irregular treatment and observation times, which facilitates interventions on treatment intensities. We connected the setting to identifiability results from causal inference to highlight the conditions under which the estimates are meaningful, and proved the estimator's correctness. EDQ is a "direct" method based on regression and, as demonstrated in experiments, is readily applicable to high-capacity sequence modeling architectures. To the best of our knowledge, it is the first available solution to this estimation problem that is applied with such architectures. Several limitations motivate exciting future research. Empirically, we plan to apply the method to large real-world datasets and additional simulators (Namkoong et al., 2020; Oberst and Sontag, 2019). The method does not handle censoring, which is required for reliable application in survival analysis and real trial data. Further developements include policy optimization in the setting we studied here and deriving bounds on errors due to unobserved confounding.

---

[6]this is up to numerical optimization issues, as we see FQE can outperform it in certain cases

ACKNOWLEDGMENTS

We thank Stefan Groha for helpful discussion in early stages of the project. This work was supported by National Science Foundation Award 1922658, NIH/NHLBI Award R01HL148248, NSF CAREER Award 2145542, NSF Award 2404476, ONR N00014-23-1-2634, IITP with a grant from ROK-MSIT in connection with the Global AI Frontier Lab International Collaborative Research, Apple, and Optum.

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
