## A ADDITIONAL COMMENTS ON EXPERIMENTAL ASPECTS

Here we slightly expand on the comment about computational complexity in the main text, and give more details about the cancer simulation we use from Bica et al. (2020); Geng et al. (2017); Seedat et al. (2022); Vanderschueren et al. (2023).

*A comment on computational complexity:* As commented in the main text, the per-iteration runtime of EDQ is similar to that of FQE, which is a common tool in large-scale offline RL problems; for example, Paine et al. (2020); Voloshin et al. (2021) use it in benchmarks and evaluations. The difference in computation times between EDQ and FQE is due to sampling from the target policy, or more accurately $\tilde{P}_t^a$, in order to draw the treatments used in the $Q$-update, i.e., $\delta$ and $\widetilde{\mathcal{H}}_{t+\delta}$ in algorithm 2. In most applications, the added complexity due to this difference is small relative to the cost of evaluating the $Q$-function and its gradients. In turn, the cost of function evaluation is the same for FQE and EDQ. The computational complexity of sampling from $\tilde{P}_t^a$ depends on how it is represented and implemented. For instance, we may specify policies by allowing evaluations of $\lambda^a(u|\mathcal{H}_u)$, and sample using the thinning algorithm (Lewis and Shedler, 1979; Ogata, 1981); with neural networks that allow sampling the time-to-next-event (e.g., see (McDermott et al., 2023; Nagpal et al., 2021) for examples of event time prediction); or with closed-form decision rules. For example, in the time-to-failure simulation, to determine treatment times we sample exponential variables every times the vital feature crosses a certain threshold.

*Cancer simulator:* The tumor growth simulation we use is adapted from Bica et al. (2020); Melnychuk et al. (2022); Seedat et al. (2022) and is based on the work of Geng et al. (2017). Tumor volumes $V(t)$ are simulated as finite differences from the following differential equation,

$$\frac{dV(t)}{dt} = \left( \underbrace{\rho \log\left(\frac{K}{V(t)}\right)}_{\text{Tumor growth}} - \underbrace{\beta_c C(t)}_{\text{Chemotherapy}} - \underbrace{(\alpha_r d(t) + \beta_r d(t)^2)}_{\text{Radiotherapy}} + \underbrace{e_t}_{\text{Noise}} \right) V(t).$$

Here $C(t)$ is the chemotherapy concentration, $d(t)$ represents the level of radiothearpy. $\rho, K, \beta_c, \alpha_r, \beta_r$ are effect parameters drawn for each patient from a prior distribution described in Geng et al. (2017), and $e_t \sim \mathcal{N}(0, 0.0001)$ is a noise term. To create irregularly sampled observations of the tumor volume, at each time step we draw a value from a Bernoulli distribution to decide whether the trajectory contains the tumor volume at this time step or not. The success probability is a function of the average tumor volume over the most recent 15 volumes (both observed and unobserved). If we denote a missing value by $\emptyset$ and the observation at timestep $t$ by $X_t$ (which equals $\emptyset$ if there is no sample at this timestep and $V(t)$ otherwise), then sampling times are drawn according to the following probabilities:

$$p(V_t \neq \emptyset | \mathcal{H}_t) = \sigma\left( \frac{\bar{V}_{t-15:t}}{V_{\max}} - 1.5 \right)$$

The policies we use to decide on treatments draw binary decisions of whether or not to apply chemotherapy and radiotherapy at each timestep. Denoting these decisions by random variables $C_t$ and $d_t$, they are drawn according to $P(R_t = 1 | \mathcal{H}_t) = \sigma\left(\gamma \cdot (v_{\text{last}} - \beta) + t - t_{\text{last}}\right)$, where $v_{\text{last}}$ is the last observed volume before time $t$ and $t_{\text{last}}$ is the last time that treatment was applied before $t$. The same probabilities are applied for $d_t$. For more details on the specifics of the simulation, see the code implementation.

## B PROOFS

We begin with some notation and additional definitions, in appendix B.2 we prove the consistency result for our method, and in appendix B.3 we give its discrete-time version. To avoid cluttered notation and longer proof, we will give the proof of theorem 1 for unmarked processes. Adding a distribution of marks is a trivial extension that does not alter the main steps of the derivation.

### B.1 NOTATION AND DEFINITIONS

For a multivariate point process we use the following notations:

- $\lambda_\bullet(\cdot)$ is the sum $\sum_e \lambda^e(\cdot)$, in our case this will include the components $e \in \{a, x, y\}$.

- For any $s > t$ and any distribution or intensity, e.g. $\lambda$, we will use the conditioning $\lambda(\cdot|\mathcal{H}_s = \mathcal{H}_t)$ to denote the event where jumps until time $t + \delta$ are those that appear in $\mathcal{H}_t$. That is, no events occur in the interval $(t, s]$.

- $\mathcal{H}_t \cup \{(t + \delta, e)\}$ is the event in which jumps until time $t + \delta$ are those that appear in $\mathcal{H}_t$, and the next jump after that happens at time $t + \delta$ and is of type $e$ (i.e. $N^e(t + \delta) = N^e(t) + 1$ and $N^e(t + s) = N^e(t)$ for $s \in (0, \delta)$).

- Given a trajectory $\widetilde{\mathcal{H}}$ and time $t$, we define $\delta^e(t) = \min\{s - t : s > t, (s, \cdot) \in \widetilde{\mathcal{H}}^e\}$ as the time gap from time $t$ to the first jump of process $N_e$ in trajectory $\mathcal{H}$ after $t$, that is for $e \in \{x, y, a, a_{\text{obs}}\}$.
  *Note:* This is a slight abuse of notation from the main paper, where we defined $\delta_{\widetilde{\mathcal{H}}}(t) = \min\{u - t : u > t, (u, \cdot) \in \widetilde{\mathcal{H}}^{a, a_{\text{obs}}}\}$: dependence on $\widetilde{\mathcal{H}}$ is omitted and will be included whenever the trajectory is not clear from context, and we add a superscript (e.g. $\delta^{a, a_{\text{obs}}}$) to specify the type of event we look for.

- Since the processes $N^x, N^y$ play a similar role throughout the derivation, as the parts of the process whose intensities are invariant under the intervention, we will shorten notation to $\lambda^v(t + \delta|\mathcal{H}_{t+\delta})\mathbb{E}_P[Y|\mathcal{H}_t \cup \{(t + \delta, v)\}] := \lambda^x(t + \delta|\mathcal{H}_{t+\delta})\mathbb{E}_P[Y|\mathcal{H}_t \cup \{(t + \delta, x)\}] + \lambda^y(t + \delta|\mathcal{H}_{t+\delta})\mathbb{E}_P[Y|\mathcal{H}_t \cup \{(t + \delta, y)\}]$ and $\delta^v := \delta_x \wedge \delta_y$.

We assume that all processes have well-defined densities and intensity functions, and that $P$ is absolutely continuous w.r.t. $P_{\text{obs}}$, $P \ll P_{\text{obs}}$. This means that the conditional expectations taken w.r.t. $P$, which we use in our derivation, are well defined. We also adopt the convention where $N^e(t)$ is almost surely finite for any $t \in [0, T]$ and $e \in \{a, x, y\}$ (Andersen et al., 2012). This means that the number of events in the interval $[0, T]$ is countable. We also use the notation $\mathbf{1}[\cdot]$ for the indicator function that returns 1 if the condition inside it is satisfied and 0 otherwise.

## B.2 Proof of Formal Results

Below we prove theorem 1 where the result, eq. (2), implies that performing dynamic programming using the $Q$-function from the earliest disagreement time between observed data, and the data sampled from the target distribution, results in a correct estimator. We derive that equation from the lemma below, which is similar to a tower property of conditional expectations with respect to the first jump that occurs in any component of the process.

**Lemma 1.** *Let $P, P_{\text{obs}}$ be multivariate marked decision point processes, $\tilde{P}$ the corresponding augmented process, $t \in [0, T)$, and $\mathcal{H}_t$ a history of events that is measurable w.r.t $P$. It holds that*

$$\mathbb{E}_P[Y|\mathcal{H}_t] = \mathbb{E}_{\widetilde{\mathcal{H}} \sim \widetilde{P}(\cdot|\mathcal{H}_t)}\Big[\mathbf{1}\big[\delta^a(t) < \delta^v(t) \wedge \delta^{a_{\text{obs}}}(t)\big]\mathbb{E}_P[Y|\mathcal{H}_t \cup (t + \delta^a(t), a)] +$$
$$\mathbf{1}\big[\delta^v(t) < \delta^a(t) \wedge \delta^{a_{\text{obs}}}(t)\big]\mathbb{E}_P[Y|\mathcal{H}_t \cup (t + \delta^v(t), v)] +$$
$$\mathbf{1}\big[\delta^{a_{\text{obs}}}(t) < \delta^a(t) \wedge \delta^v(t)\big]\mathbb{E}_P[Y|\mathcal{H}_{t+\delta^{a_{\text{obs}}}(t)} = \mathcal{H}_t] +$$
$$\mathbf{1}\big[\delta^{a_{\text{obs}}}(t) \wedge \delta^a(t) \wedge \delta^v(t) > T - t\big]\mathbb{E}_P[Y|\mathcal{H}_T = \mathcal{H}_t]\Big] \quad (4)$$

*Proof.* Note that all the conditional expectations in the above expression exist since $P \ll P_{\text{obs}}$. Denoting the next jump time with a variable $T_{\text{next}}$ and its type by $E_{\text{next}}$, when conditioning on some history $\mathcal{H}_s$, the law of total probability suggests that $P(Y|\mathcal{H}_t) = \int \sum_{e \in \{a, x, y\}} P(Y|\mathcal{H}_t, T_{\text{next}} = t + \delta, E_{\text{next}=e})P(T_{\text{next}} = t + \delta, E_{\text{next}=e}|\mathcal{H}_t)dt$. In point processes, likelihoods of the form $P(T_{\text{next}} = t + \delta, E_{\text{next}} = e|\mathcal{H}_t)$ are given by $\exp\{-\int_t^{t+\delta} \lambda_\bullet(s|\mathcal{H}_s = \mathcal{H}_t)\} \cdot \lambda^e(t + \delta|\mathcal{H}_{t+\delta} = \mathcal{H}_t)$. Expanding $\mathbb{E}_P[Y|\mathcal{H}_t]$ with the law of total probability and these likelihoods, while accounting for the option that no jump occurs in $(t, T]$, we obtain the following expression.

$$\mathbb{E}_P[Y|\mathcal{H}_t] = \exp\left\{-\int_t^T \lambda_\bullet(s|\mathcal{H}_s = \mathcal{H}_t)\right\}\mathbb{E}_P[Y|\mathcal{H}_T = \mathcal{H}_t] +$$
$$\int_0^{T-t} \exp\left\{-\int_t^{t+\delta} \lambda_\bullet(s|\mathcal{H}_s = \mathcal{H}_t)ds\right\}$$

$$\Big(\lambda^a(t+\delta|\mathcal{H}_{t+\delta}=\mathcal{H}_t)\mathbb{E}_P\left[Y|\mathcal{H}_t\cup\{(t+\delta,a)\}\right]+$$
$$\lambda^x(t+\delta|\mathcal{H}_{t+\delta}=\mathcal{H}_t)\mathbb{E}_P\left[Y|\mathcal{H}_t\cup\{(t+\delta,x)\}\right]+$$
$$\lambda^y(t+\delta|\mathcal{H}_{t+\delta}=\mathcal{H}_t)\mathbb{E}_P\left[Y|\mathcal{H}_t\cup\{(t+\delta,y)\}\right]\Big)\mathrm{d}\delta. \tag{5}$$

Next we write down each item in eq. (4),

$$\mathbb{E}_{\widetilde{\mathcal{H}}\sim\widetilde{P}(\cdot|\mathcal{H}_t)}\left[\mathbf{1}\left[\delta^a(t)<\delta^v(t)\wedge\delta^{a_{\mathrm{obs}}}(t)\right]\cdot\mathbb{E}\left[Y|\mathcal{H}_t\cup(t+\delta^a(t),a)\right]\right]=$$
$$\int_0^{T-t}\lambda^a(t+\delta|\mathcal{H}_{t+\delta}=\mathcal{H}_t)\exp\{-\int_t^{t+\delta}\lambda^a(s|\mathcal{H}_s=\mathcal{H}_t)\mathrm{d}s\}$$
$$\exp\{-\int_t^{t+\delta}\lambda_{\mathrm{obs},\bullet}(s|\mathcal{H}_s=\mathcal{H}_t)\mathrm{d}s\}\mathbb{E}\left[Y|\mathcal{H}_t\cup(t+\delta,a)\right]\mathrm{d}\delta=$$
$$\int_0^{T-t}\lambda^a(t+\delta|\mathcal{H}_{t+\delta}=\mathcal{H}_t)\exp\{-\int_t^{t+\delta}\lambda_\bullet(s|\mathcal{H}_s=\mathcal{H}_t)\mathrm{d}s\}$$
$$\exp\{-\int_t^{t+\delta}\lambda^a_{\mathrm{obs}}(s|\mathcal{H}_s=\mathcal{H}_t)\mathrm{d}s\}\mathbb{E}\left[Y|\mathcal{H}_t\cup(t+\delta,a)\right]\mathrm{d}\delta=$$
$$\int_0^{T-t}\lambda^a(t+\delta|\mathcal{H}_{t+\delta}=\mathcal{H}_t)\exp\{-\int_t^{t+\delta}\lambda_\bullet(s|\mathcal{H}_s=\mathcal{H}_t)\mathrm{d}s\}\mathbb{E}\left[Y|\mathcal{H}_t\cup(t+\delta,a)\right]$$
$$\cdot(1-1+\exp\{-\int_t^{t+\delta}\lambda^a_{\mathrm{obs}}(s|\mathcal{H}_s=\mathcal{H}_t)\mathrm{d}s\})\mathrm{d}\delta. \tag{6}$$

The first equality simply expands the expectation as an integration over all possible stopping times for $N_a$ (according to the definition of $\widetilde{P}$, see definition 4). The second equality holds since the intensities $\lambda^x_{\mathrm{obs}},\lambda^y_{\mathrm{obs}}$ are equal to $\lambda_x,\lambda_y$ respectively. Then finally we simply add and subtract 1 from the last item. Similarly, for the second item in eq. (4)

$$\mathbb{E}_{\widetilde{\mathcal{H}}\sim\widetilde{P}(\cdot|\mathcal{H}_t)}\left[\mathbf{1}\left[\delta^v(t)<\delta^a(t)\wedge\delta^{a_{\mathrm{obs}}}(t)\right]\cdot\mathbb{E}\left[Y|\mathcal{H}_t\cup(t+\delta^v(t),v)\right]\right]=$$
$$\int_0^{T-t}\lambda^v(t+\delta|\mathcal{H}_{t+\delta}=\mathcal{H}_t)\exp\{-\int_t^{t+\delta}\lambda_\bullet(s|\mathcal{H}_s=\mathcal{H}_t)\mathrm{d}s\}\mathbb{E}\left[Y|\mathcal{H}_t\cup(t+\delta,v)\right]$$
$$\cdot(1-1+\exp\{-\int_t^{t+\delta}\lambda^a_{\mathrm{obs}}(s|\mathcal{H}_s=\mathcal{H}_t)\mathrm{d}s\})\mathrm{d}\delta. \tag{7}$$

The last item in eq. (4) is

$$\mathbb{E}_{\widetilde{\mathcal{H}}\sim\widetilde{P}(\cdot|\mathcal{H}_t)}\left[\mathbf{1}\left[\delta^{a_{\mathrm{obs}}}(t)\wedge\delta^a(t)\wedge\delta^v(t)>T-t\right]\cdot\mathbb{E}_P\left[Y|\mathcal{H}_T=\mathcal{H}_t\right]\right]=$$
$$\left(1-1+\exp\left\{-\int_t^T\lambda^a_{\mathrm{obs}}(s|\mathcal{H}_s=\mathcal{H}_t)\right\}\right)\exp\left\{-\int_t^T\lambda_\bullet(s|\mathcal{H}_s=\mathcal{H}_t)\mathrm{d}s\right\}\mathbb{E}_P\left[Y|\mathcal{H}_T=\mathcal{H}_t\right] \tag{8}$$

Adding up eq. (6), eq. (7) and eq. (8) while pulling out all the items multiplied by 1 to the last brackets in the bottom expression, we get,

$$\mathbb{E}_{\widetilde{\mathcal{H}}\sim\widetilde{P}(\cdot|\mathcal{H}_t)}\left[\mathbf{1}\left[\delta^a(t)<\delta^v(t)\wedge\delta^{a_{\mathrm{obs}}}(t)\right]\cdot\mathbb{E}_P\left[Y|\mathcal{H}_t\cup(t+\delta^a(t),a)\right]+\right.$$
$$\mathbf{1}\left[\delta^v(t)<\delta^a(t)\wedge\delta^{a_{\mathrm{obs}}}(t)\right]\cdot\mathbb{E}_P\left[Y|\mathcal{H}_t\cup(t+\delta^v(t),v)\right]+$$
$$\left.\mathbf{1}\left[\delta^{a_{\mathrm{obs}}}(t)\wedge\delta^a(t)\wedge\delta^v(t)>T-t\right]\cdot\mathbb{E}_P\left[Y|\mathcal{H}_T=\mathcal{H}_t\right]\right]=$$
$$\left(\int_0^{T-t}\lambda^a(t+\delta|\mathcal{H}_{t+\delta}=\mathcal{H}_t)\exp\{-\int_t^{t+\delta}\lambda_\bullet(s|\mathcal{H}_s=\mathcal{H}_t)\mathrm{d}s\}\mathbb{E}\left[Y|\mathcal{H}_t\cup(t+\delta,a)\right]\right.$$
$$\left.\cdot(-1+\exp\{-\int_t^{t+\delta}\lambda^a_{\mathrm{obs}}(s|\mathcal{H}_s=\mathcal{H}_t)\mathrm{d}s\})\mathrm{d}\delta\right)$$

$$+\left(\int_0^{T-t}\lambda^v(t+\delta|\mathcal{H}_{t+\delta}=\mathcal{H}_t)\exp\{-\int_t^{t+\delta}\lambda_\bullet(s|\mathcal{H}_s=\mathcal{H}_t)\mathrm{d}s\}\mathbb{E}\left[Y|\mathcal{H}_t\cup(t+\delta,v)\right]\right.$$

$$\left.\cdot(-1+\exp\{-\int_t^{t+\delta}\lambda_{\mathrm{obs}}^a(s|\mathcal{H}_s=\mathcal{H}_t)\mathrm{d}s\})\mathrm{d}\delta\right)$$

$$+\left(\exp\left\{-\int_t^T\lambda_\bullet(s|\mathcal{H}_s=\mathcal{H}_t)\mathrm{d}s\right\}\mathbb{E}_P\left[Y|\mathcal{H}_T=\mathcal{H}_t\right]\right.$$

$$\left.\cdot\left(-1+\exp\left\{-\int_t^T\lambda_{\mathrm{obs}}^a(s|\mathcal{H}_s=\mathcal{H}_t)\right\}\right)\right)$$

$$+\left(\int_0^{T-t}\lambda^a(t+\delta|\mathcal{H}_{t+\delta}=\mathcal{H}_t)\exp\{-\int_t^{t+\delta}\lambda_\bullet(s|\mathcal{H}_s=\mathcal{H}_t)\mathrm{d}s\}\mathbb{E}\left[Y|\mathcal{H}_t\cup(t+\delta,a)\right]\right.$$

$$+\int_0^{T-t}\lambda^v(t+\delta|\mathcal{H}_{t+\delta}=\mathcal{H}_t)\exp\{-\int_t^{t+\delta}\lambda_\bullet(s|\mathcal{H}_s=\mathcal{H}_t)\mathrm{d}s\}\mathbb{E}\left[Y|\mathcal{H}_t\cup(t+\delta,v)\right]$$

$$\left.+\exp\left\{-\int_t^T\lambda_\bullet(s|\mathcal{H}_s=\mathcal{H}_t)\mathrm{d}s\right\}\mathbb{E}_P\left[Y|\mathcal{H}_T=\mathcal{H}_t\right]\right).$$

The items in the last brackets equal the right-hand-side of eq. (5). Plugging this in we rewrite,

$$\mathbb{E}_{\widetilde{\mathcal{H}}\sim\widetilde{P}(\cdot|\mathcal{H}_t)}\left[\mathbf{1}\left[\delta^a(t)<\delta^v(t)\wedge\delta^{a_\mathrm{obs}}(t)\right]\cdot\mathbb{E}_P\left[Y|\mathcal{H}_t\cup(t+\delta^a(t),a)\right]+\right.$$
$$\mathbf{1}\left[\delta^v(t)<\delta^a(t)\wedge\delta^{a_\mathrm{obs}}(t)\right]\cdot\mathbb{E}_P\left[Y|\mathcal{H}_t\cup(t+\delta^v(t),v)\right]+$$
$$\left.\mathbf{1}\left[\delta^{a_\mathrm{obs}}(t)\wedge\delta^a(t)\wedge\delta^v(t)>T-t\right]\cdot\mathbb{E}_P\left[Y|\mathcal{H}_T=\mathcal{H}_t\right]\right]=$$

$$\mathbb{E}_P\left[Y|\mathcal{H}_t\right]+$$

$$-\left(1-\exp\left\{-\int_t^T\lambda_{\mathrm{obs}}^a(s|\mathcal{H}_s=\mathcal{H}_t)\mathrm{d}s\right\}\right)\exp\left\{-\int_t^T\lambda_\bullet(s|\mathcal{H}_s=\mathcal{H}_t)\mathrm{d}s\right\}\mathbb{E}_P[Y|\mathcal{H}_T=\mathcal{H}_t]$$

$$-\int_0^{T-t}\left[\left(\lambda^v(t+\delta|\mathcal{H}_{t+\delta}=\mathcal{H}_t)\mathbb{E}_P[Y|\mathcal{H}_t\cup(t+\delta,x)]\right.\right.$$

$$\left.+\lambda^a(t+\delta|\mathcal{H}_{t+\delta}=\mathcal{H}_t)\mathbb{E}_P[Y|\mathcal{H}_t\cup(t+\delta,a)]\right)$$

$$\left.\cdot\exp\{-\int_t^{t+\delta}\lambda_\bullet(s|\mathcal{H}_s=\mathcal{H}_t)\mathrm{d}s\}\cdot\left(1-\exp\{-\int_t^{t+\delta}\lambda_{\mathrm{obs}}^a(s|\mathcal{H}_s=\mathcal{H}_t)\mathrm{d}s\}\right)\right]\mathrm{d}\delta. \quad (9)$$

Note that we have,

$$1-\exp\{-\int_t^{t+\delta}\lambda_{\mathrm{obs}}^a(s|\mathcal{H}_s=\mathcal{H}_t)\mathrm{d}s\}=$$

$$\int_t^{t+\delta}\lambda_{\mathrm{obs}}^a(s|\mathcal{H}_s=\mathcal{H}_t)\exp\{-\int_t^s\lambda_{\mathrm{obs}}^a(s|\mathcal{H}_s=\mathcal{H}_t)\mathrm{d}u\}\mathrm{d}s, \quad (10)$$

because the left-hand-side is 1 minus the probability that $N_a^{\mathrm{obs}}$ does not jump in the interval $(t,t+\delta]$, and the integration on the right hand side is the probability that the process jumps at least once (where the first jump is at time $s$).

Next, we write the third item of eq. (4) to see that it cancels the residual above in eq. (9).

$$\mathbb{E}_{\widetilde{\mathcal{H}}\sim\widetilde{P}(\cdot|\mathcal{H}_t)}\left[\mathbf{1}\left[\delta^{a_\mathrm{obs}}(t)<\delta^a(t)\wedge\delta^v(t)\right]\cdot\mathbb{E}\left[Y|\mathcal{H}_{t+\delta^{a_\mathrm{obs}}}=\mathcal{H}_t\right]\right]=$$

$$\int_0^{T-t}\lambda_{\mathrm{obs}}^a(t+\delta|\mathcal{H}_{t+\delta}=\mathcal{H}_t)\exp\{-\int_t^{t+\delta}\lambda_{\mathrm{obs}}^a(s|\mathcal{H}_s=\mathcal{H}_t)\mathrm{d}s\}$$

$$\exp\{-\int_t^{t+\delta} \lambda_\bullet(s|\mathcal{H}_s = \mathcal{H}_t)\mathrm{d}s\}\mathbb{E}_P\left[Y|\mathcal{H}_{t+\delta} = \mathcal{H}_t\right]\mathrm{d}\delta. \tag{11}$$

We expand $\mathbb{E}_P\left[Y|\mathcal{H}_{t+\delta} = \mathcal{H}_t\right]$ again by towering expectations w.r.t to the first jump after $t+\delta$,

$$\mathbb{E}_P[Y|\mathcal{H}_{t+\delta} = \mathcal{H}_t] = \int_{t+\delta}^T \Big(\lambda^a(s'|\mathcal{H}_s = \mathcal{H}_t)\mathbb{E}_P\left[Y|\mathcal{H}_{s'} = \mathcal{H}_t \cup (s',a)\right] +$$

$$\lambda^v(s'|\mathcal{H}_{s'} = \mathcal{H}_t)\mathbb{E}_P\left[Y|\mathcal{H}_{s'} = \mathcal{H}_t \cup (s',v)\right]\Big)\exp\{-\int_{t+\delta}^{s'}\lambda_\bullet(u|\mathcal{H}_u = \mathcal{H}_t)\mathrm{d}u\}\mathrm{d}s'$$

$$+ \mathbb{E}_P[Y|\mathcal{H}_T = \mathcal{H}_t]\exp\{-\int_{t+\delta}^T \lambda_\bullet(u|\mathcal{H}_u = \mathcal{H}_t)\mathrm{d}u\}$$

Multiplying the left hand side by $\exp\{-\int_t^{t+\delta}\lambda_\bullet(s|\mathcal{H}_s = \mathcal{H}_t)\mathrm{d}s\}$ we get a similar item where the integration on $\lambda_\bullet$ starts from $t$ instead of $t+\delta$,

$$\exp\{-\int_t^{t+\delta}\lambda_\bullet(s|\mathcal{H}_s = \mathcal{H}_t)\mathrm{d}s\}\mathbb{E}_P\left[Y|\mathcal{H}_{t+\delta} = \mathcal{H}_t\right] =$$

$$\int_{t+\delta}^T \Big(\lambda^a(s'|\mathcal{H}_{s'} = \mathcal{H}_t)\mathbb{E}_P\left[Y|\mathcal{H}_{s'} = \mathcal{H}_t \cup (s',a)\right] +$$

$$\lambda^v(s'|\mathcal{H}_{s'} = \mathcal{H}_t)\mathbb{E}_P\left[Y|\mathcal{H}_{s'} = \mathcal{H}_t \cup (s',v)\right]\Big)\exp\{-\int_t^{s'}\lambda_\bullet(u|\mathcal{H}_u = \mathcal{H}_t)\mathrm{d}u\}\mathrm{d}s'$$

$$+ \mathbb{E}_P[Y|\mathcal{H}_T = \mathcal{H}_t]\exp\{-\int_t^T \lambda_\bullet(u|\mathcal{H}_u = \mathcal{H}_t)\mathrm{d}u\}$$

Let us denote the gray item by $(*)$, and multiply by the observed part in eq. (11). In the first equality we will pull the integration on $s'$ outside, then we will change the order of integration, change variables by a constant shift ($\delta' = s' - t, \tilde{s} = \delta + t$), and push one integration back inside.

$$\int_0^{T-t} \lambda_{\mathrm{obs}}^a(t+\delta|\mathcal{H}_{t+\delta} = \mathcal{H}_t)\exp\{-\int_t^{t+\delta}\lambda_{\mathrm{obs}}^a(s|\mathcal{H}_s = \mathcal{H}_t)\mathrm{d}s\} \cdot (*) =$$

$$\int_0^{T-t} \lambda_{\mathrm{obs}}^a(t+\delta|\mathcal{H}_{t+\delta} = \mathcal{H}_t)\exp\{-\int_t^{t+\delta}\lambda_{\mathrm{obs}}^a(s|\mathcal{H}_s = \mathcal{H}_t)\mathrm{d}s\}\cdot$$

$$\left[\int_{t+\delta}^T \Big(\lambda^a(s'|\mathcal{H}_{s'} = \mathcal{H}_t)\mathbb{E}_P\left[Y|\mathcal{H}_{s'} = \mathcal{H}_t \cup (s',a)\right] +\right.$$

$$\left. \lambda^v(s'|\mathcal{H}_{s'} = \mathcal{H}_t)\mathbb{E}_P\left[Y|\mathcal{H}_{s'} = \mathcal{H}_t \cup (s',v)\right]\Big)\exp\{-\int_t^{s'}\lambda_\bullet(u|\mathcal{H}_u = \mathcal{H}_t)\mathrm{d}u\}\mathrm{d}s'\right]\mathrm{d}\delta$$

$$= \int_0^{T-t}\int_{t+\delta}^T \lambda_{\mathrm{obs}}^a(t+\delta|\mathcal{H}_{t+\delta} = \mathcal{H}_t)\exp\{-\int_t^{t+\delta}\lambda_{\mathrm{obs}}^a(s|\mathcal{H}_s = \mathcal{H}_t)\mathrm{d}s\}\cdot$$

$$\Big(\lambda^a(s'|\mathcal{H}_{s'} = \mathcal{H}_t)\mathbb{E}_P\left[Y|\mathcal{H}_{s'} = \mathcal{H}_t \cup (s',a)\right] +$$

$$\lambda^v(s'|\mathcal{H}_{s'} = \mathcal{H}_t)\mathbb{E}_P\left[Y|\mathcal{H}_{s'} = \mathcal{H}_t \cup (s',v)\right]\Big)\exp\{-\int_t^{s'}\lambda_\bullet(u|\mathcal{H}_u = \mathcal{H}_t)\mathrm{d}u\}\mathrm{d}s'\mathrm{d}\delta$$

$$= \int_t^T\int_0^{s'-t} \lambda_{\mathrm{obs}}^a(t+\delta|\mathcal{H}_{t+\delta} = \mathcal{H}_t)\exp\{-\int_t^{t+\delta}\lambda_{\mathrm{obs}}^a(s|\mathcal{H}_s = \mathcal{H}_t)\mathrm{d}s\}\cdot$$

$$\Big(\lambda^a(s'|\mathcal{H}_{s'} = \mathcal{H}_t)\mathbb{E}_P\left[Y|\mathcal{H}_{s'} = \mathcal{H}_t \cup (s',a)\right] +$$

$$\lambda^v(s'|\mathcal{H}_{s'} = \mathcal{H}_t)\mathbb{E}_P\left[Y|\mathcal{H}_{s'} = \mathcal{H}_t \cup (s',v)\right]\Big)\exp\{-\int_t^{s'}\lambda_\bullet(u|\mathcal{H}_u = \mathcal{H}_t)\mathrm{d}u\}\mathrm{d}\delta\mathrm{d}s'$$

$$= \int_0^{T-t}\int_t^{t+\delta'} \lambda_{\mathrm{obs}}^a(\tilde{s}|\mathcal{H}_{\tilde{s}} = \mathcal{H}_t)\exp\{-\int_t^{\tilde{s}}\lambda_{\mathrm{obs}}^a(s|\mathcal{H}_s = \mathcal{H}_t)\mathrm{d}s\}\cdot$$

$$\Big(\lambda^a(t+\delta'|\mathcal{H}_{t+\delta'}=\mathcal{H}_t)\mathbb{E}_P\left[Y|\mathcal{H}_{t+\delta'}=\mathcal{H}_t\cup(t+\delta',a)\right]+$$

$$\lambda^v(t+\delta'|\mathcal{H}_{t+\delta'}=\mathcal{H}_t)\mathbb{E}_P\left[Y|\mathcal{H}_{t+\delta'}=\mathcal{H}_t\cup(t+\delta',v)\right]\Big)\cdot$$

$$\exp\{-\int_t^{t+\delta'}\lambda_\bullet(u|\mathcal{H}_u=\mathcal{H}_t)\mathrm{d}u\}\mathrm{d}\tilde{s}\mathrm{d}\delta'$$

$$=\int_0^{T-t}\Big(\lambda^a(t+\delta'|\mathcal{H}_{t+\delta'}=\mathcal{H}_t)\mathbb{E}_P\left[Y|\mathcal{H}_{t+\delta'}=\mathcal{H}_t\cup(t+\delta',a)\right]+$$

$$\lambda^v(t+\delta'|\mathcal{H}_{t+\delta'}=\mathcal{H}_t)\mathbb{E}_P\left[Y|\mathcal{H}_{t+\delta'}=\mathcal{H}_t\cup(t+\delta',v)\right]\Big)\cdot$$

$$\exp\{-\int_t^{t+\delta'}\lambda_\bullet(u|\mathcal{H}_u=\mathcal{H}_t)\mathrm{d}u\}\left(\int_t^{t+\delta'}\lambda_{\mathrm{obs}}^a(\tilde{s}|\mathcal{H}_{\tilde{s}}=\mathcal{H}_t)\exp\{-\int_t^{\tilde{s}}\lambda_{\mathrm{obs}}^a(s|\mathcal{H}_s=\mathcal{H}_t)\mathrm{d}s\}\mathrm{d}\tilde{s}\right)\mathrm{d}\delta'$$

Plugging everything back into eq. (11), we color in red the same item colored red above, where we change the name of variables back from $\delta',\tilde{s}$ to $\delta,s$ for convenience. The remaining item in the equation is obtained by collecting all the items that multiply $\mathbb{E}_P[Y|\mathcal{H}_T=\mathcal{H}_t]$ in the obtained expression.

$$\mathbb{E}_{\widetilde{\mathcal{H}}\sim\widetilde{P}(\cdot|\mathcal{H}_t)}\Big[\mathbf{1}\big[\delta^{a_{\mathrm{obs}}}(t)<\delta^a(t)\wedge\delta^v(t)\big]\mathbb{E}\left[Y|\mathcal{H}_{t+\delta^{a_{\mathrm{obs}}}}=\mathcal{H}_t\right]\Big]=$$

$$\int_0^{T-t}\Big(\lambda^a(t+\delta|\mathcal{H}_{t+\delta}=\mathcal{H}_t)\mathbb{E}_P\left[Y|\mathcal{H}_{t+\delta}=\mathcal{H}_t\cup(t+\delta,a)\right]+$$

$$\lambda^v(t+\delta|\mathcal{H}_{t+\delta}=\mathcal{H}_t)\mathbb{E}_P\left[Y|\mathcal{H}_{t+\delta}=\mathcal{H}_t\cup(t+\delta,x)\right]\Big)$$

$$\exp\{-\int_t^{t+\delta}\lambda_\bullet(s|\mathcal{H}_s=\mathcal{H}_t)\mathrm{d}s\}\left(\int_t^{t+\delta}\lambda_{\mathrm{obs}}^a(s|\mathcal{H}_s=\mathcal{H}_t)\exp\{-\int_t^s\lambda_{\mathrm{obs}}^a(u|\mathcal{H}_u=\mathcal{H}_t)\mathrm{d}u\}\mathrm{d}s\right)\mathrm{d}\delta$$

$$+\left(\int_0^{T-t}\lambda_{\mathrm{obs}}^a(t+\delta|\mathcal{H}_{t+\delta}=\mathcal{H}_t)\exp\{-\int_t^{t+\delta}\lambda_{\mathrm{obs}}^a(s|\mathcal{H}_s=\mathcal{H}_t)\mathrm{d}s\}\mathrm{d}\delta\right)\cdot$$

$$\mathbb{E}_P[Y|\mathcal{H}_T=\mathcal{H}_t]\exp\{-\int_t^T\lambda_\bullet(s|\mathcal{H}_s=\mathcal{H}_t)\mathrm{d}s\}=$$

$$\int_0^{T-t}\Big(\lambda^a(t+\delta|\mathcal{H}_{t+\delta}=\mathcal{H}_t)\mathbb{E}_P\left[Y|\mathcal{H}_{t+\delta}=\mathcal{H}_t\cup(t+\delta,a)\right]+$$

$$\lambda^v(t+\delta|\mathcal{H}_{t+\delta}=\mathcal{H}_t)\mathbb{E}_P\left[Y|\mathcal{H}_{t+\delta}=\mathcal{H}_t\cup(t+\delta,x)\right]\Big)$$

$$\exp\{-\int_t^{t+\delta}\lambda_\bullet(s|\mathcal{H}_s=\mathcal{H}_t)\mathrm{d}s\}\left(1-\exp\{-\int_t^{t+\delta}\lambda_{\mathrm{obs}}^a(s|\mathcal{H}_s=\mathcal{H}_t)\mathrm{d}s\}\right)\mathrm{d}\delta$$

$$+\left(1-\exp\{-\int_t^T\lambda_{\mathrm{obs}}^a(s|\mathcal{H}_s=\mathcal{H}_t)\mathrm{d}s\}\right)\cdot\mathbb{E}_P[Y|\mathcal{H}_T=\mathcal{H}_t]\exp\{-\int_t^T\lambda_\bullet(s|\mathcal{H}_s=\mathcal{H}_t)\mathrm{d}s\}.$$

In the last equality we plugged in eq. (10). Now it can be seen that the above expression cancels with the residual of eq. (9), which means that eq. (4) holds as claimed. $\qquad\square$

As we explain in the sequel, theorem 1 follows directly from the lemma below.

**Lemma 2.** *For any $t\in[0,T)$ and $k\in\mathbb{N}$, define $\delta^k(t)>0$ such that $t+\delta^k(t)$ is the time of the $k$-th event after $t$ in a trajectory $\widetilde{\mathcal{H}}$, where $\delta^0(t)=0$ as an edge case.[7] That is, assuming $\widetilde{\mathcal{H}}=\{(t_j,v_j)\}_{j\in\mathbb{N}}$ then $\delta_{\mathcal{H}}^k(t):=\min\{t_j-t:t_{j-k+1}>t\}$. Analogously, we define $\delta^{k,e}(t)$ as $\delta_{\widetilde{\mathcal{H}}^e}^k(t)$, which is the $k$-th event of type $e$. For all $d\in\mathbb{N}_+$ we have that*

$$\mathbb{E}_P[Y|\mathcal{H}_t]=\mathbb{E}_{\widetilde{\mathcal{H}}\sim\widetilde{P}(\cdot|\mathcal{H}_t)}\Big[ \tag{12}$$

---

[7]as explained in appendix B.1, the full notation should be $\delta_{\widetilde{\mathcal{H}}}^k(t)$, but $\widetilde{\mathcal{H}}$ will be clear from context.

$$\sum_{k=1}^{d}\Big(\mathbf{1}\big[\delta^{k-1,v}(t)\le\delta^{a}(t)<\delta^{k,v}(t)\wedge\delta^{a_{\mathrm{obs}}}(t)\wedge T-t\big]\mathbb{E}_{P}\big[Y|\mathcal{H}_{t}\cup\widetilde{\mathcal{H}}^{\backslash a_{\mathrm{obs}}}_{(t,t+\delta^{a}(t)]}\big]$$

$$+\mathbf{1}\big[\delta^{k-1,v}(t)\le\delta^{a_{\mathrm{obs}}}(t)<\delta^{k,v}(t)\wedge\delta^{a}(t)\wedge T-t\big]\mathbb{E}_{P}\big[Y|\mathcal{H}_{t}\cup\widetilde{\mathcal{H}}^{\backslash a_{\mathrm{obs}}}_{(t,t+\delta^{a_{\mathrm{obs}}}(t)]}\big]\Big)$$

$$+\mathbf{1}\big[T-t<\delta^{d,v}(t)\wedge\delta^{a_{\mathrm{obs}}}(t)\wedge\delta^{a}(t)\big]\mathbb{E}_{P}[Y|\mathcal{H}_{t}\cup\widetilde{\mathcal{H}}^{\backslash a_{\mathrm{obs}}}_{(t,T]}]$$

$$+\mathbf{1}\big[\delta^{d,v}(t)<\delta^{a_{\mathrm{obs}}}(t)\wedge\delta^{a}(t)\wedge T-t\big]\mathbb{E}_{P}\big[Y|\mathcal{H}_{t}\cup\widetilde{\mathcal{H}}^{\backslash a_{\mathrm{obs}}}_{(t,t+\delta^{d,v}(t)]}\big]\Big].$$

Now let us recall theorem 1 and prove it, assuming that lemma 2 holds. Then we will prove lemma 2, which completes the proofs of our claims.

**Theorem 1.** *Let $P_{\mathrm{obs}}$ be a marked decision point processes, $P$ the process obtained by replacing the policy with $(\lambda^{a},\pi)$, and $\tilde{P}$ the augmented process obtained from $P$, $P_{\mathrm{obs}}$ in definition 4. Further, let $t\in[0,T)$, and $\mathcal{H}_{t}$ measurable w.r.t $P$. Under Assumption 2, we have that*

$$\mathbb{E}_{P}\left[Y|\mathcal{H}_{t}\right]=\mathbb{E}_{\widetilde{\mathcal{H}}\sim\widetilde{P}(\cdot|\mathcal{H}_{t})}\left[\mathbb{E}_{P}\left[Y\;\Big|\;\mathcal{H}_{t+\delta_{\widetilde{\mathcal{H}}}(t)}=\mathcal{H}_{t}\cup\widetilde{\mathcal{H}}^{\backslash a_{\mathrm{obs}}}_{\left(t,t+\delta_{\widetilde{\mathcal{H}}}(t)\right]}\right]\right]. \tag{2}$$

*Proof of theorem 1.* Examine eq. (12) when $d\to\infty$, because we assume the number of events is finite it holds that

$$\lim_{d\to\infty}\widetilde{P}\left(\delta^{d,v}(t)<\delta^{a_{\mathrm{obs}}}(t)\wedge\delta^{a}(t)\wedge T-t|\mathcal{H}_{t}\right)=0.$$

That is because otherwise, the next treatments should occur after an infinite number of events, and we need to have infinitely many observations before time $T-t$. By convention, we define $\delta^{k,v}(t)=\infty$ whenever there fewer than $k$ events of type $v$ in the interval $(t,T]$. Therefore, also due to the finite amount of events in $(t,T]$, it holds that

$$\lim_{d\to\infty}\widetilde{P}\left(T-t<\delta^{d,v}(t)\wedge\delta^{a_{\mathrm{obs}}}(t)\wedge\delta^{a}(t)|\mathcal{H}_{t}\right)=\lim_{d\to\infty}\widetilde{P}\left(T-t<\delta^{a_{\mathrm{obs}}}(t)\wedge\delta^{a}(t)|\mathcal{H}_{t}\right).$$

Then we have,

$$\lim_{d\to\infty}\mathbb{E}_{\widetilde{\mathcal{H}}\sim\widetilde{P}(\cdot|\mathcal{H}_{t})}\Bigg[$$

$$\sum_{k=1}^{d}\Big(\mathbf{1}\big[\delta^{k-1,v}(t)\le\delta^{a}(t)<\delta^{k,v}(t)\wedge\delta^{a_{\mathrm{obs}}}(t)\wedge T-t\big]\mathbb{E}_{P}\big[Y|\mathcal{H}_{t}\cup\widetilde{\mathcal{H}}^{\backslash a_{\mathrm{obs}}}_{(t,t+\delta^{a}(t)]}\big]$$

$$+\mathbf{1}\big[\delta^{k-1,v}(t)\le\delta^{a_{\mathrm{obs}}}(t)<\delta^{k,v}(t)\wedge\delta^{a}(t)\wedge T-t\big]\mathbb{E}_{P}\big[Y|\mathcal{H}_{t}\cup\widetilde{\mathcal{H}}^{\backslash a_{\mathrm{obs}}}_{(t,t+\delta^{a_{\mathrm{obs}}}(t)]}\big]\Big)$$

$$+\mathbf{1}\big[T-t<\delta^{d,v}(t)\wedge\delta^{a_{\mathrm{obs}}}(t)\wedge\delta^{a}(t)\big]\mathbb{E}_{P}[Y|\mathcal{H}_{t}\cup\widetilde{\mathcal{H}}^{\backslash a_{\mathrm{obs}}}_{(t,T]}]$$

$$+\mathbf{1}\big[\delta^{d,v}(t)<\delta^{a_{\mathrm{obs}}}(t)\wedge\delta^{a}(t)\wedge T-t\big]\mathbb{E}_{P}\big[Y|\mathcal{H}_{t}\cup\widetilde{\mathcal{H}}^{\backslash a_{\mathrm{obs}}}_{(t,t+\delta^{d,v}(t)]}\big]\Bigg]=$$

$$\lim_{d\to\infty}\mathbb{E}_{\widetilde{\mathcal{H}}\sim\widetilde{P}(\cdot|\mathcal{H}_{t})}\Bigg[$$

$$\sum_{k=1}^{d}\Big(\mathbf{1}\big[\delta^{k-1,v}(t)\le\delta^{a}(t)<\delta^{k,v}(t)\wedge\delta^{a_{\mathrm{obs}}}(t)\wedge T-t\big]\mathbb{E}_{P}\big[Y|\mathcal{H}_{t}\cup\widetilde{\mathcal{H}}^{\backslash a_{\mathrm{obs}}}_{(t,t+\delta^{a}(t)]}\big]$$

$$+\mathbf{1}\big[\delta^{k-1,v}(t)\le\delta^{a_{\mathrm{obs}}}(t)<\delta^{k,v}(t)\wedge\delta^{a}(t)\wedge T-t\big]\mathbb{E}_{P}\big[Y|\mathcal{H}_{t}\cup\widetilde{\mathcal{H}}^{\backslash a_{\mathrm{obs}}}_{(t,t+\delta^{a_{\mathrm{obs}}}(t)]}\big]\Big)$$

$$+\mathbf{1}\big[T-t<\delta^{a_{\mathrm{obs}}}(t)\wedge\delta^{a}(t)\big]\mathbb{E}_{P}[Y|\mathcal{H}_{t}\cup\widetilde{\mathcal{H}}^{\backslash a_{\mathrm{obs}}}_{(t,T]}]\Bigg]$$

Note that the limit when $d\to\infty$ exists since from lemma 2 the expectation has the same value for any value of $d$. Next we make two observations:

- The conditioning in the $\mathbb{E}_P[Y|\ldots]$ terms can be rewritten as $\mathcal{H}_t \cup \widetilde{\mathcal{H}}^{\backslash a_{\mathrm{obs}}}_{(t,t+\delta_{\widetilde{\mathcal{H}}}(t)]}$ for all items. This is because by definition of $\delta_{\widetilde{\mathcal{H}}}(t)$ as $\min\{u-t : u > t, (u,\cdot) \in \widetilde{\mathcal{H}}^{a,a_{\mathrm{obs}}}\}$, or $\delta_{\widetilde{\mathcal{H}}}(t) = T - t$ when the set is empty,

$$\mathbf{1}\big[\delta^{k-1,v}(t) \le \delta^a(t) < \delta^{k,v}(t) \wedge \delta^{a_{\mathrm{obs}}}(t) \wedge T - t\big] = 1 \Rightarrow \delta_{\widetilde{\mathcal{H}}}(t) = \delta^a(t),$$
$$\mathbf{1}\big[\delta^{k-1,v}(t) \le \delta^{a_{\mathrm{obs}}}(t) < \delta^{k,v}(t) \wedge \delta^a(t) \wedge T - t\big] = 1 \Rightarrow \delta_{\widetilde{\mathcal{H}}}(t) = \delta^{a_{\mathrm{obs}}}(t),$$
$$\mathbf{1}\big[T - t < \delta^{a_{\mathrm{obs}}}(t) \wedge \delta^a(t)\big] = 1 \Rightarrow \delta_{\widetilde{\mathcal{H}}}(t) = T - t.$$

  In the next step we will replace these times with $\delta_{\widetilde{\mathcal{H}}}(t)$ for all items.

- All the events in the indicators are mutually exclusive, and exactly one of them occurs for each $\widetilde{\mathcal{H}}$. This is since $\delta^a(t) \wedge \delta^{a_{\mathrm{obs}}}(t)$ either occurs between the $k-1$ and $k$-th event of type $v$ for some value of $k$, or $\delta^a(t) \wedge \delta^{a_{\mathrm{obs}}}(t) > T - t$.

Combining these observations into the expressions we developed for $\mathbb{E}_P[Y|\mathcal{H}_t]$, we conclude the proof,

$$\mathbb{E}_P[Y|\mathcal{H}_t] =$$

$$\lim_{d\to\infty} \mathbb{E}_{\widetilde{\mathcal{H}}\sim\widetilde{P}(\cdot|\mathcal{H}_t)}\Bigg[$$

$$\sum_{k=1}^d \Big(\mathbf{1}\big[\delta^{k-1,v}(t) \le \delta^a(t) < \delta^{k,v}(t) \wedge \delta^{a_{\mathrm{obs}}}(t) \wedge T - t\big]\mathbb{E}_P\big[Y|\mathcal{H}_t \cup \widetilde{\mathcal{H}}^{\backslash a_{\mathrm{obs}}}_{(t,t+\delta^a(t)]}\big]$$

$$+ \mathbf{1}\big[\delta^{k-1,v}(t) \le \delta^{a_{\mathrm{obs}}}(t) < \delta^{k,v}(t) \wedge \delta^a(t) \wedge T - t\big]\mathbb{E}_P\big[Y|\mathcal{H}_t \cup \widetilde{\mathcal{H}}^{\backslash a_{\mathrm{obs}}}_{(t,t+\delta^{a_{\mathrm{obs}}}(t)]}\big]\Big)$$

$$+ \mathbf{1}\big[T - t < \delta^{a_{\mathrm{obs}}}(t) \wedge \delta^a(t)\big]\mathbb{E}_P[Y|\mathcal{H}_t \cup \widetilde{\mathcal{H}}^{\backslash a_{\mathrm{obs}}}_{(t,T]}]\Bigg] =$$

$$\lim_{d\to\infty} \mathbb{E}_{\widetilde{\mathcal{H}}\sim\widetilde{P}(\cdot|\mathcal{H}_t)}\Bigg[$$

$$\sum_{k=1}^d \Big(\mathbf{1}\big[\delta^{k-1,v}(t) \le \delta^a(t) < \delta^{k,v}(t) \wedge \delta^{a_{\mathrm{obs}}}(t) \wedge T - t\big]\mathbb{E}_P\big[Y|\mathcal{H}_t \cup \widetilde{\mathcal{H}}^{\backslash a_{\mathrm{obs}}}_{(t,t+\delta_{\widetilde{\mathcal{H}}}(t)]}\big]$$

$$+ \mathbf{1}\big[\delta^{k-1,v}(t) \le \delta^{a_{\mathrm{obs}}}(t) < \delta^{k,v}(t) \wedge \delta^a(t) \wedge T - t\big]\mathbb{E}_P\big[Y|\mathcal{H}_t \cup \widetilde{\mathcal{H}}^{\backslash a_{\mathrm{obs}}}_{(t,t+\delta_{\widetilde{\mathcal{H}}}(t)]}\big]\Big)$$

$$+ \mathbf{1}\big[T - t < \delta^{a_{\mathrm{obs}}}(t) \wedge \delta^a(t)\big]\mathbb{E}_P[Y|\mathcal{H}_t \cup \widetilde{\mathcal{H}}^{\backslash a_{\mathrm{obs}}}_{(t,t+\delta_{\widetilde{\mathcal{H}}}(t)]}]\Bigg] =$$

$$\mathbb{E}_{\widetilde{\mathcal{H}}\sim\widetilde{P}(\cdot|\mathcal{H}_t)}\big[\mathbb{E}_P[Y|\mathcal{H}_t \cup \widetilde{\mathcal{H}}^{\backslash a_{\mathrm{obs}}}_{(t,t+\delta_{\widetilde{\mathcal{H}}}(t)]}]\big]$$

$$\square$$

Next, let us complete the proof of the required lemma that we assumed to hold.

*Proof of lemma 2.* For $d = 1$, eq. (12) is exactly eq. (4) which we already proved in lemma 1, and we will proceed by induction. Assume for some $d - 1 > 1$ that eq. (12) holds, this hypothesis is written below,

$$\mathbb{E}_P[Y|\mathcal{H}_t] = \mathbb{E}_{\widetilde{\mathcal{H}}\sim\widetilde{P}(\cdot|\mathcal{H}_t)}\Bigg[ \tag{13}$$

$$\sum_{k=1}^{d-1} \Big(\mathbf{1}\big[\delta^{k-1,v}(t) \le \delta^a(t) < \delta^{k,v}(t) \wedge \delta^{a_{\mathrm{obs}}}(t) \wedge T - t\big]\mathbb{E}_P\big[Y|\mathcal{H}_t \cup \widetilde{\mathcal{H}}^{\backslash a_{\mathrm{obs}}}_{(t,t+\delta^a(t)]}\big]$$

$$+ \mathbf{1}\big[\delta^{k-1,v}(t) \le \delta^{a_{\mathrm{obs}}}(t) < \delta^{k,v}(t) \wedge \delta^a(t) \wedge T - t\big]\mathbb{E}_P\big[Y|\mathcal{H}_t \cup \widetilde{\mathcal{H}}^{\backslash a_{\mathrm{obs}}}_{(t,t+\delta^{a_{\mathrm{obs}}}(t)]}\big]$$

$$+ \mathbf{1}\big[T - t < \delta^{d-1,v}(t) \wedge \delta^{a_{\mathrm{obs}}}(t) \wedge \delta^a(t)\big]\mathbb{E}_P[Y|\mathcal{H}_t \cup \widetilde{\mathcal{H}}^{\backslash a_{\mathrm{obs}}}_{(t,T]}]\Big)$$

$$+ \mathbf{1}\big[\delta^{d-1,v}(t) \le \delta^{a_{\mathrm{obs}}}(t) \wedge \delta^a(t) \wedge T - t\big] \mathbb{E}_P\big[Y|\mathcal{H}_t \cup \widetilde{\mathcal{H}}^{\backslash a_{\mathrm{obs}}}_{(t,t+\delta^{d-1,v}(t)]}\big]\big].$$

Using the notation $t_{d-1} = t + \delta^{d-1,v}(t)$ for shorthand, we may rewrite the last summand as

$$\mathbb{E}_{\widetilde{\mathcal{H}} \sim \widetilde{P}(\cdot|\mathcal{H}_t)}\Big[\mathbf{1}\big[\delta^{d-1,v}(t) \le \delta^{a_{\mathrm{obs}}}(t) \wedge \delta^a(t) \wedge T - t\big] \mathbb{E}_P\big[Y|\mathcal{H}_t \cup \widetilde{\mathcal{H}}^{\backslash a_{\mathrm{obs}}}_{(t,t_{d-1}]}\big]\Big] =$$

$$\mathbb{E}_{\widetilde{\mathcal{H}}_{t_{d-1}} \sim \widetilde{P}(\cdot|\mathcal{H}_t)}\bigg[$$

$$\mathbb{E}_{\widetilde{\mathcal{H}}_{(t_{d-1},T]} \sim \widetilde{P}(\cdot|\widetilde{\mathcal{H}}_{t_{d-1}})}\Big[\mathbf{1}\big[\delta^{d-1,v}(t) \le \delta^{a_{\mathrm{obs}}}(t) \wedge \delta^a(t) \wedge T - t\big] \mathbb{E}_P\big[Y|\mathcal{H}_t \cup \widetilde{\mathcal{H}}^{\backslash a_{\mathrm{obs}}}_{(t,t_{d-1}]}\big]\Big]\bigg] =$$

$$\mathbb{E}_{\widetilde{\mathcal{H}}_{t_{d-1}} \sim \widetilde{P}(\cdot|\mathcal{H}_t)}\Big[\mathbf{1}\big[\delta^{d-1,v}(t) \le \delta^{a_{\mathrm{obs}}}(t) \wedge \delta^a(t) \wedge T - t\big] \mathbb{E}_P\big[Y|\mathcal{H}_t \cup \widetilde{\mathcal{H}}^{\backslash a_{\mathrm{obs}}}_{(t,t_{d-1}]}\big]\Big].$$

The first equality above used the law of total probability, while the second holds since the event $\mathbf{1}\big[\delta^{d-1,v}(t) \le \delta^{a_{\mathrm{obs}}}(t) \wedge \delta^a(t) \wedge T - t\big]$ and expectation $\mathbb{E}_P\big[Y|\mathcal{H}_t \cup \widetilde{\mathcal{H}}^{\backslash a_{\mathrm{obs}}}_{(t,t_{d-1}]}\big]$ only depend on events up to time $t_{d-1}$. Next we expand the latter term, $\mathbb{E}_P\big[Y|\mathcal{H}_t \cup \widetilde{\mathcal{H}}^{\backslash a_{\mathrm{obs}}}_{(t,t_{d-1}]}\big]$ according to lemma 1 and plug-in to the equation above.

$$\mathbb{E}_{\widetilde{\mathcal{H}}_{t_{d-1}} \sim \widetilde{P}(\cdot|\mathcal{H}_t)}\Big[\mathbf{1}\big[\delta^{d-1,v}(t) \le \delta^{a_{\mathrm{obs}}}(t) \wedge \delta^a(t) \wedge T - t\big] \mathbb{E}_P\big[Y|\mathcal{H}_t \cup \widetilde{\mathcal{H}}^{\backslash a_{\mathrm{obs}}}_{(t,t_{d-1}]}\big]\Big] =$$

$$\mathbb{E}_{\widetilde{\mathcal{H}}_{t_{d-1}} \sim \widetilde{P}(\cdot|\mathcal{H}_t)}\bigg[\mathbf{1}\big[\delta^{d-1,v}_{\widetilde{\mathcal{H}}}(t) \le \delta^{a_{\mathrm{obs}}}_{\widetilde{\mathcal{H}}}(t) \wedge \delta^a_{\widetilde{\mathcal{H}}}(t) \wedge T - t\big] \cdot$$

$$\mathbb{E}_{\widetilde{\widetilde{\mathcal{H}}} \sim \widetilde{P}\big(\cdot\big|\mathcal{H}_t \cup \widetilde{\mathcal{H}}^{\backslash a_{\mathrm{obs}}}_{(t,t_{d-1}]}\big)}\bigg[\Big(\mathbf{1}\big[\delta^a_{\widetilde{\widetilde{\mathcal{H}}}}(t_{d-1}) < \delta^v_{\widetilde{\widetilde{\mathcal{H}}}}(t_{d-1}) \wedge \delta^{a_{\mathrm{obs}}}_{\widetilde{\widetilde{\mathcal{H}}}}(t_{d-1})\big] \cdot$$

$$\mathbb{E}_P\Big[Y|\mathcal{H}_t \cup \widetilde{\mathcal{H}}^{\backslash a_{\mathrm{obs}}}_{(t,t_{d-1}]} \cup \big(t_{d-1} + \delta^a_{\widetilde{\widetilde{\mathcal{H}}}}(t_{d-1}), a\big)\Big]\Big) +$$

$$\Big(\mathbf{1}\big[\delta^{a_{\mathrm{obs}}}_{\widetilde{\widetilde{\mathcal{H}}}}(t_{d-1}) < \delta^a_{\widetilde{\widetilde{\mathcal{H}}}}(t_{d-1}) \wedge \delta^v_{\widetilde{\widetilde{\mathcal{H}}}}(t_{d-1})\big] \cdot$$

$$\mathbb{E}_P\Big[Y|\mathcal{H}_{t+\delta^{a_{\mathrm{obs}}}_{\widetilde{\widetilde{\mathcal{H}}}}(t_{d-1})} = \mathcal{H}_t \cup \widetilde{\mathcal{H}}^{\backslash a_{\mathrm{obs}}}_{(t,t_{d-1}]}\Big]\Big) +$$

$$\Big(\mathbf{1}\big[\delta^{a_{\mathrm{obs}}}_{\widetilde{\widetilde{\mathcal{H}}}}(t_{d-1}) \wedge \delta^a_{\widetilde{\widetilde{\mathcal{H}}}}(t_{d-1}) \wedge \delta^v_{\widetilde{\widetilde{\mathcal{H}}}}(t_{d-1}) > T - t_{d-1}\big] \cdot$$

$$\mathbb{E}_P\Big[Y|\mathcal{H}_T = \mathcal{H}_t \cup \widetilde{\mathcal{H}}^{\backslash a_{\mathrm{obs}}}_{(t,t_{d-1}]}\Big]\Big) +$$

$$\Big(\mathbf{1}\big[\delta^v_{\widetilde{\widetilde{\mathcal{H}}}}(t_{d-1}) < \delta^a_{\widetilde{\widetilde{\mathcal{H}}}}(t_{d-1}) \wedge \delta^{a_{\mathrm{obs}}}_{\widetilde{\widetilde{\mathcal{H}}}}(t_{d-1})\big] \cdot$$

$$\mathbb{E}_P\Big[Y|\mathcal{H}_t \cup \widetilde{\mathcal{H}}^{\backslash a_{\mathrm{obs}}}_{(t,t_{d-1}]} \cup \big(t_{d-1} + \delta^v_{\widetilde{\widetilde{\mathcal{H}}}}(t_{d-1}), v\big)\Big]\Big)\bigg]\bigg]$$

Next we pull out the expectation over $\widetilde{\widetilde{\mathcal{H}}}$, and then use the law of total probability to turn this into a single expectation over a trajectory $\widetilde{\mathcal{H}}$ drawn from $\widetilde{P}(\cdot|\mathcal{H}_t)$. Each occurrence of $\widetilde{\widetilde{\mathcal{H}}}$ will then be changed to $\widetilde{\mathcal{H}}$ accordingly.

$$\mathbb{E}_{\widetilde{\mathcal{H}}_{t_{d-1}} \sim \widetilde{P}(\cdot|\mathcal{H}_t)}\Big[\mathbf{1}\big[\delta^{d-1,v}(t) \le \delta^{a_{\mathrm{obs}}}(t) \wedge \delta^a(t) \wedge T - t\big] \mathbb{E}_P\big[Y|\mathcal{H}_t \cup \widetilde{\mathcal{H}}^{\backslash a_{\mathrm{obs}}}_{(t,t_{d-1}]}\big]\Big] =$$

$$\mathbb{E}_{\widetilde{\mathcal{H}}_{t_{d-1}} \sim \widetilde{P}(\cdot|\mathcal{H}_t)} \mathbb{E}_{\widetilde{\widetilde{\mathcal{H}}} \sim \widetilde{P}\big(\cdot\big|\mathcal{H}_t \cup \widetilde{\mathcal{H}}^{\backslash a_{\mathrm{obs}}}_{(t,t_{d-1}]}\big)}\Big[\mathbf{1}\big[\delta^{d-1,v}_{\widetilde{\mathcal{H}}}(t) \le \delta^{a_{\mathrm{obs}}}_{\widetilde{\mathcal{H}}}(t) \wedge \delta^a_{\widetilde{\mathcal{H}}}(t) \wedge T - t\big] \cdot$$

$$\Big[\Big(\mathbf{1}\big[\delta^a_{\widetilde{\widetilde{\mathcal{H}}}}(t_{d-1}) < \delta^v_{\widetilde{\widetilde{\mathcal{H}}}}(t_{d-1}) \wedge \delta^{a_{\mathrm{obs}}}_{\widetilde{\widetilde{\mathcal{H}}}}(t_{d-1})\big] \cdot$$

$$\mathbb{E}_P\Big[Y|\mathcal{H}_t \cup \widetilde{\mathcal{H}}^{\backslash a_{\mathrm{obs}}}_{(t,t_{d-1}]} \cup \big(t_{d-1} + \delta^a_{\widetilde{\widetilde{\mathcal{H}}}}(t_{d-1}), a\big)\Big]\Big) +$$

$$\left(\mathbf{1}\left[\delta_{\underset{\mathcal{H}}{\approx}}^{a_{\mathrm{obs}}}(t_{d-1}) < \delta_{\underset{\mathcal{H}}{\approx}}^{a}(t_{d-1}) \wedge \delta_{\underset{\mathcal{H}}{\approx}}^{v}(t_{d-1})\right]\cdot\right.$$

$$\left.\mathbb{E}_P\left[Y|\mathcal{H}_{t+\delta_{\underset{\mathcal{H}}{\approx}}^{a_{\mathrm{obs}}}(t_{d-1})} = \mathcal{H}_t \cup \widetilde{\mathcal{H}}_{(t,t_{d-1}]}^{\backslash a_{\mathrm{obs}}}\right]\right)+$$

$$\left(\mathbf{1}\left[\delta_{\underset{\mathcal{H}}{\approx}}^{a_{\mathrm{obs}}}(t_{d-1}) \wedge \delta_{\underset{\mathcal{H}}{\approx}}^{a}(t_{d-1}) \wedge \delta_{\underset{\mathcal{H}}{\approx}}^{v}(t_{d-1}) > T - t_{d-1}\right]\cdot\right.$$

$$\left.\mathbb{E}_P\left[Y|\mathcal{H}_T = \mathcal{H}_t \cup \widetilde{\mathcal{H}}_{(t,t_{d-1}]}^{\backslash a_{\mathrm{obs}}}\right]\right)+$$

$$\left(\mathbf{1}\left[\delta_{\underset{\mathcal{H}}{\approx}}^{v}(t_{d-1}) < \delta_{\underset{\mathcal{H}}{\approx}}^{a}(t_{d-1}) \wedge \delta_{\underset{\mathcal{H}}{\approx}}^{a_{\mathrm{obs}}}(t_{d-1})\right]\cdot\right.$$

$$\left.\left.\left.\mathbb{E}_P\left[Y|\mathcal{H}_t \cup \widetilde{\mathcal{H}}_{(t,t_{d-1}]}^{\backslash a_{\mathrm{obs}}} \cup (t_{d-1} + \delta_{\underset{\mathcal{H}}{\approx}}^{v}(t_{d-1}), v)\right]\right)\right)\right] =$$

$$\mathbb{E}_{\widetilde{\mathcal{H}}\sim\widetilde{P}(\cdot|\mathcal{H}_t)}\left[\mathbf{1}\left[\delta_{\widetilde{\mathcal{H}}}^{d-1,v}(t) \le \delta_{\widetilde{\mathcal{H}}}^{a_{\mathrm{obs}}}(t) \wedge \delta_{\widetilde{\mathcal{H}}}^{a}(t) \wedge T - t\right]\cdot\right.$$

$$\left[\left(\mathbf{1}\left[\delta_{\widetilde{\mathcal{H}}}^{a}(t_{d-1}) < \delta_{\widetilde{\mathcal{H}}}^{v}(t_{d-1}) \wedge \delta_{\widetilde{\mathcal{H}}}^{a_{\mathrm{obs}}}(t_{d-1})\right]\cdot\right.$$

$$\mathbb{E}_P\left[Y|\mathcal{H}_t \cup \widetilde{\mathcal{H}}_{(t,t_{d-1}]}^{\backslash a_{\mathrm{obs}}} \cup (t_{d-1} + \delta_{\widetilde{\mathcal{H}}}^{a}(t_{d-1}), a)\right]\right)+$$

$$\left(\mathbf{1}\left[\delta_{\widetilde{\mathcal{H}}}^{a_{\mathrm{obs}}}(t_{d-1}) < \delta_{\widetilde{\mathcal{H}}}^{a}(t_{d-1}) \wedge \delta_{\widetilde{\mathcal{H}}}^{v}(t_{d-1})\right]\cdot\right.$$

$$\mathbb{E}_P\left[Y|\mathcal{H}_{t+\delta_{\widetilde{\mathcal{H}}}^{a_{\mathrm{obs}}}(t_{d-1})} = \mathcal{H}_t \cup \widetilde{\mathcal{H}}_{(t,t_{d-1}]}^{\backslash a_{\mathrm{obs}}}\right]\right)+$$

$$\left(\mathbf{1}\left[\delta_{\widetilde{\mathcal{H}}}^{a_{\mathrm{obs}}}(t_{d-1}) \wedge \delta_{\widetilde{\mathcal{H}}}^{a}(t_{d-1}) \wedge \delta_{\widetilde{\mathcal{H}}}^{v}(t_{d-1}) > T - t_{d-1}\right]\cdot\right.$$

$$\mathbb{E}_P\left[Y|\mathcal{H}_T = \mathcal{H}_t \cup \widetilde{\mathcal{H}}_{(t,t_{d-1}]}^{\backslash a_{\mathrm{obs}}}\right]\right)+$$

$$\left(\mathbf{1}\left[\delta_{\widetilde{\mathcal{H}}}^{v}(t_{d-1}) < \delta_{\widetilde{\mathcal{H}}}^{a}(t_{d-1}) \wedge \delta_{\widetilde{\mathcal{H}}}^{a_{\mathrm{obs}}}(t_{d-1})\right]\cdot\right.$$

$$\left.\left.\left.\mathbb{E}_P\left[Y|\mathcal{H}_t \cup \widetilde{\mathcal{H}}_{(t,t_{d-1}]}^{\backslash a_{\mathrm{obs}}} \cup (t_{d-1} + \delta_{\widetilde{\mathcal{H}}}^{v}(t_{d-1}), v)\right]\right)\right)\right] \tag{14}$$

Let us simplify the multiples of all the indicators that appear in eq. (14), while dropping the $\widetilde{\mathcal{H}}$ subscripts since they are clear from context.

$$\mathbf{1}\left[\delta_{\widetilde{\mathcal{H}}}^{d-1,v}(t) \le \delta_{\widetilde{\mathcal{H}}}^{a_{\mathrm{obs}}}(t) \wedge \delta_{\widetilde{\mathcal{H}}}^{a}(t) \wedge T - t\right] \cdot \mathbf{1}\left[\delta_{\widetilde{\mathcal{H}}}^{a}(t_{d-1}) < \delta_{\widetilde{\mathcal{H}}}^{a_{\mathrm{obs}}}(t_{d-1}) \wedge \delta_{\widetilde{\mathcal{H}}}^{v}(t_{d-1})\right] =$$
$$\mathbf{1}\left[\delta^{d-1,v}(t) \le \delta^{a}(t) < \delta^{d,v}(t) \wedge \delta^{a_{\mathrm{obs}}}(t) \wedge T - t\right],$$
$$\mathbf{1}\left[\delta_{\widetilde{\mathcal{H}}}^{d-1,v}(t) \le \delta_{\widetilde{\mathcal{H}}}^{a_{\mathrm{obs}}}(t) \wedge \delta_{\widetilde{\mathcal{H}}}^{a}(t) \wedge T - t\right] \cdot \mathbf{1}\left[\delta_{\widetilde{\mathcal{H}}}^{a_{\mathrm{obs}}}(t_{d-1}) < \delta_{\widetilde{\mathcal{H}}}^{a}(t_{d-1}) \wedge \delta_{\widetilde{\mathcal{H}}}^{v}(t_{d-1})\right] =$$
$$\mathbf{1}\left[\delta^{d-1,v}(t) \le \delta^{a_{\mathrm{obs}}}(t) < \delta^{d,v}(t) \wedge \delta^{a}(t) \wedge T - t\right],$$
$$\mathbf{1}\left[\delta_{\widetilde{\mathcal{H}}}^{d-1,v}(t) \le \delta_{\widetilde{\mathcal{H}}}^{a_{\mathrm{obs}}}(t) \wedge \delta_{\widetilde{\mathcal{H}}}^{a}(t) \wedge T - t\right] \cdot \mathbf{1}\left[\delta_{\widetilde{\mathcal{H}}}^{a_{\mathrm{obs}}}(t_{d-1}) \wedge \delta_{\widetilde{\mathcal{H}}}^{a}(t_{d-1}) \wedge \delta_{\widetilde{\mathcal{H}}}^{v}(t_{d-1}) > T - t_{d-1}\right] =$$
$$\mathbf{1}\left[\delta^{d,v}(t) \wedge \delta^{a_{\mathrm{obs}}}(t) \wedge \delta^{a}(t) > T - t\right],$$
$$\mathbf{1}\left[\delta_{\widetilde{\mathcal{H}}}^{d-1,v}(t) \le \delta_{\widetilde{\mathcal{H}}}^{a_{\mathrm{obs}}}(t) \wedge \delta_{\widetilde{\mathcal{H}}}^{a}(t) \wedge T - t\right] \cdot \mathbf{1}\left[\delta_{\widetilde{\mathcal{H}}}^{v}(t_{d-1}) < \delta_{\widetilde{\mathcal{H}}}^{a}(t_{d-1}) \wedge \delta_{\widetilde{\mathcal{H}}}^{a_{\mathrm{obs}}}(t_{d-1})\right] =$$
$$\mathbf{1}\left[\delta^{d,v}(t) < \delta^{a_{\mathrm{obs}}}(t) \wedge \delta^{a}(t) \wedge T - t\right]. \tag{15}$$

Switching these equalities from 15 into eq. (14), we get

$$\mathbb{E}_{\widetilde{\mathcal{H}}_{t_{d-1}}\sim\widetilde{P}(\cdot|\mathcal{H}_t)}\left[\mathbf{1}\left[\delta^{d-1,v}(t) \le \delta^{a_{\mathrm{obs}}}(t) \wedge \delta^{a}(t) \wedge T - t\right]\mathbb{E}_P\left[Y|\mathcal{H}_t \cup \widetilde{\mathcal{H}}_{(t,t_{d-1}]}^{\backslash a_{\mathrm{obs}}}\right]\right] =$$

$$\mathbb{E}_{\widetilde{\mathcal{H}}\sim\widetilde{P}(\cdot|\mathcal{H}_t)}\left[\left(\mathbf{1}\left[\delta^{d-1,v}(t) \le \delta^{a}(t) < \delta^{d,v}(t) \wedge \delta^{a_{\mathrm{obs}}}(t) \wedge T - t\right]\cdot\right.\right.$$

$$\left.\mathbb{E}_P\left[Y|\mathcal{H}_t \cup \widetilde{\mathcal{H}}_{(t,t_{d-1}]}^{\backslash a_{\mathrm{obs}}} \cup (t_{d-1} + \delta_{\widetilde{\mathcal{H}}}^{a}(t_{d-1}), a)\right]\right)+$$

$$\left(\mathbf{1}\big[\delta^{d-1,v}(t) \le \delta^{a_{\mathrm{obs}}}(t) < \delta^{d,v}(t) \wedge \delta^{a}(t) \wedge T - t\big]\cdot\right.$$

$$\mathbb{E}_P\Big[Y|\mathcal{H}_{t+\delta^{a_{\mathrm{obs}}}_{\widetilde{\mathcal{H}}}(t_{d-1})} = \mathcal{H}_t \cup \widetilde{\mathcal{H}}^{\backslash a_{\mathrm{obs}}}_{(t,t_{d-1}]}\Big]\Big) +$$

$$\left(\mathbf{1}\big[\delta^{d,v}(t) \wedge \delta^{a_{\mathrm{obs}}}(t) \wedge \delta^{a}(t) > T - t\big]\cdot\right.$$

$$\mathbb{E}_P\Big[Y|\mathcal{H}_T = \mathcal{H}_t \cup \widetilde{\mathcal{H}}^{\backslash a_{\mathrm{obs}}}_{(t,t_{d-1}]}\Big]\Big) +$$

$$\left(\mathbf{1}\big[\delta^{d,v}(t) < \delta^{a_{\mathrm{obs}}}(t) \wedge \delta^{a}(t) \wedge T - t\big]\cdot\right.$$

$$\left.\left.\left.\mathbb{E}_P\Big[Y|\mathcal{H}_t \cup \widetilde{\mathcal{H}}^{\backslash a_{\mathrm{obs}}}_{(t,t_{d-1}]} \cup (t_{d-1} + \delta^{v}_{\widetilde{\mathcal{H}}}(t_{d-1}), v)\Big]\Big)\right)\right]\right].$$

Then it is easy to deduce that the conditioning sets can be simplified as follows,

$$\mathbb{E}_{\widetilde{\mathcal{H}}_{t_{d-1}} \sim \widetilde{P}(\cdot|\mathcal{H}_t)}\left[\mathbf{1}\big[\delta^{d-1,v}(t) \le \delta^{a_{\mathrm{obs}}}(t) \wedge \delta^{a}(t) \wedge T - t\big]\mathbb{E}_P\big[Y|\mathcal{H}_t \cup \widetilde{\mathcal{H}}^{\backslash a_{\mathrm{obs}}}_{(t,t_{d-1}]}\big]\right] =$$

$$\mathbb{E}_{\widetilde{\mathcal{H}} \sim \widetilde{P}(\cdot|\mathcal{H}_t)}\left[\left(\mathbf{1}\big[\delta^{d-1,v}(t) \le \delta^{a}(t) < \delta^{d,v}(t) \wedge \delta^{a_{\mathrm{obs}}}(t) \wedge T - t\big]\cdot\right.\right.$$

$$\mathbb{E}_P\Big[Y|\mathcal{H}_t \cup \widetilde{\mathcal{H}}^{\backslash a_{\mathrm{obs}}}_{(t,\delta^{a}(t)]}\Big]\Big) +$$

$$\left(\mathbf{1}\big[\delta^{d-1,v}(t) \le \delta^{a_{\mathrm{obs}}}(t) < \delta^{d,v}(t) \wedge \delta^{a}(t) \wedge T - t\big]\cdot\right.$$

$$\mathbb{E}_P\Big[Y|\mathcal{H}_t \cup \widetilde{\mathcal{H}}^{\backslash a_{\mathrm{obs}}}_{(t,t+\delta^{a_{\mathrm{obs}}}(t)]}\Big]\Big) +$$

$$\left(\mathbf{1}\big[\delta^{d,v}(t) \wedge \delta^{a_{\mathrm{obs}}}(t) \wedge \delta^{a}(t) > T - t\big]\cdot\right.$$

$$\mathbb{E}_P\Big[Y|\mathcal{H}_T = \mathcal{H}_t \cup \widetilde{\mathcal{H}}^{\backslash a_{\mathrm{obs}}}_{(t,T]}\Big]\Big) +$$

$$\left(\mathbf{1}\big[\delta^{d,v}(t) < \delta^{a_{\mathrm{obs}}}(t) \wedge \delta^{a}(t) \wedge T - t\big]\cdot\right.$$

$$\left.\left.\mathbb{E}_P\Big[Y|\mathcal{H}_t \cup \widetilde{\mathcal{H}}^{\backslash a_{\mathrm{obs}}}_{(t,t+\delta^{d,v}(t)]}\Big]\Big)\right]\right].$$

Plugging this back into eq. (13) and using the linearity of expectation gives us exactly the equality in eq. (12) and concludes the proof. $\qquad\square$

## B.3   DISCRETE TIME VERSION

For the discrete-time version we keep a similar notation, but take time increments of 1 and call the target policy $\pi$, which takes a history of the process and outputs a distribution over possible treatments. The trajectory $\mathcal{H}$ now simplifies to the form $\{(\mathbf{x}_1, \mathbf{y}_1, \mathbf{a}_1), (\mathbf{x}_2, \mathbf{y}_2, \mathbf{a}_2), \dots, (\mathbf{x}_T, \mathbf{y}_T, \mathbf{a}_T)\}$ and similarly for the history $\mathcal{H}_t$. The analogous claim to theorem 1 for these decision processes follows from the lemma we prove below by setting $d = T - t$.

**Lemma 3.** *For any $\mathcal{H}$, $t \in [0, T)$ and $1 \le d \le T - t$ such that $\mathcal{H}_t$ is measurable w.r.t $P$ we have that*

$$\mathbb{E}_P[Y|\mathcal{H}_t] = \mathbb{E}_{\widetilde{\mathcal{H}} \sim \widetilde{P}(\cdot|\mathcal{H}_t)}\left[\vphantom{\sum_{k=1}^{d}}\right.$$

$$\sum_{k=1}^{d}\left(\mathbf{1}_{a_{t+k} \ne a_{\mathrm{obs},t+k}} \prod_{i=1}^{k-1} \mathbf{1}_{a_{\mathrm{obs},t+k-i} = a_{t+k-i}} \mathbb{E}_P[Y|\mathcal{H}_{t+k} = \mathcal{H}_t \cup \widetilde{\mathcal{H}}^{\backslash a_{\mathrm{obs}}}_{[t+1,t+k]}]\right)$$

$$\left.+ \prod_{k=1}^{d} \mathbf{1}_{a_{t+k} = a_{\mathrm{obs},t+k}} \mathbb{E}_P[Y|\mathcal{H}_{t+d} = \mathcal{H}_t \cup \widetilde{\mathcal{H}}^{\backslash a_{\mathrm{obs}}}_{[t+1,t+d]}]\right]. \tag{16}$$

*Note that for $k = 1$, we define $\prod_{i=1}^{k-1} \mathbf{1}_{a_{t+k-i} = a_{\mathrm{obs},t+k-i}} = 1$.*

*Proof.* Throughout the proof we will use conditioning on $\widetilde{\mathcal{H}}_{t+k}$ (where $0 \leq k \leq d$) as a shorthand for conditioning on $\mathcal{H}_{t+k} = \mathcal{H}_t \cup \widetilde{\mathcal{H}}_{[t+1,t+k]}^{\backslash a_{\mathrm{obs}}}$, as the meaning is clear from context. We will prove the claim by induction on $d$. The base case for $d = 1$ follows simply from observing that the value drawn for $a_{\mathrm{obs},t+1}$ does not change the items inside the expectation $\mathbb{E}_{\widetilde{\mathcal{H}} \sim \widetilde{P}(\cdot | \mathcal{H}_t)}[\ldots]$. Let us write this down in detail.

$$
\mathbb{E}_{\widetilde{\mathcal{H}} \sim \widetilde{P}(\cdot | \mathcal{H}_t)}\left[ \mathbf{1}_{a_{t+1} \neq a_{\mathrm{obs},t+1}} \mathbb{E}_P[Y | \widetilde{\mathcal{H}}_{t+1}] + \mathbf{1}_{a_{t+1} = a_{\mathrm{obs},t+1}} \mathbb{E}_P[Y | \widetilde{\mathcal{H}}_{t+1}] \right] =
$$

$$
\mathbb{E}_{\widetilde{\mathcal{H}} \sim \widetilde{P}(\cdot | \mathcal{H}_t)}\left[ \mathbb{E}_P[Y | \widetilde{\mathcal{H}}_{t+1}] \right] =
$$

$$
\mathbb{E}_{\widetilde{\mathcal{H}}_{t+1} \sim P_{\mathrm{obs}}(\cdot | \mathcal{H}_t)}\left[ \mathbb{E}_P[Y | \widetilde{\mathcal{H}}_{t+1}] \right] =
$$

$$
\mathbb{E}_{\mathbf{x}_{t+1}, \mathbf{y}_{t+1} \sim P_{\mathrm{obs}}(\cdot | \mathcal{H}_t), a_{t+1} \sim \pi(\cdot | \mathcal{H}_t, \mathbf{x}_{t+1}, \mathbf{y}_{t+1})} \Bigg[
$$
$$
\mathbb{E}_{a_{\mathrm{obs},t+1} \sim \pi_{\mathrm{obs}}(\cdot | \mathcal{H}_t, \mathbf{x}_{t+1}, \mathbf{y}_{t+1})}\left[ \mathbb{E}_P[Y | \widetilde{\mathcal{H}}_{t+1}] \right] \Bigg] =
$$

$$
\mathbb{E}_{\mathbf{x}_{t+1}, \mathbf{y}_{t+1} \sim P_{\mathrm{obs}}(\cdot | \mathcal{H}_t), a_{t+1} \sim \pi(\cdot | \mathcal{H}_t, \mathbf{x}_{t+1}, \mathbf{y}_{t+1})}\left[ \mathbb{E}_P[Y | \widetilde{\mathcal{H}}_{t+1}] \right] =
$$

$$
\mathbb{E}_{\mathbf{x}_{t+1}, \mathbf{y}_{t+1} \sim P(\cdot | \mathcal{H}_t), a_{t+1} \sim \pi(\cdot | \mathcal{H}_t, \mathbf{x}_{t+1}, \mathbf{y}_{t+1})}\left[ \mathbb{E}_P[Y | \widetilde{\mathcal{H}}_{t+1}] \right] =
$$

$$
\mathbb{E}_P[Y | \mathcal{H}_t]. \tag{17}
$$

The first equality, as we argued earlier is due to the conditioning set $\widetilde{\mathcal{H}}_{t+1}$ not depending on $a_{\mathrm{obs},t+1}$ (as we defined earlier, it refers to $\mathcal{H}_t \cup \widetilde{\mathcal{H}}_{[t+1,t+1]}^{\backslash a_{\mathrm{obs}}}$). Intuitively, this is true because we only expand the expectation one step forward in time. In the identity we wish to prove, eq. (16), the $a_{\mathrm{obs}}$ treatments do change the item within $\mathbb{E}_{\widetilde{\mathcal{H}} \sim \widetilde{P}(\cdot | \mathcal{H}_t)}[\ldots]$ since they determine the earliest disagreement time. The second equality is obtained by marginalizing over the sampled trajectories after time $t+1$, as they are also not included in $\widetilde{\mathcal{H}}_{t+1}$. Then the third equality writes the sampling of $\mathbf{y}_{t+1}, \mathbf{x}_{t+1}, a_{t+1}, a_{\mathrm{obs},t+1}$ explicitly, and the fourth marginalizes over $a_{\mathrm{obs},t+1}$ as we already mentioned it does not appear in the expectation. Then we use the equality $P(X_{t+1}, Y_{t+1} | \mathcal{H}_t) = P_{\mathrm{obs}}(X_{t+1}, Y_{t+1} | \mathcal{H}_t)$, to write the expectation as sampling $\mathbf{x}_{t+1}, \mathbf{y}_{t+1}, a_{t+1}$ from $P$. Finally, we use the tower property of conditional expectation and arrive at the desired expression. Next, assume that the claim holds for some $d-1$. We write down the induction hypothesis and then marginalize over all the event after time $t + d - 1$ in $\widetilde{\mathcal{H}}$, as they do not appear in the arguments of the expectation.

$$
\mathbb{E}_P[Y | \mathcal{H}_t] = \mathbb{E}_{\widetilde{\mathcal{H}} \sim \widetilde{P}(\cdot | \mathcal{H}_t)}\left[ \sum_{k=1}^{d-1} \left( \mathbf{1}_{a_{t+k} \neq a_{\mathrm{obs},t+k}} \prod_{i=1}^{k-1} \mathbf{1}_{a_{t+k-i} = a_{\mathrm{obs},t+k-i}} \cdot \mathbb{E}_P[Y | \widetilde{\mathcal{H}}_{t+k}] \right) \right.
$$
$$
\left. + \prod_{k=1}^{d-1} \mathbf{1}_{a_{t+k} = a_{\mathrm{obs},t+k}} \cdot \mathbb{E}_P[Y | \widetilde{\mathcal{H}}_{t+d-1}] \right]
$$
$$
= \mathbb{E}_{\widetilde{\mathcal{H}}_{t+d-1} \sim \widetilde{P}(\cdot | \mathcal{H}_t)}\left[ \sum_{k=1}^{d-1} \left( \mathbf{1}_{a_{t+k} \neq a_{\mathrm{obs},t+k}} \prod_{i=1}^{k-1} \mathbf{1}_{a_{t+k-i} = a_{\mathrm{obs},t+k-i}} \cdot \mathbb{E}_P[Y | \widetilde{\mathcal{H}}_{t+k}] \right) \right.
$$
$$
\left. + \prod_{k=1}^{d-1} \mathbf{1}_{a_{t+k} = a_{\mathrm{obs},t+k}} \cdot \mathbb{E}_P[Y | \widetilde{\mathcal{H}}_{t+d-1}] \right]. \tag{18}
$$

As in the base case, we can expand the expectation $\mathbb{E}[Y | \widetilde{\mathcal{H}}_{t+d-1}]$ one time step forward,

$$
\mathbb{E}_P[Y | \widetilde{\mathcal{H}}_{t+d-1}] = \mathbb{E}_{\widetilde{\mathcal{H}}_{t+d} \sim \widetilde{P}(\cdot | \widetilde{\mathcal{H}}_{t+d-1})}\left[ \mathbb{E}_P\left[Y | \widetilde{\mathcal{H}}_{t+d}\right] \cdot \mathbf{1}_{a_{t+d} \neq a_{\mathrm{obs},t+d}} \right. \tag{19}
$$

$$+\mathbb{E}_P\left[Y|\widetilde{\mathcal{H}}_{t+d}\right]\cdot\mathbf{1}_{a_{t+d}=a_{\text{obs},t+d}}\right].$$

Now plug this in eq. (18) to obtain

$$\mathbb{E}_P[Y|\mathcal{H}_t]=\mathbb{E}_{\widetilde{\mathcal{H}}_{t+d-1}\sim\widetilde{P}(\cdot|\mathcal{H}_t)}\left[\sum_{k=1}^{d-1}\left(\mathbf{1}_{a_{t+k}\neq a_{\text{obs},t+k}}\prod_{i=1}^{k-1}\mathbf{1}_{a_{t+k-i}=a_{\text{obs},t+k-i}}\cdot\mathbb{E}_P[Y|\widetilde{\mathcal{H}}_{t+k}]\right)\right.$$

$$+\prod_{k=1}^{d-1}\mathbf{1}_{a_{t+k}=a_{\text{obs},t+k}}\cdot\left(\mathbb{E}_{\widetilde{\mathcal{H}}_{t+d}\sim\widetilde{P}(\cdot|\widetilde{\mathcal{H}}_{t+d-1})}\left[\mathbb{E}_P\left[Y|\widetilde{\mathcal{H}}_{t+d}\right]\cdot\mathbf{1}_{a_{t+d}\neq a_{\text{obs},t+d}}\right.\right.$$

$$\left.\left.\left.+\mathbb{E}_P\left[Y|\widetilde{\mathcal{H}}_{t+d}\right]\cdot\mathbf{1}_{a_{t+d}=a_{\text{obs},t+d}}\right]\right)\right]$$

$$=\mathbb{E}_{\widetilde{\mathcal{H}}_{t+d-1}\sim\widetilde{P}(\cdot|\mathcal{H}_t)}\left[\mathbb{E}_{\widetilde{\mathcal{H}}_{t+d}\sim\widetilde{P}(\cdot|\widetilde{\mathcal{H}}_{t+d-1})}\left[\right.\right.$$

$$\sum_{k=1}^{d-1}\left(\mathbf{1}_{a_{t+k}\neq a_{\text{obs},t+k}}\prod_{i=1}^{k-1}\mathbf{1}_{a_{t+k-i}=a_{\text{obs},t+k-i}}\cdot\mathbb{E}_P[Y|\widetilde{\mathcal{H}}_{t+k}]\right)$$

$$+\prod_{k=1}^{d-1}\mathbf{1}_{a_{t+k}=a_{\text{obs},t+k}}\cdot\left(\mathbb{E}_P\left[Y|\widetilde{\mathcal{H}}_{t+d}\right]\cdot\mathbf{1}_{a_{t+d}\neq a_{\text{obs},t+d}}\right.$$

$$\left.\left.\left.+\mathbb{E}_P\left[Y|\widetilde{\mathcal{H}}_{t+d}\right]\cdot\mathbf{1}_{a_{t+d}=a_{\text{obs},t+d}}\right)\right]\right]$$

$$=\mathbb{E}_{\widetilde{\mathcal{H}}_{t+d}\sim\widetilde{P}(\cdot|\mathcal{H}_t)}\left[\sum_{k=1}^{d-1}\left(\mathbf{1}_{a_{t+k}\neq a_{\text{obs},t+k}}\prod_{i=1}^{k-1}\mathbf{1}_{a_{t+k-i}=a_{\text{obs},t+k-i}}\cdot\mathbb{E}_P[Y|\widetilde{\mathcal{H}}_{t+k}]\right)\right.$$

$$+\prod_{k=1}^{d-1}\mathbf{1}_{a_{t+k}=a_{\text{obs},t+k}}\cdot\left(\mathbb{E}_P\left[Y|\widetilde{\mathcal{H}}_{t+d}\right]\cdot\mathbf{1}_{a_{t+d}\neq a_{\text{obs},t+d}}\right.$$

$$\left.\left.+\mathbb{E}_P\left[Y|\widetilde{\mathcal{H}}_{t+d}\right]\cdot\mathbf{1}_{a_{t+d}=a_{\text{obs},t+d}}\right)\right]$$

$$=\mathbb{E}_{\widetilde{\mathcal{H}}_{t+d}\sim\widetilde{P}(\cdot|\mathcal{H}_t)}\left[\sum_{k=1}^{d}\left(\mathbf{1}_{a_{t+k}\neq a_{\text{obs},t+k}}\prod_{i=1}^{k-1}\mathbf{1}_{a_{t+k-i}=a_{\text{obs},t+k-i}}\cdot\mathbb{E}_P[Y|\widetilde{\mathcal{H}}_{t+k}]\right)\right.$$

$$\left.+\prod_{k=1}^{d}\mathbf{1}_{a_{t+k}=a_{\text{obs},t+k}}\cdot\mathbb{E}_P\left[Y|\widetilde{\mathcal{H}}_{t+d}\right]\right)\right]$$

$$=\mathbb{E}_{\widetilde{\mathcal{H}}\sim\widetilde{P}(\cdot|\mathcal{H}_t)}\left[\sum_{k=1}^{d}\left(\mathbf{1}_{a_{t+k}\neq a_{\text{obs},t+k}}\prod_{i=1}^{k-1}\mathbf{1}_{a_{t+k-i}=a_{\text{obs},t+k-i}}\cdot\mathbb{E}_P[Y|\widetilde{\mathcal{H}}_{t+k}]\right)\right.$$

$$\left.+\prod_{k=1}^{d}\mathbf{1}_{a_{t+k}=a_{\text{obs},t+k}}\cdot\mathbb{E}_P\left[Y|\widetilde{\mathcal{H}}_{t+d}\right]\right)\right]$$

The equality between the first and last item is exactly the expression we wish to prove. The first transition plugs eq. (19) into eq. (18), the second pushes the expectation over $\mathbf{x}_{t+d}, \mathbf{y}_{t+d}, a_{t+d}, a_{\text{obs},t+d}$ outside, the third uses the law of total expectation. Then we rearrange

$$\prod_{k=1}^{d-1}\mathbf{1}_{a_{t+k}=a_{\text{obs},t+k}}\cdot\left(\mathbb{E}_P\left[Y|\widetilde{\mathcal{H}}_{t+d}\right]\cdot\mathbf{1}_{a_{t+d}\neq a_{\text{obs},t+d}}+\mathbb{E}_P\left[Y|\widetilde{\mathcal{H}}_{t+d}\right]\cdot\mathbf{1}_{a_{t+d}=a_{\text{obs},t+d}}\right)$$

$$=\left(\mathbf{1}_{a_{t+d}\neq a_{\text{obs},t+d}}\cdot\prod_{k=1}^{d-1}\mathbf{1}_{a_{t+k}=a_{\text{obs},t+k}}\cdot\mathbb{E}_P\left[Y|\widetilde{\mathcal{H}}_{t+d}\right]\right)+\left(\prod_{k=1}^{d}\mathbf{1}_{a_{t+k}=a_{\text{obs},t+k}}\cdot\mathbb{E}_P\left[Y|\widetilde{\mathcal{H}}_{t+d}\right]\right),$$

and push the first item into the summation over $k$. $\qquad\square$

To obtain the identity with the earliest disagreement time, set $d = T - t$ in the statement we proved, and note that for any value of $\widetilde{\mathcal{H}}$ in the expectation above, exactly one of the following items equals 1,

$$\mathbf{1}_{a_{t+k} \neq a_{\text{obs}, t+k}} \cdot \prod_{i=1}^{k-1} \mathbf{1}_{a_{t+k-i} = a_{\text{obs}, t+k-i}} \text{ for } k \in [T-t], \text{ and } \prod_{k=1}^{d} \mathbf{1}_{a_{t+k} = a_{\text{obs}, t+k}},$$

and the value of $k$ for which the above item equals 1 is $k = \delta_{\widetilde{\mathcal{H}}}(t)$ (and $T$ when $\prod_{k=1}^{d} \mathbf{1}_{a_{t+k} = a_{\text{obs}, t+k}} = 1$), hence $t + \delta_{\widetilde{\mathcal{H}}}(t)$ is the earliest disagreement time, and eq. (16) reduces to eq. (2) as claimed.

## C  ADDITIONAL DISCUSSION ON RELATED WORK

As outlined in section 4, several techniques have been proposed for scalable estimation of causal effects in sequential decision-making, with more limited development in the case of irregular observation times. One set of approaches (Bica et al., 2020; Lim, 2018; Melnychuk et al., 2022), that only apply to discrete time processes and static policies, can be roughly characterized as follows. A prediction model $f(\mathcal{H}_t, \mathcal{H}_T^a; \boldsymbol{\theta})$ for the outcome $Y$ is learned, where $\mathcal{H}_t$ is the observed history of events and $\mathcal{H}_T^a$ is the set of future treatments we would like to reason about. That is, in our notation we would like $f(\mathcal{H}_t, \mathcal{H}_T^a; \boldsymbol{\theta})$ to estimate $\mathbb{E}_{P_{\mathcal{H}_T^a}}[Y|\mathcal{H}_t]$, where $P_{\mathcal{H}_T^a}$ assigns the treatments in $\mathcal{H}_T^a$ w.p. 1. In potential outcomes notation, this corresponds to $\mathbb{E}[Y^{\mathbf{a}}|\mathcal{H}_t]$, where $Y^{\mathbf{a}}$ is a random variable that outputs the outcome under a set of static future treatments $\mathbf{a}$. All methods involve learning a representation of history $\mathbf{Z}_t = \phi(\mathcal{H}_t; \eta)$, and combine two important elements for achieving correct estimates.

1. To yield correct causal estimates under an observational distribution that is not sequentially randomized, methods either estimate products of propensity weights (Lim, 2018), or add a loss to make $\mathbf{Z}_t$ non-predictive of the treatment $A_t$, $\phi$ is then called a balancing representation.

2. To facilitate prediction of $Y$ under a set of future treatments in the interval $(t, T]$, either $\phi$ is taken as a sequence model, or a separate "decoder" network is learned (Bica et al., 2020; Lim, 2018). A sequence model is trained with inputs where $\mathcal{H}_{i,T}^{x,y} \setminus \mathcal{H}_{i,t}^{x,y}$, i.e. the covariates in a projection interval $(t, T]$ are masked, while the decoder takes $\mathbf{Z}_t$ and $\mathcal{H}_T^a$ as inputs. Both are trained to predict the outcome $Y$ and serve as an estimator for $\mathbb{E}_{P_{\text{obs}}}[Y|\mathbf{Z}_t, \mathcal{H}_T^a]$, which recovers the correct causal effect under sequential exchangeability. Notice that these techniques preclude estimation with dynamic treatments, i.e. policies.

For irregular sampling, Seedat et al. (2022) follow the same recipe but choose a neural CDE architecture. This interpolates the latent path $\mathbf{Z}_t$ in intervals between jump times of the processes, and is shown empirically to be more suitable when working with data that is subsampled from a complete trajectory of features in continuous time. The solution is not equipped to estimate interventions on continuous treatment times (in our notation, $\lambda^a$). As mentioned earlier, Vanderschueren et al. (2023) handle informative sampling times with inverse weighting based on the intensity $\lambda$. However, this is a different problem setting from ours, as they do not seek to intervene on sampling times but wish to solve a case where outcomes, features and treatments always jump simultaneously. In our setting, intervening on $\lambda^a$ with such simultaneous jumps would result in $\lambda_{\text{obs}}^{x,y} \neq \lambda^{x,y}$, which is not the focus of our work. Finally, we also note the required assumption for causal validity that is claimed in these works is roughly $P_{\text{obs}}(A_t = a_t|\mathcal{H}_T) = P_{\text{obs}}(A_t = a_t|\mathcal{H}_t)$. The assumption is unreasonable since $\mathcal{H}_T$ includes future *factual* outcomes that depend on the taken action, instead of the more standard exchangeability assumption that posits independence of *potential* outcomes.

The G-estimation solution of (Li et al., 2021) for discrete time decision processes fits models for both $\pi_{\text{obs}}(A(t)|\mathcal{H}_{t-1}, X(t))$, and $P_{\text{obs}}(X(t), Y(t)|\mathcal{H}_{t-1})$. Then at inference time, they replace $\pi_{\text{obs}}$ with the desired policy $\pi$ and estimate trajectories or conditional expectations of $Y$ with monte-carlo simulations. A straightforward generalization of this approach to decision point processes can be devised by fitting the intensities $\lambda_{\text{obs}}$ and replacing $\lambda^a$ for inference. While we believe that this is an interesting direction for future work, we do not pursue it further in our experiments, since developing architectures and methods for learning generative models under irregular sampling deserves a dedicated and in-depth exploration.

## D    ADDITIONAL DISCUSSION ON IDENTIFIABILITY

To make the discussion on identifiability from section 2.2 more concise, we refer here to terms and results from Røysland et al. (2022), and explain how they apply to our setting. Then we prove corollary 1

### D.1    CAUSAL IDENTIFICATION TERMINOLOGY

**Filtrations and restrictions.**    In our notation, we condition on past events $\mathcal{H}_t$, while in most formal treatments of stochastic processes conditioning is performed on a filtration $\mathcal{F}_t$ of a $\sigma$-algbera $\mathcal{F}$. $\mathcal{F}$ is the full set of information generated by the processes $N = [N^a, N^x, N^y, N^u]$. The notation $\mathcal{F}^v$ is used for the $\sigma$-algebra generated by the process $N^v$ alone, hence to a reduced set of information with regards to $\mathcal{F}$. Note that here we use the index $v$ instead of $e$ which we used in the main paper. This is to avoid confusion with the notation for edges in a graph $G$. In the notation of Røysland et al. (2022), there are also corresponding filtrations $\mathcal{F}_t^v$ of $\mathcal{F}^v$. Finally the restriction $P|_{\mathcal{F}^v}$ is used to denote the restriction of probability measure $P$ to the sub $\sigma$-algebra $\mathcal{F}^v$. Intuitively, $P|_{\mathcal{F}^{x,a,y}}$ is the measure that ignores all the information generated by the unobserved processes $N^u$ by marginalizing them. Outside this section, appendix D, the notation $P$ is used to refer directly to $P|_{\mathcal{F}^{x,a,y}}$ as we work under the ignorability assumption, which means inference under the restricted probability distribution yields valid causal effects. In the rest of this section we explain why this validity holds, hence $P$ refers to the full process with the unobserved information included.

Following the above discussion on filtrations, expressions such as $\lambda^v(t|\mathcal{H}_t)$ should be read as conditioning on a subset of the sample space where the events in the time interval $[0, t)$ coincide with $\mathcal{H}_t$.

**Interventions and causal validity.**    Under the assumption on independent increments, item 2 in Assumption 1, we have that densities for a trajectory $\mathcal{H}$ take on the form:

$$P_{\text{obs}}(\mathcal{H}) = \exp\left(-\int_0^T \lambda_{\text{obs},\bullet}(t|\mathcal{H}_t)\mathrm{d}t\right) \prod_{z \in \{a,x,y\}, t_{z,i} \in \mathcal{H}^z} \lambda_{\text{obs}}^z(t_{z,i}|\mathcal{H}_{t_{z,i}}).$$

Interventions on $N^a$ then mean we replace the intensity $\lambda_{\text{obs}}^a$ (which may depend on $\mathcal{H}^u$) with an intensity $\lambda^a$ that only depends on $\mathcal{H}^{x,a,y}$ (formally, this means it is $\mathcal{F}^{x,a,y}$ predictable). The densities change accordingly

The question of causal validity is then whether calculating statistics under $P$, e.g. $E_P[Y|\mathcal{H}_t^{a,x,y}]$, results in the same estimation as calculating them under $P|_{\mathcal{F}^{a,x,y}}$. The difference being that $P|_{\mathcal{F}^{a,x,y}}$ intervenes on $P_{\text{obs}}|_{\mathcal{F}^{a,x,y}}$ instead of on $P_{\text{obs}}$.[8] The condition of eliminability, along with overlap, ensures that this validity holds.

**Eliminability, local independence, and the assumed graph.**    We will local independence graph $G$ is a directed graph where each node We now recall the definition of eliminability and the result we use from Røysland et al. (2022).

$P|_{\mathcal{F}^{x,y,a}}$ is the distribution that performs the intervention on $P_{\text{obs}}$ and then restricts information to the observable information, which admits the causal effect we wish to estimate. We then denote $\tilde{P}$ as the distribution obtained by the same intervention on $P_{\text{obs}}|_{\mathcal{F}^{x,y,a}}$ (i.e. where we marginalize over $U$ before the intervention). We rephrase the theorem of Røysland et al. (2022) to make it more compatible and specialized to our notation and case, it is strongly advised to review the results in their paper for a full formal treatment.

**Theorem 1** (special case of Røysland et al. (2022)). *Let $P(G)$ be a local independence model and the nodes be partitioned as in definition 3. Consider $P_{\text{obs}} \in \mathcal{P}(G)$ and a distribution $P$ obtained by an intervention on the process $N^a$ in $P_{\text{obs}}$, replacing its $\mathcal{F}$-intensity $\lambda_{\text{obs}}^a$ by a $\mathcal{F}^{x,y}$-intensity $\lambda$. Further consider $\tilde{P}$ that is obtained by performing the intervention on $P_{\text{obs}}|_{\mathcal{F}^{x,a,y}}$.*

*If $G$ satisfies eliminability, then the intensities $\lambda$ of $P|_{F^{a,x,y}}$ coincide with the intensities of $\tilde{P}$.*

---

[8]Røysland et al. (2022) also mention the assumption that $P$ itself is causally valid, i.e. that calculating effects under the distribution with the unobserved variables indeed yields valid causal effects. Here we take this assumption as a given.

To finish this overview, we explain in detail why the graph we assume in this work satisfies eliminability, and hence under the additional assumption of overlap, i.e. Assumption 2, we conclude the correctness of our estimation technique.

**Definition 5.** *A trail from $v_0$ to $v_m$ in $G = (V, E)$ is a unique set vertices $\{v_0, \ldots, v_m\} \subseteq V$ and edges $\{e_1, \ldots, e_m\} \subseteq E$ such that either $e_j = v_{j-1} \to v_j$ or $e_j = v_{j-1} \leftarrow v_j$ for every $j = 1, \ldots, m$.*

*The trail is* blocked *by $C \subseteq V$ if either (i) $V$ contains a vertex on the trail that is not a collieder (i.e. there is no $j$ such that both $e_j = v_{j-1} \to v_j$ and $e_{j+1} = v_j \leftarrow v_j$), or (ii) $v_j$ is a collider for some $j \in [m]$, while $v_j \notin C$ and $v \notin C$ for any $v$ which is a descendant of $v$.*

*The trail is* allowed *if $e_m = v_{m-1} \to v_m$. We say that $A \subseteq V$ is $\delta$ separated from $u$ by $C$ if for any $a \in A$, $\{u\} \cup C$ blocks all allowed trails from $a$ to $u$.*

Didelez (2008) gives results that tie $\delta$-separation to *local independence*, namely that under some regularity conditions if $A$ is $\delta$-separated from $v$ by $C$, then $v$ is locally independent of $A$ given $C \cup v$. Local independence in turn is a condition on the intensities of the process, namely that the $\mathcal{F}^{\{v\} \cup C \cup A}$ intensity of $N^v$ is indistinguishable from its $\mathcal{F}^{\{v\} \cup C}$-intensity. Intuitively, the $\mathcal{F}^{\{v\} \cup C}$-intensity is the intensity that does not include information from the past of $N^A$. Formally, it may be obtained using the innovation theorem (Andersen et al., 2012, II.4.2). Then to show that the graph we assumed in our derivation satisfies eliminability and conclude our claims, we will explain why the appropriate $\delta$-separation properties hold.

**Claim 1.** *The graph $G$ in fig. 2 satisfies eliminability.*

*Proof.* Consider $U_1$, it is easy to verify that $N^a$ is locally independent of $U_1$ given $N^x, N^y$ (the second option from definition 3). This is because $N^x, N^y$ blocks all directed paths between $U_1$ and $N^a$ on the graph, and in paths where one of these nodes is a collider, the other is not. Next we consider $U_2$, and claim that $(N^y, N^x)$ are locally independent of $U_2$ given $(N^x, N^y, N^a)$, which means the first bullet in definition 3 is satisfied. To see this local independence holds, consider any allowed path in $G$ from $U_2$ to $N^x$ or $N^y$. Since allowed trails must end with incoming edges to the last node, then either $N^a$ must be a non-collider on such an allowed trail (e.g. in the trail $U_2 \to N^a \to N^x$), or either $N^x$ or $N^y$ must be a non-collider (e.g. in the trail $U_2 \to N^a \leftarrow N^y \to N^x$). In both cases, these non-colliders are in the conditioning set, and thus $U_2$ is $\delta$-separated from $N^x, N^y$ by $(N^a)$. The required local-independence follows from this.[9] $\square$

### D.2 PROOF OF COROLLARY 1

Equipped with the proper definitions of identifiability, we can now conclude our discussion on EDQ. Let us recall corollary 1.

**Corollary 1.** *Under assumption 1, a Q-function satisfying eq. (3) yields the causal effect of the intervention that replaces $(\lambda^a_{\mathrm{obs}}, \pi_{\mathrm{obs}})$ with $(\lambda^a, \pi)$.*

Following our discussion on identifiability, we see that assumption 1 guarantees that calculating conditional expectations w.r.t a distribution where we replace $\lambda^a_{\mathrm{obs}}$ by $\lambda^a$ in $P_{\mathrm{obs}}|_{\mathcal{F}^{x,a,y}}$, yields a correct causal effect. For the rest of this section we will call the resulting distribution $P$. This is a slight abuse of notation, since $P$ was used to refer to the interventional distribution under on $P_{\mathrm{obs}}$, but we use it here to avoid clutter.

To finish the proof of this corollary, we need to show that any $Q$-function that satisfies the self-consistency equation in eq. (3) (which is what algorithm 2 seeks to produce) must also satisfy $Q(\mathcal{H}_t) = \mathbb{E}_P[Y_{>t}|\mathcal{H}_t]$ for every measurable $\mathcal{H}_t$. We first derive the self-consistency condition from eq. (2). Then we show that the only function that satisfies this condition is the conditional expectation we wish to estimate, $\mathbb{E}_P[Y_{>t}|\mathcal{H}_t]$. Equation (2) is rewritten below,

$$\mathbb{E}_P\left[Y|\mathcal{H}_t\right] = \mathbb{E}_{\widetilde{\mathcal{H}} \sim \widetilde{P}(\cdot|\mathcal{H}_t)}\left[\mathbb{E}_P\left[Y \mid \mathcal{H}_{t+\delta_{\widetilde{\mathcal{H}}}(t)} = \mathcal{H}_t \cup \widetilde{\mathcal{H}}^{\backslash a_{\mathrm{obs}}}_{(t,t+\delta_{\widetilde{\mathcal{H}}}(t)]}\right]\right].$$

---

[9] the condition also includes the independence of $U_{>2}$, but in our case $U_{>2} = \emptyset$.

We write $Y = \sum_k Y_k$ as the sum of rewards that have been observed in the trajectory and a random variable that is the sum of future rewards.

$$\mathbb{E}_P[Y|\mathcal{H}_t] = \mathbb{E}_{\widetilde{\mathcal{H}}\sim\widetilde{P}(\cdot|\mathcal{H}_t)}\left[\mathbb{E}_P\left[Y_{>t+\delta_{\widetilde{\mathcal{H}}}(t)} + \sum_{(t_k,y_k)\in\mathcal{H}_{t+\delta_{\widetilde{\mathcal{H}}}(t)}^y} y_k \ \Big|\ \mathcal{H}_{t+\delta_{\widetilde{\mathcal{H}}}(t)} = \mathcal{H}_t \cup \widetilde{\mathcal{H}}_{(t,t+\delta_{\widetilde{\mathcal{H}}}(t)]}^{\backslash a_{\mathrm{obs}}}\right]\right]$$

Then we subtract the rewards until time $t$, $\sum_{(t_k,y_k)\in\mathcal{H}_t^y} y_k$ from both sides of the equation.

$$\mathbb{E}_P[Y|\mathcal{H}_t] - \sum_{(t_k,y_k)\in\mathcal{H}_t^y} y_k = \mathbb{E}_{\widetilde{\mathcal{H}}|\mathcal{H}_t}\left[\sum_{\substack{(t_k,y_k)\in\widetilde{\mathcal{H}}^y:\\ t_k\in(t,t+\delta_{\widetilde{\mathcal{H}}}(t)]}} y_k + \mathbb{E}_P\left[Y_{>t+\delta_{\widetilde{\mathcal{H}}}(t)}\Big|\mathcal{H}_{t+\delta_{\widetilde{\mathcal{H}}}(t)} = \mathcal{H}_t \cup \widetilde{\mathcal{H}}_{(t,t+\delta_{\widetilde{\mathcal{H}}}(t)]}^{\backslash a_{\mathrm{obs}}}\right]\right]$$

This is exactly eq. (3), which we can simplify slightly to

$$\mathbb{E}_P[Y_{>t}|\mathcal{H}_t] = \mathbb{E}_{\widetilde{\mathcal{H}}|\mathcal{H}_t}\left[\sum_{\substack{(t_k,y_k)\in\widetilde{\mathcal{H}}^y:\\ t_k\in(t,t+\delta_{\widetilde{\mathcal{H}}}(t)]}} y_k + \mathbb{E}_P\left[Y_{>t+\delta_{\widetilde{\mathcal{H}}}(t)}\Big|\mathcal{H}_{t+\delta_{\widetilde{\mathcal{H}}}(t)} = \mathcal{H}_t \cup \widetilde{\mathcal{H}}_{(t,t+\delta_{\widetilde{\mathcal{H}}}(t)]}^{\backslash a_{\mathrm{obs}}}\right]\right]$$

Next we show that if there is a unique function $Q(\cdot)$ that satisfies both: (1) $Q(\mathcal{H}_T) = \mathbb{E}_P[Y|\mathcal{H}_T]$, i.e. the estimator returns the correct outcome when it is given a full trajectory (notice the outcome is deterministic in this case), and (2) $Q(\cdot)$ satisfies the recursion in the above equation, namely

$$Q(\mathcal{H}_t) = \mathbb{E}_{\widetilde{\mathcal{H}}|\mathcal{H}_t}\left[\sum_{\substack{(t_k,y_k)\in\widetilde{\mathcal{H}}^y:\\ t_k\in(t,t+\delta_{\widetilde{\mathcal{H}}}(t)]}} y_k + Q\big(\mathcal{H}_t \cup \widetilde{\mathcal{H}}_{(t,t+\delta_{\widetilde{\mathcal{H}}}(t)]}^{\backslash a_{\mathrm{obs}}}\big)\right]. \tag{20}$$

Due to this uniqueness, we will gather that $Q(\mathcal{H}_t)$ must coincide with $\mathbb{E}_P[Y_{>t}|\mathcal{H}_t]$ for all measurable $\mathcal{H}_t$ and conclude the proof.

**Lemma 4.** *Assume $Q_1, Q_2$ are functions that satisfy both eq. (20) and $Q_1(\widetilde{\mathcal{H}}_T) = Q_2(\widetilde{\mathcal{H}}_T) = \mathbb{E}[Y|\widetilde{\mathcal{H}}_T]$ for all $\mathcal{H}_T$. Then $Q_1(\mathcal{H}_t) = Q_2(\mathcal{H}_t)$ for all measurable $\mathcal{H}_t$.*

*Proof.* Since both $Q_1, Q_2$ satisfy eq. (20), it holds that

$$Q_1(\mathcal{H}_t) - Q_2(\mathcal{H}_t) = \mathbb{E}_{\widetilde{\mathcal{H}}|\mathcal{H}_t}[Q_1(\mathcal{H}_t \cup \widetilde{\mathcal{H}}_{(t,t+\delta_{\widetilde{\mathcal{H}}}(t)]}^{\backslash a_{\mathrm{obs}}}) - Q_2(\mathcal{H}_t \cup \widetilde{\mathcal{H}}_{(t,t+\delta_{\widetilde{\mathcal{H}}}(t)]}^{\backslash a_{\mathrm{obs}}})]$$

Applying eq. (20) repeatedly to $Q_1(\mathcal{H}_t \cup \widetilde{\mathcal{H}}_{(t,t+\delta_{\widetilde{\mathcal{H}}}(t)]}^{\backslash a_{\mathrm{obs}}}) - Q_2(\mathcal{H}_t \cup \widetilde{\mathcal{H}}_{(t,t+\delta_{\widetilde{\mathcal{H}}}(t)]}^{\backslash a_{\mathrm{obs}}})$ and so on, for say $d$ times, we get that

$$Q_1(\mathcal{H}_t) - Q_2(\mathcal{H}_t) = \mathbb{E}_{\widetilde{\mathcal{H}}|\mathcal{H}_t}\Big[\mathbb{E}_{\widetilde{\mathcal{H}}|\mathcal{H}_{t+\delta_{\widetilde{\mathcal{H}}}(t)}=\mathcal{H}_t\cup\widetilde{\mathcal{H}}_{(t,t+\delta_{\widetilde{\mathcal{H}}}(t)]}^{\backslash a_{\mathrm{obs}}}}\Big[$$
$$\mathbb{E}_{\widetilde{\mathcal{H}}|\mathcal{H}_{t+\delta_{\widetilde{\mathcal{H}}}(t)+\delta_{\widetilde{\mathcal{H}}}(t+\delta_{\widetilde{\mathcal{H}}}(t))}=\mathcal{H}_t\cup\widetilde{\mathcal{H}}_{(t,t+\delta_{\widetilde{\mathcal{H}}}(t)]}^{\backslash a_{\mathrm{obs}}}\cup\widetilde{\mathcal{H}}_{(t+\delta_{\widetilde{\mathcal{H}}}(t),t+\delta_{\widetilde{\mathcal{H}}}(t)+\delta_{\widetilde{\mathcal{H}}}(t+\delta_{\widetilde{\mathcal{H}}}(t)))}^{\backslash a_{\mathrm{obs}}}}\Big[\ldots$$
$$Q_1(\mathcal{H}_t \cup \widetilde{\mathcal{H}}_{(t,t+\delta_{\widetilde{\mathcal{H}}}^d(t)]}^{\backslash a_{\mathrm{obs}}}) - Q_2(\mathcal{H}_t \cup \widetilde{\mathcal{H}}_{(t,t+\delta_{\widetilde{\mathcal{H}}}^d(t)]}^{\backslash a_{\mathrm{obs}}})\Big]\Big]\Big],$$

where we denote the $d$-th disagreement under the nested sampling above by $\delta_{\widetilde{\mathcal{H}}}^d(t)$. Notice that again there is slight abuse of notation here, since we reused the notation $\widetilde{\mathcal{H}}$ in the expectations above, where the different $\widetilde{\mathcal{H}}$ appearing above may not be identical trajectories. It holds that $\lim_{d\to\infty}\delta_{\widetilde{\mathcal{H}}}^d(t) = T$ with probability 1, since the number of events in a trajectory is finite with probability 1. Hence we have $\lim_{d\to\infty}Q_1(\mathcal{H}_t \cup \widetilde{\mathcal{H}}_{(t,t+\delta_{\widetilde{\mathcal{H}}}^d(t)]}^{\backslash a_{\mathrm{obs}}}) - Q_1(\mathcal{H}_t \cup \widetilde{\mathcal{H}}_{(t,t+\delta_{\widetilde{\mathcal{H}}}^d(t)]}^{\backslash a_{\mathrm{obs}}}) = Q_1(\mathcal{H}_t\cup\widetilde{\mathcal{H}}_{(t,T]}^{\backslash a_{\mathrm{obs}}}) - Q_1(\mathcal{H}_t\cup\widetilde{\mathcal{H}}_{(t,T]}^{\backslash a_{\mathrm{obs}}})$, and may conclude that because $Q_1(\widetilde{\mathcal{H}}_T) = Q_2(\widetilde{\mathcal{H}}_T) = \mathbb{E}[Y|\widetilde{\mathcal{H}}_T]$ for full trajectories $\widetilde{\mathcal{H}}_T$, the functions $Q_1(\cdot)$ and $Q_2(\cdot)$ must coincide,

$$Q_1(\mathcal{H}_t) - Q_2(\mathcal{H}_t) = \mathbb{E}_{\widetilde{\mathcal{H}}|\mathcal{H}_t}[Q_1(\mathcal{H}_t \cup \widetilde{\mathcal{H}}_{(t,T]}^{\backslash a_{\mathrm{obs}}}) - Q_2(\mathcal{H}_t \cup \widetilde{\mathcal{H}}_{(t,T]}^{\backslash a_{\mathrm{obs}}})] = 0.$$

$\square$