# OpenReview forum: "Time After Time: Deep-Q Effect Estimation for Interventions on When and What to do"
_ICLR.cc/2025/Conference — ICLR 2025 Poster_

### Official Review · Reviewer_jyT3 · 2024-10-27

**Soundness:** 3
**Presentation:** 2
**Contribution:** 2
**Rating:** 6
**Confidence:** 3

**Summary:**

This paper proposes Earliest Disagreement Q-Evaluation (EDQ), a method for estimating causal effects of sequential treatments with irregular timing. The key contribution is adapting Q-learning approaches to continuous-time settings while maintaining model-free properties. The work focuses on healthcare and financial applications where treatment timing is crucial.

**Strengths:**

1. The paper addresses a crucial and previously underexplored problem in causal inference: estimating the effect of interventions on treatment timing in sequential decision-making. This problem has practical implications in healthcare and finance, where treatment timing often has substantial impact on outcomes. The authors provide compelling motivation and clearly articulate why existing methods are insufficient.

2. The proposed EDQ method demonstrates impressive scalability and flexibility by being compatible with modern deep learning architectures like transformers. This is a significant advancement over existing methods that either require time discretization or don't scale to large models, making the approach more practical for real-world applications.

**Weaknesses:**

1. The paper's title promises insights into "When and What to Do," but primarily delivers a method for evaluating pre-specified timing policies. While the proposed EDQ method effectively handles off-policy evaluation of treatment timing effects, it provides no framework for discovering optimal timing strategies.  The experimental section further highlights this gap, focusing solely on estimation accuracy rather than demonstrating practical utility in finding better treatment schedules. Additionally, despite the title's suggestion, the paper explicitly omits treatment selection (the "what" aspect) to focus on timing, making the scope narrower than advertised.

2. The experimental validation relies on overly simplified simulation settings, particularly in modeling treatment delays with an exponential distribution  $\delta\sim exp(\lambda_a)$. Although the method's theoretical guarantees don't depend on specific distributional assumptions, the paper would be stronger with experiments using more realistic delay distributions and multiple vital signs. This limitation raises questions about the method's practical performance in real-world healthcare settings where treatment timing follows more complex patterns.

3. The technical presentation has several gaps that affect its accessibility and reproducibility. The paper lacks analysis of computational complexity and clear guidelines for practical implementation.

**Questions:**

1. Definition 5, which is the most important definition, is not clear. For instance, what does $\tilde P_t^a$ mean? I cannot find its definition throughout the whole article including appendix.

---

> ### Author Response · Authors · 2024-11-25
> **Response to Reviewer jyT3 (1/2)**
>
> We thank the reviewer for engaging with our paper and for the valuable comments. To respond on the questions and comments:
>
> **Paper title promises insights on When and What to do, but primarily delivers evaluation of pre-specified timings.**
> Regarding the evaluation of pre-specified policies, we titled our work “**Effect Estimation** for Interventions on When and What to do” because we focus on evaluation and not on policy optimization. It is common to handle these problems separately, and off-policy evaluation is a common standalone topic for works [1,2,3]. The conceptual leap in adapting EDQ to perform policy optimization instead of evaluation is not large, as policy iteration can be done based on an evaluation procedure, similarly to the way it is done in discrete time. However we believe this deserves separate work that deals with choices around policy optimization such as how to specify a policy that is amenable to optimization etc.
>
> As for interventions on “what” to do, we started with interventions on “when” because this is an aspect that is often unaddressed in existing literature. That said, our revision includes simulations where the policies also intervene on the “what” part, corresponding to the type of
> treatment (radiotherapy, chemotherapy, combined therapy) in the newly added tumor growth simulation. While this part does not require novel methodology beyond the one we introduced for intervention on times, it demonstrates that the method can be applied to this scenario.
>
> **Experimental validation relies on an overly simplified setting:** Please see the general comment for the cancer simulation we experimented with. In this simulation, processes draw treatments based on the past observed tumor volumes, which the reviewer may find more realistic, and it has been used in other works on effect estimation with neural nets [4,5]. We are also working to add multiple vital signs in our time-to-failure simulation to show that the method is able to cope with multivariate covariates; these will be included in the next revision.
>
> **Accessibility and reproducibility:** Please see the general comment for details on computational complexity. The complexity of EDQ and FQE are similar, please note that FQE is scalable and has been used in various large scale RL problems (see e.g. [6] for an evaluation using it). The only difference in computation time per iteration is due to sampling from the target policy in order to draw the treatments used in the $Q$ update. In most applications, the added complexity due to this difference is small w.r.t the cost of evaluating the $Q$-function, and in turn the cost of function evaluation is the same for FQE and EDQ. The computational complexity of sampling from the policy depends on how we represent and implement it. For instance, policies may be specified in terms of their intensity functions which requires sampling using the thinning algorithm [7], with neural networks that output the time-to-next-event (e.g. [8] for one example among networks that predict time), or with closed-form decision rules. For instance, in our first simulation we sample exponential variables from times when a feature crosses a certain threshold, and in the cancer simulation we sample actions from a discrete time policy (which means EDQ needs to generate a few more samples until disagreement is achieved, instead of FQE that samples one). In both simulations the time is negligible w.r.t to the evaluation of the $Q$ function.
>
> As for guidelines on practical implementation, we will release our code upon publication to enable reproducibility.
>
> **Definition of $\tilde{P}^{a}_t$:** Regrettably, while editing definition 5 we wounded up overcomplicating the term $\tilde{P}^{a}_t$. A simple and well-posed definition is as follows: $\tilde{P}^{a}_t(u \vert \mathcal{H})$ is a point process on the interval $(t, T]$ with intensity $\lambda^a(u \vert \mathcal{H}_u)$, i.e. the marginal intensity of the treatments under the policy we wish to evaluate (Given by $\lambda^a$. We may also include the distribution $\pi(a \vert \mathcal{H}_u)$ for the mark of the treatment). The reason we include the tilde symbol in $\tilde{P}$ is to emphasize that at each $u\in{(t, T]}$ the distribution conditions on $\mathcal{H}_u$, the *observed* trajectory up until that time. That is to discern this type of sampling from sampling entire trajectories of treatments and observations from the interventional distribution $P$. We amended definition 5 (now definition 4 in the revision) accordingly.

---

> > ### Comment · Reviewer_jyT3 · 2024-11-27
> >
> > Thank you for your response. I will improve my scores.

---

> > > ### Author Response · Authors · 2024-11-28
> > >
> > > Thank you very much for the helpful comments and for revising your score. We sincerely appreciate the time and effort put into the review and to taking our response into account.

---

> ### Author Response · Authors · 2024-11-25
> **Response to Reviewer jyT3 (2/2)**
>
> **References**
>
> [1] Oberst, Michael, and David Sontag. "Counterfactual off-policy evaluation with gumbel-max structural causal models." International Conference on Machine Learning. PMLR, 2019.
>
> [2] Thomas, Philip, and Emma Brunskill. "Data-efficient off-policy policy evaluation for reinforcement learning." International Conference on Machine Learning. PMLR, 2016.
>
> [3] Precup, Doina, Richard S. Sutton, and Satinder P. Singh. "Eligibility Traces for Off-Policy Policy Evaluation." Proceedings of the Seventeenth International Conference on Machine Learning. 2000.
>
> [4] Seedat, Nabeel, et al. "Continuous-Time Modeling of Counterfactual Outcomes Using Neural Controlled Differential Equations." International Conference on Machine Learning. PMLR, 2022.
>
> [5] Vanderschueren, Toon, et al. "Accounting for informative sampling when learning to forecast treatment outcomes over time." International Conference on Machine Learning. PMLR, 2023.
>
> [6] Voloshin, Cameron, et al. "Empirical Study of Off-Policy Policy Evaluation for Reinforcement Learning." Thirty-fifth Conference on Neural Information Processing Systems Datasets and Benchmarks Track (Round 1).
>
> [7] Lewis, PA W., and Gerald S. Shedler. "Simulation of nonhomogeneous Poisson processes by thinning." Naval research logistics quarterly 26.3 (1979): 403-413.
>
> [8] Nagpal, Chirag, Vincent Jeanselme, and Artur Dubrawski. "Deep parametric time-to-event regression with time-varying covariates." Survival prediction-algorithms, challenges and applications. PMLR, 2021.

---

### Official Review · Reviewer_WaHo · 2024-11-02

**Soundness:** 4
**Presentation:** 3
**Contribution:** 3
**Rating:** 6
**Confidence:** 4

**Summary:**

This paper presents a novel approach for handling irregular timing in off-policy evaluation within sequential decision-making frameworks. It extends the classical Bellman equation in fitted Q-evaluation (FQE) to account for cases with irregular time intervals and introduces the concept of measuring the "earliest disagreement time" in Q-function updates.

Overall, the paper is well-written, with a natural flow that guides the reader from foundational ideas to specific methods. While reading the introduction, I was somehow expecting a real-data analysis that could demonstrate the approach, perhaps with an example like ASCVD control or another scenario where OPE might reveal insights under varied intervention timings. I understand the challenges of conducting OPE with observational data, so this is not a request for inclusion in the rebuttal. However, such an analysis could add an interesting dimension to showcase differences in performance in real-world applications and offer potential interpretations. Some terms and assumptions could benefit from clearer explanations, but overall, this paper provides a practical and effective method for handling irregular time in OPE without imposing heavy modeling or causal assumptions.

**Strengths:**

1. The paper’s motivation is well-illustrated with examples. I appreciate the effort to clarify the setup in Section 2-3, which makes the new approach easier to understand. The comparison with classical FQE and the challenges outlined at the end of Section 3 are also helpful.

2. The simulation studies demonstrate promising results, effectively showcasing the benefits of the proposed EDQ by comparing it with FQE and MC methods.

**Weaknesses:**

1. Some notations are unclear on first mention. Here are a few areas where clarification could improve readability (please correct me if I missed something):

 (a) In Definition 4, Line 170, does $\ll$ in $P\ll P_{obs}$ denote absolute continuity? It might be helpful to define this symbol for readers who may not be familiar with the concept.

 (b) In Theorem 1, Line 251, assumptions 1-3 don’t appear to be clearly defined or referenced in Section 2.2.

 (c) In Algorithm 2, Line 279, the loss function $l(Q_t,y)$ seems to be undefined. While it may be user-specified and flexible, adding a brief note about its role and possible choices could provide helpful context.

**Questions:**

1. How does the computational time of your algorithm compare with classical FQE?

2. In cases where stages are regularly spaced in time, would EDQ reduce to classical FQE? A brief discussion or proof in the paper about how EDQ behaves in the limit as time intervals become regular would suffice.

3. In practical applications, a common workaround for irregular timing is to include the interval between the $k-1$th and the $k$th stages as a component of the state $\boldsymbol{X}$, then proceed with classical FQE for Q-function estimation. While this may lack methodological novelty, industry practitioners often find it to be a straightforward and effective way to account for "irregular time" in value estimation. Could the authors elaborate on the motivation for considering "earliest disagreement times" and explain situations where this approach might perform better?

---

> ### Author Response · Authors · 2024-11-25
> **Response to Reviewer WaHo (1/2)**
>
> We thank the reviewer for the useful suggestions and for the positive review. Please find our response to each point below:
>
> **Notation comments:** These are great points, indeed $\ll$ denotes absolute continuity, we added a clarification for that. The reference to assumptions 1-3 became inconsistent due to our final edits and we apologize for that, the assumptions are the ones detailed in section 2.2, and we fixed the theorem statement to make the reference to these assumptions precise. Finally, we agree regarding the loss function, which has to be set such that it yields the conditional expectation at optimality. We specified this as the squared loss for simplicity, where for categorical outcomes this would be set to cross entropy.
>
> **Practical workaround for Irregular Time?** This is a crucial question, hopefully our answer will illustrate the details that necessitate EDQ instead of this type of simpler workaround. *Apologies in advance*, OpenReview did not allow for complex subscripts, so in the following $j := i+1$.
>
> The suggestion for a practical workaround is to include time (or time gap) into the state $X$, i.e. the states at the $i$-th place of the trajectory are now $(t_i, \mathbf{x}_i)$. Our goal is to intervene on time, therefore time also has to be included in the treatment $A$. Continuing with this workaround, actions are of the form $(t_i, \mathbf{a}_i)$ and updates in FQE are along the form that fits $Q(t_i, \mathbf{x}_i, \mathbf{a}_i)$ to the quantity $y_j + Q(t_j, \mathbf{x}_j, \mathbf{a}_j)$.
>
> The workaround becomes problematic once we intervene on time $t_i$ and wish to formulate a correct $Q$ update. For instance, if we intervene on time $t_j$ and change it to $\tilde{t}_j$, then how should we set the state $\mathbf{x}_j$ in the $Q(t_j, \mathbf{x}_j, \mathbf{a}_j)$ term of the update (or the reward $y_j$)? We cannot simply keep it as $\mathbf{x}_j$ since the intervention on time may change the distribution of this feature. As an alternative, we could take further steps and define some null tokens for $\tilde{\mathbf{x}}_j$ where appropriate, and analyze cases where we intervene on time to set it later than observed events in the trajectory. Following this line of thought would likely eventually lead to a solution like EDQ. However, EDQ is not equivalent to the workaround suggested by the reviewer that runs FQE on adjusted feature spaces. We would be happy to hear whether this provides a full answer to the reviewer’s question.
>
> **Reduction to FQE in Discrete Time?** EDQ does not reduce to FQE in discrete time, as the answer to the previous question may hint. Instead, EDQ in discrete time, like its generalization to continuous time, is an update rule based on disagreement between the target policy and the observations. FQE and EDQ in discrete time do not apply the same update to $Q_t$ whenever the action sampled from the target policy for time $t+1$ is the same as the one in the observed trajectory. For completeness, we derive the discrete time equation for EDQ in the appendix after the proof of Theorem 1. The resulting algorithm for discrete time is perhaps closest in spirit to $N$-step off-policy evaluation methods in Reinforcement Learning (though we are not aware of works that study an earliest disagreement algorithm), which showed some favorable results in the past [1,2]. We choose to focus on the irregular sampling case, since it seems like the scenario where the earliest disagreement approach is most warranted, and also since the setting is more general than discrete time.
>
> **Computational complexity w.r.t FQE:** A response to this question is included in the general response. We will reiterate some points from it here, and include additional details.
> The only difference between EDQ and FQE in computation time per iteration is due to sampling from the target policy in order to draw the treatments used in the $Q$ update. In most applications, the added complexity due to this difference is small w.r.t the cost of evaluating the $Q$-function, and in turn the cost of function evaluation is the same for FQE and EDQ. The computational complexity of sampling from the policy depends on how we represent and implement it. For instance, policies may be specified in terms of their intensity functions which requires sampling using the thinning algorithm [3], with neural networks that output the time-to-next-event (e.g. [4] for one example among networks that predict time), or with closed-form decision rules. For instance, in our first simulation we sample exponential variables from times when a feature crosses a certain threshold, and in the cancer simulation we sample actions from a discrete time policy (which means EDQ needs to generate a few more samples until disagreement is achieved, instead of FQE that samples one). In both simulations the time is negligible w.r.t to the evaluation of the $Q$ function.

---

> > ### Comment · Reviewer_WaHo · 2024-11-25
> >
> > I appreciate the authors' detailed responses to my questions. These points addressed my concerns. Therefore, I maintain my positive score for this paper.

---

> > > ### Author Response · Authors · 2024-11-26
> > >
> > > We are glad to learn that our responses addressed your concerns, thank you very much for engaging in the discussion.

---

> ### Author Response · Authors · 2024-11-25
> **Response to Reviewer WaHo (2/2)**
>
> **References**
>
> [1] Munos, R., Stepleton, T., Harutyunyan, A., and Bellemare, M. (2016). Safe and efficient off-policy
> reinforcement learning. Advances in neural information processing systems, 29.
>
> [2] Precup, D. (2000). Eligibility traces for off-policy policy evaluation. Computer Science Department Faculty Publication Series, page 80.
>
> [3] Lewis, PA W., and Gerald S. Shedler. "Simulation of nonhomogeneous Poisson processes by thinning." Naval research logistics quarterly 26.3 (1979): 403-413.
>
> [4] Nagpal, Chirag, Vincent Jeanselme, and Artur Dubrawski. "Deep parametric time-to-event regression with time-varying covariates." Survival prediction-algorithms, challenges and applications. PMLR, 2021.

---

### Official Review · Reviewer_u2Uo · 2024-11-03

**Soundness:** 3
**Presentation:** 4
**Contribution:** 3
**Rating:** 8
**Confidence:** 3

**Summary:**

This paper introduces off-policy evaluation through the framework of stochastic point processes and presents Earliest Disagreement Q-evaluation (EDQ) as a method for estimation. Under the causal validity assumption, the authors demonstrate that EDQ can accurately identify the true policy value. Additionally, this assumption is elucidated using a local independence graph. Experiments conducted on simulated data confirm the effectiveness of the proposed approach.

**Strengths:**

**1**. The formulation of sequential causal effect estimation using point processes is novel, effectively addressing the irregular nature of treatments and covariates in various real-world applications.

**2**. The EDQ method for estimation is compelling, with corresponding consistency results rigorously demonstrated in Theorem 1.

**3**. The assumptions required for causal identification are elucidated through a local independence graph, providing an intuitive understanding of their implications.

**Weaknesses:**

Experiments should be conducted on real-world datasets or, at the very least, on simulated datasets generated from real-world data to provide more convincing validation.

**Questions:**

**1**. To my knowledge, consistency and the SUTVA assumptions are also essential in causal effect estimation. Why are these assumptions not required in your work?

**2**. What is the impact of different sequential modeling architectures (such as Transformers, RNNs, regression models, etc.) on your method? Is the estimation sensitive to the choice of architecture and hyperparameters?

---

> ### Author Response · Authors · 2024-11-25
> **Response to Reviewer u2Uo**
>
> We thank the reviewer for the positive review and useful comments. Our response to questions is below:
>
>
> **Experiments:** Please see the general response, we included an additional simulation from a realistic tumor growth simulator, also used in other papers on the topic (e.g. [1,2] use this simulator as their only experimental evaluation).
>
> For real-world data, while we plan to do future work on finding suitable data, e.g. from existing experiments, please note that to the best of our knowledge there is no such data currently available that is being used in papers on the topic. We will work to have another experiment that is semi-synthetic for the next revision (after the discussion period), e.g. by taking vitals from MIMIC and introducing synthetic treatments and outcomes, but we must emphasize that such a setting is inherently limited when the task involves sequential decisions. This is because a treatment $a$ at time $t$ cannot affect the covariates $x$ at time $u>t$, as this yet again requires a simulation (to form the subsequent covariates affected by $a$). The consequence is that we must select between working in a fully-synthetic setting (i.e. defining synthetic effects on future vitals), or having treatments that only affect future outcomes and treatments, without affecting future vitals. The latter option is somewhat unrealistic, and eliminates aspects of planning from the problem. Considering these caveats, we opted, like several other works in the field, to work in a fully synthetic setting that is based on a realistic simulator that is used in the pharmaceutical industry.
>
> **Why are consistency and SUTVA not mentioned?** The formalism we use in our section on identifiability is not based on potential outcomes notation, but rather on the graphical approach. As Pearl [1, p.128] comments, it is possible to show that assumptions included in SUTVA such as consistency, are automatically satisfied in the structural interpretation of causality. Upon reviewing the literature, there are works that explicitly mention SUTVA and those that assume the graphical interpretation. We fell into the latter, but glad to explicitly mention it if you find this important.
>
>
> **Impact of architecture and hyperparameters:** In this work our main focus is on laying out the formal framework and methodology to tackle the estimation problem on treatment times. From the experimental point of view, this involved modifying the transformer to suit our task and algorithm (that is after validating our solution with toy linear models that are not included in the paper). This includes defining different types of events, embedding times and time differences, and including a target network in the training procedure. Our experiments with hyperparameters were thus confined to the variations of these modifications, where the goal was to create the simplest working version of the algorithm. While we ended up choosing rather standard hyperparameters (e.g. learning rate of $1e-3$ with an adam optimizer, soft-Q updates with rate 0.001-0.01 etc.), some of the choices did have an impact on results, e.g. larger batch sizes helped in gaining more stabilized optimization. We will include the tools to reproduce all these experiments in our codebase. Since $Q$-methods are used with many architectures in the RL literature, we have careful optimism that the algorithm can be successfully applied to other sequence modeling architectures besides the transformer we use here, but we leave this exploration for future work. If there is a specific architecture that the reviewer would find helpful for comparison, we would be happy to take suggestions and include it in our next revision.
>
> [1] Pearl, Judea. "Causal inference in statistics: An overview." (2009): 96-146.

---

> > ### Comment · Reviewer_u2Uo · 2024-11-27
> >
> > I thank the authors for the response. I will keep my positive assessment.

---

> > > ### Author Response · Authors · 2024-11-28
> > >
> > > Thank you very much for the time and effort put into the review and the engagement in the discussion.

---

### Official Review · Reviewer_5s6h · 2024-11-03

**Soundness:** 3
**Presentation:** 2
**Contribution:** 3
**Rating:** 6
**Confidence:** 3

**Summary:**

The paper proposes a method for estimating the causal effect of policies using Earliest Disagreement Q-Evaluation (EDQ), where a policy determines both the action and its timing. Specifically, the authors aim to estimate the outcome of a target policy by leveraging dynamic programming, extending fitted Q-evaluation from discrete to continuous time. The update function is driven by the earliest point of disagreement between the behavior policy and the target policy.

**Strengths:**

- The paper considers the problem of evaluating a policy using continuous-time causal inference, focusing on off-policy evaluation.
- To tackle the challenges of policy evaluation in continuous time, where the conventional Q-function collapses, the paper leverages the property of countable decision points in a point process and introduces a simple yet efficient method based on earliest disagreement times.
- The paper also provides identifiability conditions for the causal estimands, ensuring accurate estimation.

**Weaknesses:**

- Could the authors discuss the connection between the Bellman equation in discrete time and Theorem 1? Equation (1) is equivalent to the Bellman equation for a finite-horizon problem in discrete times, and Theorem 1 appears to be a direct extension using differential equations and stopping times.
- The proposed framework allows the trajectory outcome to be a sum of time-specific outcomes (lines 79, 204-205). Are there any constraints on the number of observations in the trajectory? If the total number of $k$ is indefinite, the scale of $Y$ will vary with the number of observed time-specific outcomes, potentially affecting the outcome’s interpretation and stability.
- Writing improvement suggestions
  - In the second paragraph of Section 1, the motivation could be clearer by explicitly stating the challenges related to sequential treatment and irregular timing, with a particular emphasis on the primary challenge about decision times. In the current introduction, it is not immediately clear to me which difficulties are being discussed in lines 37-39.
  - In the third paragraph of Section 1, it would help to introduce the components of an intervention. From my understanding, there are two key components: the timing of the intervention and the intervention itself. Otherwise, readers might not yet understand whether treatment timing in line 49 is determined by the environment or chosen by the policy.
  - In line 79, $Y_k$ has not been defined yet.
  - At the end of line 88, should $d N(t)$ be $d N^l (t)$?
  - Could you clarify the distinction between a marked decision point process in Definition 1 and a marked temporal point process (Upadhyay et al. 2018)?
  - A period is missing from line 103.
  - In line 105, should each trajectory be indexed by $i$ or $m_i$? It seems that $m$ is the total number of trajectories and $i$ takes values from $[m]$.
  - What are assumptions 1-3 in line 251?
  - In equation (2), should the right-hand side condition on $\mathcal{H}_t$ as in equation (1)?

**Questions:**

See above.

---

> ### Author Response · Authors · 2024-11-25
> **Response to Reviewer 5s6h (1/2)**
>
> We thank the reviewer for the comments and hope that the responses below answer the questions they raise.
>
> **Connection between Bellman Equation in discrete time and Theorem 1:** To keep the discussion concise, we wish to note that equation 1 is an identity on conditional expectations (we used the name tower property), while the name Bellman equation is often used to refer to an optimality equation that is similar but with a value function that maximizes over the actions. While clearly the two are related, we state this to avoid any inaccuracies in the discussion, as our work studies evaluation and not optimization of policies.
>
> As the reviewer suggested, if we take the continuous time analogue of the discrete time equation 1, we can arrive at a stochastic differential equation that is satisfied by the $Q$-function. However, theorem 1 is not that equation, and it is also not a differential equation. The relation between the equation we derive in theorem 1 and the tower property of conditional expectations/Bellman equation, is perhaps best understood by examining the discrete time version of Theorem 1, which we derive in the appendix (it is referenced in lines 257-258 of the main paper in the original submission). The derived equation in discrete time is not equation 1, but rather an earliest disagreement in discrete time, and the closest methods in the literature are $N$-step policy evaluation methods in Reinforcement Learning, e.g. [1, 2] which is also mentioned in section 3 of the main paper. We have better clarified in the revision.
>
> As mentioned in the review, we take advantage of the countable amount of jump times and write the equation that is stopped at time of the earliest disagreement. The benefit of deriving Theorem 1 and not a stochastic differential equation obtained from finding an analogue of equation 1 in continuous time, is that the equation in Theorem 1 easily leads to an efficient numerical implementation for evaluation of the $Q$ function. Similar to how the discrete time Bellman equation is useful since it easily lends itself to an efficient numerical implementation, while the Hamilton-Jacobi-Bellman differential equation is not straightforward to solve numerically. We would be glad to know whether the response properly addresses the reviewer’s concerns.
>
> **Can the outcome become too large when the trajectory has many events?** The conditions of the theorem, detailed in the appendix, are that times we consider are stopped at time $T$ (i.e. are in the interval $[0, T)$) for some $T>0$ and that the number of events is bounded (this is stated in the literature by taking $T_k \rightarrow_{k\rightarrow \infty} = \infty$, where $T_k$ is the time of the $k$-th event). This means the expected rewards are at the very least non-divergent. Note that we’ve limited the results to population level statements (i.e. identity is given in expectations rather than analyzing finite samples, numerical convergence etc.). Thus they do not capture considerations like numerical stability, handling large rewards, and convergence of $Q$-iterations. To form stable algorithms on cumulative rewards, one might need to introduce discounting such as in other RL algorithms (in our simulation the reward only appears at the end of the trajectory). We added this explanation to the paper, and we thank the reviewer for bringing it to our attention.
>
> **Writing suggestions:** We thank the reviewer, we incorporated the suggestions into the revision  and answer the questions that require clarification below. We would be glad to know if anything remains unclear.
>
> **Distinction between a marked decision point process and a marked temporal point process (Upadhyay et al. 2018):** The mathematical objects we define are the same (up to them studying a Markov process, whereas we do not use this independence constraint in our presentation). In terms of the task studied, they assume online interaction while we study off-policy evaluation. Notice that the title of Upadhyay et al. 2018 is about “Learning of Marked Temporal Point Processes”, which describes well the fact that they learn a policy that is a marked temporal point process. We used the term "decision point process” for the entire joint distribution of policy, covariates and outcomes (instead of just the policy). The latter is in concert with terminology in Reinforcement Learning and sequential decision making, where the distributions are called decision processes instead of just stochastic processes (see e.g. [3]). Thus our choice to name the joint distributions "decision point process” instead of "point processes” seems suitable. We are open to reconsidering this name in case the reviewer has further thoughts about this.

---

> > ### Comment · Reviewer_5s6h · 2024-11-25
> >
> > Thank you for your detailed responses—they are very helpful. In my first point, I was referring to the Bellman expectation equation which uses the tower property, but I appreciate your clarification regarding the discrete-time analogy of Theorem 1. I have thus increased my score.

---

> > > ### Author Response · Authors · 2024-11-26
> > >
> > > Thank you very much for the response and for increasing your score.
> > >
> > > We are glad our clarifications were helpful, and hope the revision adequately resolves the points raised in the review. Thank you for clarifying your intent in the first point as well.

---

> ### Author Response · Authors · 2024-11-25
> **Response to Reviewer 5s6h (2/2)**
>
> **Assumption 1-3 in line 251:** The reference to assumptions 1-3 became inconsistent due to our final edits and we apologize for that, the assumptions are the ones detailed in section 2.2, and we fixed the theorem statement to make the reference to these assumptions precise.
>
> > In line 105, should each trajectory be indexed…
>
> Our apologies, this is a typo, the trajectory should be denoted by $\mathcal{H}_i$ and not $m_i$. We fixed this and thank the reviewer for catching this and for the rest of the comments.
>
> [1] Munos, R., Stepleton, T., Harutyunyan, A., and Bellemare, M. (2016). Safe and efficient off-policy
> reinforcement learning. Advances in neural information processing systems, 29.
>
> [2] Precup, D. (2000). Eligibility traces for off-policy policy evaluation. Computer Science Department Faculty Publication Series, page 80.
>
> [3] Uehara, M., Shi, C., and Kallus, N. (2022). A review of off-policy evaluation in reinforcement
> learning. arXiv preprint arXiv:2212.06355.

---

### Official Review · Reviewer_jmvH · 2024-11-04

**Soundness:** 3
**Presentation:** 3
**Contribution:** 3
**Rating:** 8
**Confidence:** 3

**Summary:**

The paper aims to tackle the temporal irregularity when estimating causal effects of policies, by proposing a method called EDQ (earliest disagreement Q-evaluation) based on dynamic programming. The simulation demonstrates more accurate estimations of EDQ.

**Strengths:**

- The work investigated an important problem, i.e., when to intervene with irregular times.

- The work is well motivated, given many human-related backgrounds such as healthcare and finance requires a carefully evaluation for when to provide proper intervenes.

- The paper is well structured and polished.

**Weaknesses:**

- Given the work is called into category of off-policy evaluation. Related works in off-policy evaluation should be thoroughly discussed in the paper, either in Introduction or Related work. I understand there can exist some key differences between traditional off-policy evaluation and EDQ, but should be carefully and thoroughly discussed and compared in experimental settings. Also, there exists work regarding when-to-treat problem (e.g., [1]). A further comparison and discussion regarding EDQ and those works would be great.

- Since one major motivation of the work is human-related when-to-treat problem, and the paper used a lot of healthcare examples (which is comprehensive). I’m curious about whether the work can be examined on some related settings. It’s understandable that running real-world experiments would not be feasible and high-stake, but it would be more impressive to provide experiments on some empirical motivated settings, e.g., sepsis [2], autism [3], etc.




- Minors:

It would be great if code and/or data can be released for reproducibility.

Line 99. a was defined as action and then represented number of multivariate unobserved process

Line 185. FQE needs careful citations, given it’s a well-established work in off-policy evaluation and optimization.

Line 308, 355, 363. Lower cases for bolded words?

[1] Learning When-to-Treat Policies

[2] Counterfactual Off-Policy Evaluation with Gumbel-Max Structural Causal Models

[3] Off-policy Policy Evaluation For Sequential Decisions
Under Unobserved Confounding

**Update after rebuttal**:

The authors conducted further healthcare analysis to support their major contribution, and further discussions regarding related works.

**Questions:**

Please see Weaknesses.

---

> ### Author Response · Authors · 2024-11-25
> **Response to Reviewer jmvH**
>
> We thank the reviewer for the valuable comments and for appreciating positive aspects of our work. Below we provide responses to the comments and questions raised by the reviewer.
>
> **Experiments on related settings.** We thank the reviewer for the great suggestion and reference to simulators. As mentioned in the general response, we added an experiment with a cancer simulator that is popular in works on causal inference with sequential (but non-dynamic) treatments [1, 2], and used it to study evaluation of dynamic treatments. It has also been studied under an irregular sampling setting, which makes it rather suitable for our work. We hope that this addresses concerns about required experiments.
> We have also referred to the simulators suggested by the reviewer on sepsis and diabetes [3,4] as possible additional settings.
>
> **Discussion on Off-Policy Evaluation Literature.** We thank the reviewer for pointing out references. Specifically, the seminal when-to-treat work should have been mentioned and was regrettably left out of our discussion. It deals with regret bounds and doubly-robust estimation when a single discrete treatment start (or stopping) time is scheduled. This is different from the problem we study, which seeks to schedule multiple sequential treatments in continuous time. However, it would be interesting if future work can adopt ideas like the regret bounds, and advantage learning with “universal” nuisances to the irregularly sampled setting. We did our best to provide an adequate background on all the related fields, but are happy to take further suggestions from the reviewer on other work to include. Please note that many of the algorithmic components like doubly robust estimation are complementary to our contribution, and in principle can be incorporated on top of it. We think that since our contribution is a “direct method” for irregularly sampled data, it makes the most sense to compare it with the direct method in discrete time (i.e. FQE). If there are additional baselines that the reviewer thinks can better support the results, then we are happy to incorporate them in our experiments.
>
> **Response to minor comments:**
>
> > Code release.
>
> The code will be released upon publication.
> > $a$ was defined as action and then represented number of multivariate unobserved process.
>
> We might be misunderstanding the comment, does this refer to the sentence “and a multivariate unobserved process with intensity $\lambda^u$”? If so then ‘a’ is not a number here, it is just the word “a”.
>
> > FQE citations.
>
> Thank you for the comment, we included Watkins and Dayan [5] and Le et al. [6] as citations in our revision. We are happy to take other suggestions on this as well.
> > Lower cases for bolded words.
>
>  Thanks, this is fixed in our revision.
>
> [1] Seedat, Nabeel, et al. "Continuous-Time Modeling of Counterfactual Outcomes Using Neural Controlled Differential Equations." International Conference on Machine Learning. PMLR, 2022.
>
> [2] Vanderschueren, Toon, et al. "Accounting for informative sampling when learning to forecast treatment outcomes over time." International Conference on Machine Learning. PMLR, 2023.
>
> [3] Oberst, Michael, and David Sontag. "Counterfactual off-policy evaluation with gumbel-max structural causal models." International Conference on Machine Learning. PMLR, 2019.
>
> [4] Namkoong, Hongseok, et al. "Off-policy policy evaluation for sequential decisions under unobserved confounding." Advances in Neural Information Processing Systems 33 (2020): 18819-18831.
>
> [5] Watkins, C. J. and Dayan, P. (1992). Q-learning. Machine learning, 8:279–292.
>
> [6] Le, H., Voloshin, C., and Yue, Y. (2019). Batch policy learning under constraints. In International
> Conference on Machine Learning, pages 3703–3712. PMLR.

---

> > ### Comment · Reviewer_jmvH · 2024-11-27
> > **Thank you for the response**
> >
> > Dear authors, I would like to thank you for the time and efforts to answer my questions. I'd happy to increase my score.

---

> > > ### Author Response · Authors · 2024-11-28
> > >
> > > Thank you very much for the helpful comments and for revising your score. We greatly appreciate the time and effort put into the review.

---

### Official Review · Reviewer_vFET · 2024-11-08

**Soundness:** 2
**Presentation:** 3
**Contribution:** 3
**Rating:** 6
**Confidence:** 3

**Summary:**

The authors propose a method for estimating the effect of a continuous-time treatment policy, defined as a treatment intensity function (when to treat) coupled with a mark distribution (what treatment to select), from observational data. Their approach, Earliest Disagreement Q-Evaluation (EDQ), is based on the principle that effects of observed versus proposed policies should not diverge until the point when the two policies disagree. The authors first review the identifiability of causal effects in this setting, then present the proposed estimator and associated algorithm. They construct a simulation setting in which a supposed vital sign is responsive to treatment, with diminishing returns. They show that EDQ estimates the effect of alternative (non-observed) treatment policies more accurately than comparator methods in this setting.

**Strengths:**

The authors tackle a novel problem and present an elegant solution. Identifiability of causal effects follows from previous results, and is presented clearly. The related work section is outstanding: comprehensive, highly informative, and well-written. The simulation setup is intuitive and clearly presented, and the proposed method outperforms alternatives.

**Weaknesses:**

- The key weakness is that the experimental results are underdeveloped. I would have liked to see more variations on the simulation results as well as application to at least one real dataset.

- I was confused by the notation and presentation in Definition 5, which then made it difficult to understand the details of EDQ. Specifically, I am not clear on how $\tilde{P}_t^a$ is defined or how to sample from it. I have lowered my score for this reason, but I'm happy to increase it if the authors clarify the presentation and/or if other reviewers did not have similar difficulty.

- While defining the policy as an intensity (i.e. rate) is interesting, I have a hard time imagining a realistic scenario where it would make sense to sample treatment times rather than deciding whether/how to treat at fixed or given intervals.

- (minor) The title ("When and What") is a bit misleading. I take the point that learning when to treat (i.e., the rate portion of the policy) is the challenging part, but saying both when and what in the title is misleading, since the paper and experimental results focus only on the former.

- (minor) There are quite a few typos / minor errors, particularly in later sections of the paper.

**Questions:**

- Line 89: Why do these assumptions imply that the process is self-exciting? Please elaborate.
- How well does the algorithm scale? Could the authors comment on the computational complexity?

---

> ### Author Response · Authors · 2024-11-25
> **Response to Reviewer vFET (1/2)**
>
> We thank reviewer vFET for the careful reading of our paper, and for appreciating the solution we present, and the identifiability, related work and simulation sections. We are further grateful for the valuable comments that helped us improve parts of the paper. Below we comment on each of these points.
>
> **Experimental results underdeveloped:** Please see the general response. We’ve added experiments on a cancer simulator developed in [1] and used in other papers on causal inference with irregularly sampled data, e.g. [2,3]. Note that [2,3] do not evaluate dynamic policies, and also suffice with this dataset for the purpose of their simulations. We think our additional simulation has some qualitative differences, e.g. densely sampled observations and irregularly sampled treatments, that complement the cancer simulation well.
>
> As for a real dataset, please note that without performing an experiment in the real world, it is very difficult to obtain a credible “real” dataset. This is further aggravated when we are interested in a scenario of *sequential* decision making like ours. For instance, we might take the route of using real patient vitals from MIMIC-IV, and introducing synthetic treatments and outcomes to form a semi-synthetic dataset. However, then a treatment $a$ at time $t$ cannot affect the covariates $x$ at time $u>t$, as this yet again requires a simulation to reason about counterfactual covariates. The consequence is that we must select between working in a synthetic setting, or having treatments that only affect future outcomes and treatments, without affecting future vitals. Like several other works in the field, we chose the former, and included a more realistic simulator for this purpose.
> A semi-synthetic experiment where treatments do not affect future covariates is rather unrealistic, and eliminates aspects of planning from the problem (i.e. effect estimation in this case can be done without accounting for which state the current treatment may lead us to). Nonetheless, we will work to include such a simulation in our next revision.
>
> **Definition of $\tilde{P}^a_t$:** Regrettably, while editing definition 5 (now definition 4 in our revision) we wounded up overcomplicating the term $\tilde{P}^{a}_t$. A simple and well-posed definition is as follows: $\tilde{P}^{a}_t(u \vert \mathcal{H})$ is a point process on the interval $(t, T]$ with intensity $\lambda^a(u \vert \mathcal{H}_u)$, i.e. the marginal intensity of the treatments under the policy we wish to evaluate (Given by $\lambda^a$. We may also include the distribution $\pi(a \vert \mathcal{H}_u)$ for the mark of the treatment). The reason we include the tilde symbol in $\tilde{P}$ is to emphasize that at each $u\in{(t, T]}$ the distribution conditions on $\mathcal{H}_u$, the *observed* trajectory up until that time. That is to discern this type of sampling from sampling entire trajectories of treatments and observations from the interventional distribution $P$.
>
>
> We emphasize that in practice, this just means taking the observed history up to time $u$ and inserting it into the policy of interest to obtain whether or not there is a treatment in the next increment.
> There are several specific algorithms to do this correctly, depending on how the policy is represented. For instance the thinning algorithm for sampling from point processes when we are given access to the intensity function. We amended the definition accordingly.
>
> **Defining an intensity as policy is interesting, but in reality decisions in fixed intervals are more relevant:** We agree with the reviewer that deciding whether to treat at fixed time intervals is a practically interesting case. Indeed, this is one of the intensities that one could specify for our method (e.g. the cancer simulation is performed in discrete times, but with sparse occurrence of events). There are a few benefits to formalizing the problem with a more general object such as intensities. Besides providing a more honest representation of the real world data generating process in applications like healthcare, the method and formalism allow us to bypass the need to assume a minimal fixed time interval. Instead the data is handled as an “event stream”, as reflected by our choice of architecture (which represents the time gaps between events as features) and other dedicated architectures that model this type of data [4, 5]. Furthermore, it supports the specification of policies in different manners. For instance, we may be explicitly provided with the intensity function as we have in our first simulation; we could specify a discrete time policy; it is also possible to have a neural network that outputs the time of the next treatment; or even a simple decision rule.

---

> ### Author Response · Authors · 2024-11-25
> **Response to Reviewer vFET (2/2)**
>
> **The title is a bit misleading.** Please see the general comment, we added a simulation on what to do by intervening on application of chemotherapy and radiotherapy in the tumor growth example of the experimental section. In addition to having added this, we are also adding an intervention on treatment dosages for our existing time-to-failure experiment. However, this might only be added in our next revision.
>
> **There are quite a few typos toward the end of the paper.** Thank you, we have done a thorough pass to fix grammatical errors and typos in the paper.
>
> **Answers to questions:**
> > Why do these assumptions imply that the process is self-exciting?
>
> Self-exciting here was simply meant to convey that the intensity may depend on the history of the process, not necessarily that jumps at certain times make jumps at subsequent times more likely. The literature uses the term for both meanings. We changed the terminology to clarify that the jumps can depend on history in an arbitrary fashion instead of the narrow definition of strictly increasing intensities upon additional jumps.
>
> > How well does the algorithm scale?
>
> A response to this question is included in the general response. We will reiterate some points from it here, and include additional details.
> * The only difference between EDQ and FQE in computation time per iteration is due to sampling from the target policy in order to draw the treatments used in the $Q$ update. Note that FQE is scalable and has been used in various large scale RL problems (see e.g. [6] for an evaluation using it).
> * In most applications, the added complexity due to this difference is small w.r.t the cost of evaluating the $Q$-function, and in turn the cost of function evaluation is the same for FQE and EDQ. The computational complexity of sampling from the policy depends on how we represent and implement it. For instance, policies may be specified in terms of their intensity functions which requires sampling using the thinning algorithm [7], with neural networks that output the time-to-next-event (e.g. [8] for one example among networks that predict time), or with closed-form decision rules. For instance, in our first simulation we sample exponential variables from times when a feature crosses a certain threshold, and in the cancer simulation we sample actions from a discrete time policy (which means EDQ needs to generate a few more samples until disagreement is achieved, instead of FQE that samples one). In both simulations the time is negligible w.r.t to the evaluation of the $Q$ function.
>
>
> [1] Geng, Changran, Harald Paganetti, and Clemens Grassberger. "Prediction of treatment response for combined chemo-and radiation therapy for non-small cell lung cancer patients using a bio-mathematical model." Scientific reports 7.1 (2017): 13542.
>
> [2] Seedat, Nabeel, et al. "Continuous-Time Modeling of Counterfactual Outcomes Using Neural Controlled Differential Equations." International Conference on Machine Learning. PMLR, 2022.
>
> [3] Vanderschueren, Toon, et al. "Accounting for informative sampling when learning to forecast treatment outcomes over time." International Conference on Machine Learning. PMLR, 2023.
>
> [4] McDermott, Matthew, et al. "Event Stream GPT: a data pre-processing and modeling library for generative, pre-trained transformers over continuous-time sequences of complex events." Advances in Neural Information Processing Systems 36 (2023): 24322-24334.
>
> [5] Mei, Hongyuan, and Jason M. Eisner. "The neural hawkes process: A neurally self-modulating multivariate point process." Advances in neural information processing systems 30 (2017).
>
> [6] Voloshin, Cameron, et al. "Empirical Study of Off-Policy Policy Evaluation for Reinforcement Learning." Thirty-fifth Conference on Neural Information Processing Systems Datasets and Benchmarks Track (Round 1).
>
> [7] Lewis, PA W., and Gerald S. Shedler. "Simulation of nonhomogeneous Poisson processes by thinning." Naval research logistics quarterly 26.3 (1979): 403-413.
>
> [8] Nagpal, Chirag, Vincent Jeanselme, and Artur Dubrawski. "Deep parametric time-to-event regression with time-varying covariates." Survival prediction-algorithms, challenges and applications. PMLR, 2021.

---

> ### Comment · Reviewer_vFET · 2024-11-25
>
> Thanks for your response. I appreciate the updates and clarifications and look forward to the revised version. I have increased my score to Marginal Accept.

---

> > ### Author Response · Authors · 2024-11-26
> >
> > Thank you very much for the response and for updating your score.
> >
> > We think there is a revision available now through the OpenReview page, which includes the updates mentioned in our responses. We hope you find it interesting, and we will upload a more polished revision once revision uploads will become available again. Thanks again for the engagement both in the review and the discussion.

---

### Author Response · Authors · 2024-11-25
**General Response**

We thank the authors for their thorough and insightful reviews. We are glad to see that the reviewers engaged with the paper, recognized that we:
* Tackle an underexplored [jyT3, vEFT, u2UO] and crucial [jyT3, jmvH] problem
* Present an elegant/compelling [vEFT, u2UO], impressively scalable [jyT3] solution
* Elucidate identifiability conditions [vFET, 5s6h, u2Uo]
* Provide intuitive simulations that demonstrate the efficacy of our method [vFET, WaHo].

The comments put forward by the reviewers are insightful and will help us significantly improve the paper. We address them in depth with a response to each review, and summarize below our response to questions that were raised by more than one reviewer.

**Experimental evaluation:** Reviewers commented that additional experiments would be appreciated. In our revision, we included experiments on a cancer growth simulator, following previous work on large scale effect estimation with sequential treatments [1,2,3]. [2,3] also use them to study irregular sampling times, but do not work with dynamic treatments. Reviewers further noted that the experiments do not include an intervention on “what” to do, but only on “when”. In the cancer simulation, policies determine the type of treatment as well as its timing.  Additionally, we are adding an intervention on dosage for our existing time-to-failure simulation, along with higher dimensional covariates. These experiments might not finish until the end of discussion but will be included in our next revision.

**Computational complexity:** reviewers [vFET, WaHo, jyT3] asked that we comment about the computational complexity of EDQ. To give a brief response to this question, we consider two parts that together form an iteration of both EDQ and FQE: (i) evaluating $Q(\mathcal{H}_t)$ and $Q(\tilde{\mathcal{H}}_u)$ where $u=t+\delta$ (for FQE $\delta=1$) and performing a gradient update, (ii) sample actions from the target policy to form $\tilde{\mathcal{H}}$. **In most cases**, where $Q$-functions are parameterized by neural networks, **the first part is substantially more costly than the second**, scaling as $O(d)$ where $d$ is the number of parameters in the network. The cost of sampling an action from a policy in both FQE and EDQ depends on the policy we are interested in and its implementation (e.g. it may involve sampling times from a point process using the thinning algorithm [4], when we are given access to the intensity function), and we specify a few options in the response to reviewers that asked about this. A comment on this, along with examples of how one may implement policies has been added to section 5.1 and the appendix. Apologies for not subscripting $t+\delta$ directly, OpenReview didn't allow complex subscripts.

We appreciate your thorough reviews and are happy to answer any further comments.

[1] Bica, I., Alaa, A. M., Jordon, J., and van der Schaar, M. (2020). Estimating counterfactual treatment outcomes over time through adversarially balanced representations. In International Conference on Learning Representations.

[2] Seedat, Nabeel, et al. "Continuous-Time Modeling of Counterfactual Outcomes Using Neural Controlled Differential Equations." International Conference on Machine Learning. PMLR, 2022.

[3] Vanderschueren, Toon, et al. "Accounting for informative sampling when learning to forecast treatment outcomes over time." International Conference on Machine Learning. PMLR, 2023.

[4] Lewis, PA W., and Gerald S. Shedler. "Simulation of nonhomogeneous Poisson processes by thinning." Naval research logistics quarterly 26.3 (1979): 403-413.

---

### Meta-Review · Area_Chair_7kb3 · 2024-12-21

**Metareview:**

The paper considered estimating the effect of a continuous-time treatment policy, defined as a treatment intensity function (when to treat) coupled with a mark distribution (what treatment to select), focusing on off-policy evaluation. The authors tackle a novel problem and present an elegant solution. The paper leverages the property of countable decision points in a point process and introduces a simple yet efficient method based on earliest disagreement times. The paper also provides identifiability conditions for the causal estimands, ensuring accurate estimation. The simulation setup is intuitive and clearly presented, and the proposed method outperforms alternatives. There are several suggestions and concerns raised by reviewers and the authors addressed them carefully by conducting further healthcare analysis to support their major contribution and further discussions regarding related works, strengthening the experimental results, clarifying the computational complexity, and many others. We appreciate the updates and clarifications and encourage the authors to incorporate all suggestions into the revised version.

**Additional Comments On Reviewer Discussion:**

There are several suggestions and concerns raised by reviewers and the authors addressed them carefully by conducting further healthcare analysis to support their major contribution and further discussions regarding related works, strengthening the experimental results, clarifying the computational complexity, and many others. We appreciate the updates and clarifications and encourage the authors to incorporate all suggestions into the revised version.

---

### Decision · Program_Chairs · 2025-01-22

Accept (Poster)